# MULTI-REDUNET: INTERPRETABLE CLASS-WISE DECOMPOSITION OF REDUNET

**Fengrong Li**
National University of Singapore
e0506462@u.nus.edu

**Delin Chu**
National University of Singapore
matchudl@nus.edu.sg

## ABSTRACT

ReduNet has emerged as a promising white-box neural architecture grounded in the principle of maximal coding rate reduction, offering interpretability in deep feature learning. However, its practical applicability is hindered by computational complexity and limited ability to exploit class-specific structures, especially in undersampled regimes. In this work, we propose Multi-ReduNet and its variant Multi-ReduNet-LastNorm, which decompose the global learning objective into class-wise subproblems. These extensions preserve the theoretical foundation of ReduNet while improving training efficiency by reducing matrix inversion costs and enhancing feature separability. We provide a concise theoretical justification for the class-wise decomposition and show through experiments on diverse datasets that our models retain interpretability while achieving superior efficiency and discriminative power under limited supervision. Our findings suggest that class-wise extensions of ReduNet broaden its applicability, bridging the gap between interpretability and practical scalability in deep learning.

## 1 INTRODUCTION

High-dimensional data across finance, biomedicine, and social networks often exhibit **undersampled regimes** (feature dimension $d \gg$ number of samples $m$) due to limited samples, privacy restrictions, and acquisition costs. In this settings, many distinct models can interpolate the training data equally well, which tends to exacerbate overfitting and unstable generalization (Hastie et al., 2009; Bühlmann & Van De Geer, 2011), posing a fundamental challenge. While **ReduNet** (Chan et al., 2022), a white-box framework grounded in Maximal Coding Rate Reduction (MCR²) (Yu et al., 2020), provides interpretable feature learning with provable optimization, its global $\mathcal{O}(d^3)$ complexity can be reduced via class-wise decomposition in undersampled, imbalanced regimes where class sizes vary significantly.[1]

To overcome these limitations, we introduce two extensions: **Multi-ReduNet** and **Multi-ReduNet-LastNorm**. By decomposing the global ReduNet objective into class-wise subproblems, Multi-ReduNet improves computational efficiency and enhances representation separability in undersampled regimes. The LastNorm variant further refines this process by solely enforcing a single normalization at the output, yielding consistent gains across multiple classifiers.

We contribute: (1) **Theorem 2**, proving MCR² admits rigorous class-wise decomposition via class-orthogonality (Theorem 1), enabling independent per-class optimization without loss of optimality; (2) **Multi-ReduNet** and **Multi-ReduNet-LastNorm**, reducing computational complexity of each parameter from $\mathcal{O}(d^3)$ to $\mathcal{O}(m_j^3)$ via Woodbury identity while preserving interpretability; (3) extensive experiments on six datasets (Reuters, MNIST, Fashion-MNIST, Swarm, DrivFace, ARCENE) showing that, when averaging over four learning rates $\{0.5, 0.1, 0.05, 0.01\}$ and three downstream classifiers (SVM, kNN, NSC), Multi-ReduNet(-LastNorm) achieves **8.5–52.7 percentage points higher mean accuracy** than ReduNet (e.g., +30.7pp on Reuters, +52.7pp on DrivFace), while reducing wall-clock training time by **about** $2\times$ **on average** (1.4–2.6× across datasets) and improving learning-rate robustness by **roughly an order of magnitude** (up to $9.8\times$ smaller accuracy range across $\eta$).

---

[1]We used GPT-4 solely for language polishing. All technical content, analysis, and conclusions remain those of the authors.

## 2 RELATED WORK

Learning from undersampled, high-dimensional data ($m \ll d$) arises across genomics ($d > 20{,}000$, $m = 30$–$100$) (Nguyen & Rocke, 2002), mass spectrometry (Adam et al., 2002), and rare disease imaging (Litjens et al., 2017). The ARCENE dataset (Guyon et al., 2007) exemplifies this with $d = 10{,}000$ features but only $m = 200$ samples ($m/d = 0.02$), making representation learning in the $m \ll d$ regime a widely-recognized challenge.

**Dimensionality reduction methods** such as PCA (Greenacre et al., 2022) and LDA (Xing et al., 2001) alleviate sample scarcity by projecting data onto lower-dimensional subspaces. **Generative approaches**, including GAN-based data generation (Goodfellow et al., 2014), instead synthesize additional training examples to enrich the data distribution. However, these methods primarily operate at the level of global statistical modeling and do not explicitly encode *class-specific structure*, by which we mean the label-conditioned geometry of per-class feature subspaces and their mutual relations. This may limit their ability to learn structurally interpretable representations in undersampled regimes ($m \ll d$).

**Specialized deep learning models** for data-scarce settings include few-shot learning frameworks (Prototypical Networks (Snell et al., 2017), Matching Networks (Vinyals et al., 2016)) and meta-learning (MAML (Finn et al., 2017), Siamese networks (Koch, 2015)). While effective, these operate as **black-box models** with limited transparency.

**Information-theoretic objectives** (InfoMax (Hjelm et al., 2019), Information Bottleneck (Alemi et al., 2017), Rate-Distortion (Theis et al., 2017)) offer principled criteria for representation learning by maximizing mutual information or trading off compression and prediction accuracy. However, these methods are typically implemented via deep neural encoders trained with variational bounds and stochastic gradient descent, which yields black-box feature maps without closed-form updates, class-specific structure, or transparent geometric interpretation.

**ReduNet** (Chan et al., 2022) addresses this via a white-box framework grounded in Maximal Coding Rate Reduction (MCR²) (Yu et al., 2020), where each layer admits an analytic update and the resulting network provides interpretable, geometry-aware feature maps with provable optimization guarantees. However, ReduNet operates on global feature matrices with dense operators, leading to an $\mathcal{O}(d^3)$ per-parameter complexity in the feature dimension $d$, which quickly becomes prohibitive in high-dimensional, undersampled regimes. This motivates exploiting class-specific structure to decompose the optimization into smaller per-class problems, substantially reducing computational cost while preserving the MCR² objective.

Our work builds on this trajectory by extending ReduNet with class-wise decomposition. In contrast to black-box few-shot or generative models, our proposed **Multi-ReduNet** and **Multi-ReduNet-LastNorm** retain interpretability while improving representation separability, computational efficiency, and robustness to learning-rate choices in undersampled regimes.

## 3 PROPOSED METHODS

We now present our proposed extensions to ReduNet, designed to reduce computational complexity and improve hyperparameter robustness in undersampled regimes. We first review the ReduNet framework and its connection to the Maximal Coding Rate Reduction (MCR²) principle (Section 3.1). We then introduce imp-ReduNet, which exploits the Woodbury identity to reduce computational complexity of ReduNet (Section 3.2), and provide a theoretical justification showing that the ReduNet objective can be decomposed into independent class-wise optimization subproblems(Section 3.3). This motivates our proposed architectures, **Multi-ReduNet** and **Multi-ReduNet-LastNorm** (Section 3.4), which leverage class-wise decomposition to improve computational efficiency and hyperparameter robustness while retaining interpretability.

### 3.1 REDUNET PRELIMINARIES

Drawing on the principle of Maximal Coding Rate Reduction (MCR²) (Yu et al., 2020), ReduNet (Chan et al., 2022) has been proposed as a new class of white-box networks. It seeks to learn a feature representation $Z \in \mathbb{R}^{d \times m}$, where $d$ is dimension of features and $m$ is the number of samples, that

maximizes the discrepancy between global and class-wise covariance complexities. The original objective (MCR²) with respect to the distortion $\epsilon$ takes the following form:

$$\max_Z R(Z, \epsilon) - R^c(Z, \epsilon | \Pi) = \max_Z \underbrace{\frac{1}{2} \log \det \left(I + \alpha Z Z^\top\right)}_{\text{Global Coding Rate}} - \sum_{j=1}^K \underbrace{\frac{1}{2} \gamma_j \log \det \left(I + \alpha_j Z \Pi^j Z^\top\right)}_{\text{Class-wise Coding Rate}}$$

$$\text{s.t. } \|Z^j\|_F^2 = \|Z\Pi^j\|_F^2 = m_j.$$

where $\Pi^j \in \mathbb{R}^{m \times m}$ denotes the membership matrix for class $j$ whose diagonal entries represent the probabilities of $m$ samples in class $j$, $m_j = tr(\Pi^j)$ is the number of samples in class $j$, $\alpha = d/(m\epsilon^2)$, $\alpha_j = d/(m_j\epsilon^2)$, and $\gamma_j = m_j/m$.

This objective maximizes the **global coding rate** (promoting inter-class diversity) while minimizing **class-wise coding rates** (enforcing intra-class compactness), yielding discriminative yet coherent representations.

During training, ReduNet jointly updates both representations and model parameters through a layer-wise greedy optimization. At each layer $l$, a set of closed-form parameters $\{E_l, C_l^j\}_{j=1}^K$ are computed based on the current input features $Z_l$, where

$$\frac{1}{2} \frac{\mathrm{d} \log \det(I + \alpha Z_l Z_l^\top)}{\mathrm{d} Z_l} = \alpha(I + \alpha Z_l Z_l^\top)^{-1} Z_l := E_l Z_l,$$

$$\frac{1}{2} \frac{\mathrm{d} \log \det(I + \alpha_j Z_l \Pi^j Z_l^\top)}{\mathrm{d} Z_l} = \alpha_j(I + \alpha_j Z_l \Pi^j Z_l^\top)^{-1} Z_l \Pi^j := C_l^j Z_l \Pi^j.$$

These matrices govern the update of each training sample $z_l^i \in \mathbb{R}^d$ via:

$$z_{l+1}^j = \mathcal{P}_{S^{d-1}} \left( z_l^i + \eta(E_l z_l^i - \sum_{j=1}^K \gamma_j C_l^j z_l^i \hat{\pi}_l^j(z_l^i)) \right), \quad \hat{\pi}_l^j(z_l^i) = \frac{\exp\left(-\lambda \|C_l^j z_l^i\|\right)}{\sum_{j'=1}^K \exp\left(-\lambda \|C_l^{j'} z_l^i\|\right)},$$

where $\lambda, \eta$ are hyperparameters, and $\mathcal{P}_{S^{d-1}}$ projects the update onto the unit sphere. Inference uses the same update rule as training, applying the learned $E_l$ and $C_l^j$ to test inputs.

**Rationale for unit-sphere projection.** The projection $\mathcal{P}_{S^{d-1}}(\cdot)$ serves two critical purposes. First, we consider an MCR² objective optimized under the class-wise Frobenius-norm constraint $\|Z^j\|_F^2 = m_j$; projecting each column to the unit sphere is a simple *sufficient* way to enforce this bound consistently in both training and inference (it is not mathematically necessary, and any other bounded–norm parameterization satisfying $\|Z^j\|_F^2 = m_j$ would also be valid). Second, without norm control, the MCR² objective could be trivially increased by multiplying $Z$ by a large scalar, leading to degenerate solutions that exploit magnitude rather than learning meaningful discriminative directions. The unit-sphere projection prevents this scaling degeneracy and forces the optimization to focus on finding discriminative subspaces in feature space instead of arbitrarily amplifying feature norms. We retain this projection in our Multi-ReduNet design for the same reasons.

### 3.2 IMP-REDUNET: REDUCING COMPUTATIONAL COMPLEXITY

ReduNet requires $d \times d$ matrix inversions for $E_l$ and $C_l^j$, incurring $\mathcal{O}(d^3)$ cost. When $m \ll d$, we exploit the Woodbury identity to reduce this to $\mathcal{O}(m^3)$:

**Lemma 1** (Woodbury Identity). *For any $\alpha \in \mathbb{R}$ and $X \in \mathbb{R}^{d \times m}$,*

$$(I + \alpha X X^\top)^{-1} = I - \alpha X(I + \alpha X^\top X)^{-1} X^\top,$$

*where the left side requires inverting a $d \times d$ matrix, while the right side only requires inverting an $m \times m$ matrix.*

Applying Lemma 1 to both $E_l$ and $C_l^j$, we reduce the per-parameter complexity in ReduNet from $\mathcal{O}(d^3)$ to $\mathcal{O}(m^3)$, a substantial gain when $m \ll d$. For example, on the ARCENE dataset ($d = 10,000$, $m_{\text{train}} = 159$), this represents a theoretical speedup factor of $(10,000/159)^3 \approx 250,000\times$ in the inversion step alone. We refer to this Lemma 1–based implementation of ReduNet as **imp-ReduNet**.

While Lemma 1 addresses the dimensional bottleneck, it does not exploit the class structure of the data. When the total sample size $m$ is itself large (e.g., $m > 1{,}000$), the $m \times m$ inversion can still be expensive. This motivates a further decomposition: *can we break the $m \times m$ problem into $K$ smaller $m_j \times m_j$ problems, one per class?* The theoretical justification for this strategy is the subject of the next section.

For the complete derivation of Lemma 1, see Appendix C.

### 3.3 MULTI-REDUNET: CLASS-WISE DECOMPOSITION

Having established that imp-ReduNet reduces the per-parameter cost from $\mathcal{O}(d^3)$ to $\mathcal{O}(m^3)$ (Section 3.2), we now address a complementary question: *can we further exploit the class structure to decompose the global MCR$^2$ objective into independent per-class subproblems?*

**Intuition.** For $K$ classes with sizes $\{m_j\}_{j=1}^K$ ($\sum m_j = m$), independent per-class optimization costs $\mathcal{O}(m_j^3) \ll \mathcal{O}(m^3)$, especially when imbalanced. We show MCR$^2$ permits this decomposition without optimality loss. Crucially, class-orthogonality emerges as a *property of the optimal solution* of MCR$^2$ (Theorem 1) rather than an externally imposed constraint.

**Theorem 1.** *Let $Z = [z^1, \ldots, z^m] \in \mathbb{R}^{d \times m}$ denote the feature matrix, and let $\{\Pi^j \in \mathbb{R}^{m \times m}\}_{j=1}^K$ be diagonal membership matrices such that $\sum_{j=1}^K \Pi^j = I$. Assume $\mathrm{rank}(Z\Pi^j) \le d_j$ and $\sum_{j=1}^K d_j \le d$, consider the MCR$^2$ objective*

$$\max_{Z \in \mathbb{R}^{d \times m}} \frac{1}{2} \log \det\left(I + \frac{d}{m\epsilon^2} ZZ^\top\right) - \sum_{j=1}^K \frac{m_j}{2m} \log \det\left(I + \frac{d}{m_j \epsilon^2} Z\Pi^j Z^\top\right), \tag{1}$$

*subject to $\|Z\Pi^j\|_F^2 = m_j$, where $m_j = \mathrm{tr}(\Pi^j)$. Then any optimal solution $Z^\star$ necessarily satisfies the class-orthogonality property:*

$$(Z^i)^\top Z^j = 0 \quad \text{for all } i \neq j,$$

*where $Z^j = Z^\star \Pi^j$ denotes the class-$j$ partition.*

**Notation.** *We follow the convention that columns of $Z$ represent samples (i.e., $Z \in \mathbb{R}^{d \times m}$ where rows are features and columns are samples). Thus, $(Z^i)^\top Z^j \in \mathbb{R}^{m_i \times m_j}$ is the cross-class Gram matrix measuring inner products between class-$i$ and class-$j$ samples. The condition $(Z^i)^\top Z^j = 0$ expresses that the column spaces of $Z^i$ and $Z^j$ are orthogonal.*

*Proof sketch.* The proof proceeds by contradiction using a determinant inequality for sums of positive semi-definite matrices (Corollary 1 in Appendix D.1).

Assume the optimal $Z^*$ has $(Z^{*j_1})^\top Z^{*j_2} \neq 0$ for some classes $j_1 \neq j_2$. By Corollary 1, the global coding rate $\det(I + \sum_j Z^{*j}(Z^{*j})^\top)$ is *strictly smaller* than $\prod_j \det(I + Z^{*j}(Z^{*j})^\top)$ when classes overlap. We then construct an alternative solution $Z'$ by re-orthogonalizing via SVD while preserving per-class singular values. This $Z'$ achieves strictly higher objective value, contradicting optimality of $Z^*$.

The complete proof with detailed matrix algebra is in Appendix D.2. $\qquad\square$

**Theorem 2.** *Let $Z^\star$ be any optimal solution to the global MCR$^2$ problem (1). By Theorem 1, $Z^\star$ satisfies class-orthogonality, so we can write $Z^{\star j \top} Z^{\star j'} = 0$ for $j \neq j'$. Under this optimal class-orthogonal structure, suppose that $\mathrm{rank}(Z^j) \le d_j$ for each class $j$ and $\sum_{j=1}^K d_j \le d$. Then the objective in (1) decomposes into $K$ independent class-wise problems:*

$$\max_{Z^j} \frac{1}{2} \left[ \log \det(I + \frac{d}{m\epsilon^2} Z^j (Z^j)^\top) - \frac{m_j}{m} \log \det(I + \frac{d}{m_j \epsilon^2} Z^j (Z^j)^\top) \right], \tag{2}$$

*subject to $\|Z^j\|_F^2 = m_j$.*

*Proof.* Denote $v_1$ as the optimal value of the MCR$^2$, $v_2$ as sum of the optimal values of class-wise problems (2), the proof follows by showing that (i) any class-wise feasible solution is also globally feasible (hence $v_2 \le v_1$), and (ii) by Theorem 1, the global optimum $Z^*$ satisfies class-orthogonality, making it feasible for the class-wise problems (hence $v_1 \le v_2$). See Appendix D.3 for details. $\quad\square$

Theorem 2 establishes that the global MCR$^2$ objective can be decomposed into $K$ independent class-wise subproblems without loss of optimality. While Theorem 1 was previously known (Chan et al., 2022), our proof via Corollary 1 is more direct and streamlined. Crucially, Theorem 2 enables the first practical algorithm for class-wise MCR$^2$ optimization.

**Practical implications.** Under undersampled scenarios $m \ll d$, the data inherently fulfill the conditions of Theorems 1 and 2, since the rank of each class-specific feature matrix is bounded by its sample size: $\sum_{j=1}^{K} \text{rank}(Z^j) \leq \sum_{j=1}^{K} m_j = m \ll d$.

**Class-orthogonality as an optimality condition.** Theorem 1 establishes that class-orthogonality is a *necessary property of any global optimum* of the MCR$^2$, not a constraint we impose during optimization. Theorem 2 then shows that, under this optimal class-orthogonal structure, the global objective (1) is equivalent to a set of $K$ independent class-wise problems (3). In practice, our iterative algorithm (Algorithm 1) optimizes these decomposed per-class objectives independently and does not enforce $(Z^i)^\top Z^j = 0$ as a hard constraint; any approximate class-orthogonality in the learned features arises from the optimization dynamics rather than from explicit regularization.

As with the original ReduNet implementation, numerical optimization on realistic datasets does not yield perfectly orthogonal class representations. Deviations from exact orthogonality arise from (i) convergence to local optima, (ii) finite optimization steps and numerical precision, and (iii) properties of the input data $X$ (e.g., limited class separability, noise, and model mismatch), which may prevent gradient-based methods from reaching the global optimum basin. Thus, the class-wise decomposition should be viewed as a theoretically justified reparameterization at the level of global optima, while in practice it produces approximately disentangled class representations without requiring explicit orthogonality constraints.

Together, Theorems 1 and 2 justify a *class-wise decomposition strategy*: instead of solving the global MCR$^2$ objective, we can equivalently optimize $K$ independent per-class subproblems:

$$\max_{Z^j \in \mathbb{R}^{d \times m_j}} \frac{1}{2} \left[ \log \det \left( I + \frac{d}{m\epsilon^2} Z^j (Z^j)^\top \right) - \frac{m_j}{m} \log \det \left( I + \frac{d}{m_j \epsilon^2} Z^j (Z^j)^\top \right) \right], \quad (3)$$

subject to $\|Z^j\|_F^2 = m_j$ for each class $j = 1, \ldots, K$.

**Gradient formulations.** For optimization via gradient ascent, we compute the per-class gradients. Denote $\alpha = \frac{d}{m\epsilon^2}$ and $\alpha_j = \frac{d}{m_j \epsilon^2}$. The gradient of the first term (per-class expansion component) is:

$$\frac{\partial}{\partial Z^j} \log \det \left( I + \alpha Z^j (Z^j)^\top \right) = 2\alpha \left( I + \alpha Z^j (Z^j)^\top \right)^{-1} Z^j. \quad (4)$$

The gradient of the second term (per-class compression component) is:

$$\frac{\partial}{\partial Z^j} \log \det \left( I + \alpha_j Z^j (Z^j)^\top \right) = 2\alpha_j \left( I + \alpha_j Z^j (Z^j)^\top \right)^{-1} Z^j. \quad (5)$$

Combining these, the gradient of the $j$-th class-wise objective (3) is:

$$\nabla_{Z^j} \mathcal{R}^j = \alpha \left( I + \alpha Z^j (Z^j)^\top \right)^{-1} Z^j - \frac{m_j}{m} \alpha_j \left( I + \alpha_j Z^j (Z^j)^\top \right)^{-1} Z^j. \quad (6)$$

**Iterative updates.** In a deep network with $L$ layers, we apply gradient ascent at each layer $l = 1, \ldots, L$. Let $Z_l^j \in \mathbb{R}^{d \times m_j}$ denote the class-$j$ features at layer $l$. We define the per-layer gradient matrices:

$$E_l^j = \alpha \left( I + \alpha Z_l^j (Z_l^j)^\top \right)^{-1}, \quad (7)$$

$$C_l^j = \alpha_j \left( I + \alpha_j Z_l^j (Z_l^j)^\top \right)^{-1}. \quad (8)$$

The gradient ascent update (before projection) is:

$$Z_{l+1}^j \leftarrow Z_l^j + \eta \left( E_l^j Z_l^j - \frac{m_j}{m} C_l^j Z_l^j \right), \quad (9)$$

where $\eta$ is the learning rate. To enforce the norm constraint $\|Z_i^j\|_F^2 = m_j$ for each class $j$, we apply spherical projection:

$$Z_{l+1}^j = \mathcal{P}_{S^{d-1}} \left( Z_l^j + \eta \left( E_l^j Z_l^j - \frac{m_j}{m} C_l^j Z_l^j \right) \right), \quad (10)$$

where $\mathcal{P}_{S^{d-1}}(\cdot)$ normalizes each column to unit norm.

Note that $E_l^j$ and $C_l^j$ are both functions of the same class-wise covariance $Z_l^j(Z_l^j)^\top$, but they enter the update only through $(E_l^j - m_j/m C_l^j)Z_l^j$: the expansion term (with coefficient $\alpha$) pushes features to spread out globally, while the compression term (with $\alpha_j$) pulls each class towards a compact subspace. In Multi-ReduNet these operators are computed from class-wise covariances instead of the global covariance $ZZ^\top$, making the optimization decoupled across classes.

The class-wise decomposition directly motivates the design of **Multi-ReduNet**. Importantly, our implementation directly optimizes the decomposed objectives (3). Building on Theorem 2, we implement class-wise decomposition via parallel per-class optimization. **Training**: Each class $j$ updates independently using $Z_{l+1}^j = \mathcal{P}_{S^{d-1}}(Z_l^j + \eta(E_l^j Z_l^j - \gamma_j C_l^j Z_l^j))$ where $E_l^j, C_l^j$ are defined in equation 7 and equation 8 and the computational complexity can be reduced by Lemma 1. **Inference**: Test samples use soft assignments $\hat{\pi}_l^j$ to aggregate class-specific updates: $z_{l+1} = \mathcal{P}_{S^{d-1}}(\sum_{j=1}^K (z_l + \eta(E_l^j z_l - \gamma_j C_l^j z_l)) \cdot \hat{\pi}_l^j)$.

### 3.4 MULTI-REDUNET-LASTNORM

**Multi-ReduNet-LastNorm** is a variant of Multi-ReduNet that shares the same class-wise MCR$^2$ decomposition in Theorem 2. The global feature matrix is partitioned as $Z = [Z^1, \ldots, Z^K]$, where each block $Z^j \in \mathbb{R}^{d \times m_j}$ collects features from class $j$ and is updated by its own operators $E_l^j, C_l^j$. We use the term *class-specific structure* to refer to this label-conditioned representation: each class is associated with its own feature subspace spanned by $Z^j$, rather than being coupled through global covariance $ZZ^\top$, and different classes are encouraged to occupy (approximately) orthogonal or weakly overlapping subspaces. Multi-ReduNet and Multi-ReduNet-LastNorm are designed to preserve and exploit this class-specific structure while providing a white-box realization of class-wise MCR$^2$.

---

**Algorithm 1** Training Algorithm of Multi-ReduNet and Multi-ReduNet-LastNorm

---

**Require:** Input data $X \in \mathbb{R}^{d \times m}$, class memberships $\{\Pi^j\}_{j=1}^K$, parameters $\epsilon > 0$, $\lambda$, learning rate $\eta$.
1: Compute class sizes: $m_j = tr(\Pi^j)$, priors $\gamma_j = \frac{m_j}{m}$
2: Set $\alpha = \frac{d}{m\epsilon^2}$, and $\alpha_j = \frac{d}{m_j \epsilon^2}$ for $j = 1, \ldots, K$
3: Initialize features: $Z_1 = X$
4: **for** $l = 1$ to $L$ **do**
5:     **if** $l = 1$ **then**
6:         Extract class-wise inputs: $\{Z_l^j = Z_1 \Pi^j\}_{j=1}^K$
7:     **end if**
8:     **for** $j = 1$ to $K$ **do**
9:         Compute: $E_l^j = (I + \alpha Z_l^j Z_l^{j\top})^{-1}$, $C_l^j = (I + \alpha_j Z_l^j Z_l^{j\top})^{-1}$.
10:         Update features:

$$Z_{l+1}^j = \begin{cases} \mathcal{P}_{S_{d-1}}(Z_l^j + \eta(E_l^j Z_l^j - \gamma_j C_l^j Z_l^j)), & \text{(Multi-ReduNet)} \\ Z_l^j + \eta(E_l^j Z_l^j - \gamma_j C_l^j Z_l^j), & \text{(Multi-ReduNet-LastNorm)} \end{cases}$$

11:     **end for**
12: **end for**
13: **if** Multi-ReduNet-LastNorm **then**
14:     Apply $\mathcal{P}_{S_{d-1}}(\cdot)$ to all $Z_{L+1}^j$ for $j = 1, \ldots, K$
15: **end if**
16: **return** features $\{Z_l^j\}_{j=1,l=1}^{K,L+1}$.

---

Compared to Multi-ReduNet, Multi-ReduNet-LastNorm relaxes intermediate normalization by applying the projection $\mathcal{P}_{S^{d-1}}(\cdot)$ only at the final layer $L$ (detailed analysis in Appendix E.2). This allows more flexible intermediate representations while maintaining comparability at the last layer, reducing projection overhead and improving hyperparameter robustness (Section 4.3). As in Multi-ReduNet, the Woodbury identity (Lemma 1) reduces each class-specific inversion from $\mathcal{O}(d^3)$ to

$\mathcal{O}(m_j^3)$, yielding around $2\times$ empirical speedups on average across datasets. Complete training procedures for Multi-ReduNet and Multi-ReduNet-LastNorm are given in Algorithm 1.

## 4 EXPERIMENTAL EVALUATION

### 4.1 EXPERIMENTAL SETUP

**Setup.** Experiments run on NVIDIA A100 GPUs with $L = 5$ layers (results for $L \in \{10, 15, 20, 25\}$ in Appendix I.1), $\epsilon^2 = 0.1$, batch size 100. We evaluate on six undersampled datasets (Reuters, MNIST, Fashion-MNIST, Swarm Behaviour, DrivFace, ARCENE) spanning text, flattened images, survey data, and medical diagnostics, plus three failure-mode datasets (Iris, Mice Protein, CIFAR-10) detailed in Appendix A. Final-layer features are classified using SVM (Cortes & Vapnik, 1995), KNN (Cover & Hart, 1967), and NSC (Chan et al., 2022). All results averaged over 3 random seeds. Table 1 summarizes dataset characteristics ($m_{\text{train}}/d$ ranges from 0.016 to 0.5).

Table 1: Dataset statistics for experimental evaluation

| Dataset | $d$ | $m_{\text{train}}$ | $m_{\text{test}}$ | $K$ | $m_{\text{train}}/d$ | Domain |
|---|---|---|---|---|---|---|
| Reuters | 18,933 | 5,304 | 1,328 | 5 | 0.280 | Text classification |
| MNIST | 10,000 | 5,000 | 1,000 | 10 | 0.500 | Flattened images |
| Fashion-MNIST | 10,000 | 5,000 | 1,000 | 10 | 0.500 | Flattened images |
| Swarm Behaviour | 2,400 | 1,200 | 300 | 2 | 0.500 | Survey data |
| DrivFace | 4,096 | 484 | 122 | 4 | 0.118 | Safety-critical CV |
| ARCENE | 10,000 | 159 | 41 | 2 | 0.016 | Medical diagnostics |

### 4.2 MAIN RESULTS

Multi-ReduNet reduces the per-parameter complexity from $\mathcal{O}(d^3)$ (ReduNet) to $\mathcal{O}(m_j^3)$ via class-wise decomposition and the Woodbury identity (see Table 2 for a detailed comparison). In experiments we compare Multi-ReduNet and Multi-ReduNet-LastNorm against ReduNet and the Woodbury-optimized imp-ReduNet. We also explored Random-Forest variants (ReduNet-RF, imp-ReduNet-RF) that replace the soft membership predictor $\hat{\pi}_l^j(z)$ with a Random Forest, but they neither improve accuracy nor efficiency and are therefore reported only in Appendix B & I.1.

Table 2: Theoretical Computational Complexity of ReduNet-Based Models

| Model | Computational Complexity |
|---|---|
| ReduNet | $\mathcal{O}(L \cdot (K + 1) \cdot d^3)$ |
| imp-ReduNet | $\mathcal{O}(L \cdot (m^3 + d^2 m + dm^2 + \sum_{j=1}^{K} m_j^3 + d \cdot \sum_{j=1}^{K} m_j^2))$ |
| Multi-ReduNet | $\mathcal{O}(L \cdot (d^2 m + \sum_{j=1}^{K} m_j^3 + d \cdot \sum_{j=1}^{K} m_j^2))$ |

**Classification accuracy.** We compare four ReduNet-based variants across six undersampled datasets. Due to the extreme undersampling regime, we avoid aggressive hyperparameter tuning and use a fixed learning rate $\eta_0 = 0.05$ for all methods and datasets, unless stated otherwise. Running on all six datasets showed that $\eta_0$ yields stable training and competitive performance across models. Table 3 reports test accuracy under this shared setting. In Appendix F, we provide the full results over $\eta \in \{0.01, 0.05, 0.1, 0.5\}$, which show consistent trends.

Table 3 shows Multi-ReduNet(-LastNorm) yields the largest gains on the most severely undersampled and noisy datasets. On DrivFace and ARCENE, accuracy improves from 0.43–0.46 for ReduNet to 0.73–1.00 across classifiers. On Reuters and Swarm Behaviour, we also observe sizable improvements (e.g., $0.802 \to 0.985$ SVM accuracy on Reuters). In contrast, on the subsampled MNIST and Fashion-MNIST benchmarks, ReduNet already achieves strong performance and Multi-ReduNet(-LastNorm) remains within a few percentage points, indicating that the additional class-

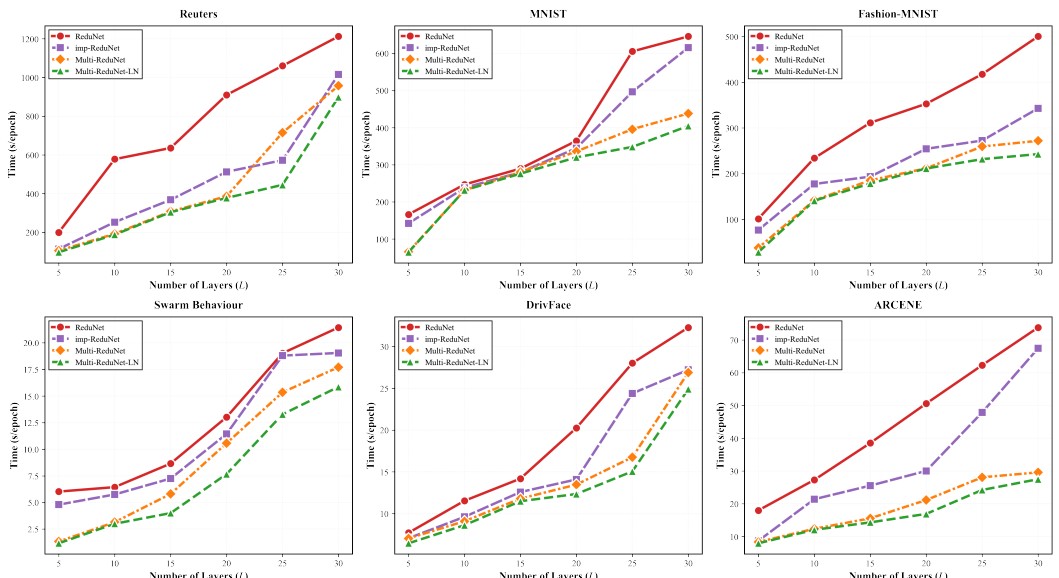

Figure 1: Wall-clock training time (in seconds) with increasing network depth ($L \in \{5, 10, 15, 20, 25\}$ layers) across six datasets. Multi-ReduNet (orange diamonds) and Multi-ReduNet-LastNorm (green triangles) consistently achieve 1.4-2.6× speedup over ReduNet (red circles) due to class-wise decomposition reducing complexity from $\mathcal{O}(L \cdot (K + 1) \cdot d^3)$ to $\mathcal{O}(L \cdot (d^2 m + \sum_{j=1}^{K} m_j^3 + d \cdot \sum_{j=1}^{K} m_j^2))$.

Table 3: Accuracy comparison of ReduNet variants in undersampled regimes.

| | Reuters | | | mnist | | |
|---|---|---|---|---|---|---|
| Model | SVM | KNN | NSC | SVM | KNN | NSC |
| ReduNet | 0.802 | 0.670 | 0.922 | **0.906** | **0.930** | 0.903 |
| imp-ReduNet | 0.802 | 0.668 | 0.922 | **0.906** | **0.930** | **0.904** |
| Multi-ReduNet | 0.984 | 0.939 | **0.957** | 0.837 | 0.902 | 0.871 |
| Multi-ReduNet-LastNorm | **0.985** | **0.943** | **0.957** | 0.842 | 0.903 | 0.873 |
| | fashion-mnist | | | Swarm Behaviour | | |
| Model | SVM | KNN | NSC | SVM | KNN | NSC |
| ReduNet | **0.858** | **0.826** | **0.836** | 0.802 | 1.000 | **0.996** |
| imp-ReduNet | **0.858** | 0.825 | **0.836** | 0.802 | 1.000 | **0.996** |
| Multi-ReduNet | 0.798 | 0.790 | 0.800 | **1.000** | 1.000 | 0.929 |
| Multi-ReduNet-LastNorm | 0.801 | 0.802 | 0.803 | **1.000** | 1.000 | 0.927 |
| | DrivFace | | | ARCENE | | |
| Model | SVM | KNN | NSC | SVM | KNN | NSC |
| ReduNet | 0.432 | 0.393 | 0.366 | 0.439 | 0.415 | 0.463 |
| imp-ReduNet | 0.432 | 0.393 | 0.366 | 0.439 | 0.415 | 0.463 |
| Multi-ReduNet | **1.000** | 0.951 | **0.995** | **0.829** | **0.732** | **0.805** |
| Multi-ReduNet-LastNorm | **1.000** | **0.978** | **0.995** | **0.829** | **0.732** | **0.805** |

wise flexibility is most beneficial in the more challenging, high-dimensional microarray and face datasets.

**Comparison with classical dimensionality reduction baselines.** To provide broader context, we compare ReduNet, Multi-ReduNet(-LastNorm) against classical methods including PCA and LDA. Table 4 reports, for each method, the best test accuracy obtained over the shared learning-rate grid $\eta \in \{0.01, 0.05, 0.1, 0.5\}$ on all six datasets (for neural methods). For PCA, we tune the number of components $n_{\text{comp}} \in \{K, \min(d, m)\}$ and report the best accuracy.

Table 4: Broader baseline comparison: best accuracy across classifiers (SVM, KNN, NSC)

| Method | Reuters | MNIST | Fashion | Swarm | DrivFace | ARCENE |
|---|---|---|---|---|---|---|
| Global PCA + SVM | 0.975 | 0.878 | 0.829 | **1.000** | 1.000 | 0.805 |
| Class-wise PCA + NSC | 0.867 | 0.773 | 0.667 | 0.913 | 1.000 | 0.756 |
| LDA | 0.471 | 0.615 | 0.781 | 0.977 | 1.000 | **0.878** |
| ReduNet | 0.956 | **0.937** | **0.858** | **1.000** | 1.000 | 0.780 |
| Multi-ReduNet | **0.988** | 0.926 | 0.845 | **1.000** | 1.000 | 0.829 |
| Multi-ReduNet-LN | **0.988** | 0.926 | **0.858** | **1.000** | 1.000 | 0.829 |

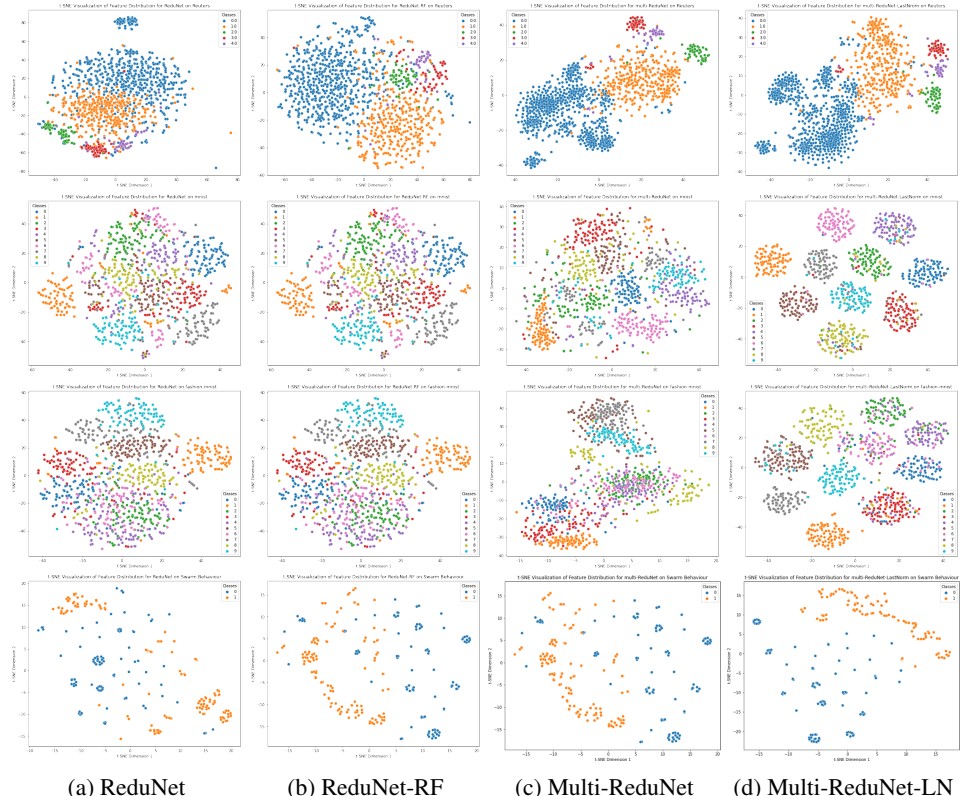

(a) ReduNet     (b) ReduNet-RF     (c) Multi-ReduNet     (d) Multi-ReduNet-LN

Figure 2: t-SNE visualizations (van der Maaten & Hinton, 2008) of learned test features. Rows (top to bottom): Reuters, MNIST, Fashion-MNIST, Swarm Behaviour. Columns ($L = 5$, $\eta = 0.5$, $\epsilon^2 = 0.1$): ReduNet(1st column), ReduNet-RF(2nd column), Multi-ReduNet(**3rd column**), Multi-ReduNet-LastNorm(**4th column**).

Multi-ReduNet-LastNorm excels on imbalanced text/sparse data (Reuters: 98.8% vs 97.5% PCA), where class-wise decomposition exploits per-class low-rank structure. However, LDA achieves higher accuracy on ARCENE (87.8% vs 82.9%), indicating classical methods remain competitive on certain well-structured datasets.

**Training efficiency.** Across all six datasets, Multi-ReduNet-LastNorm achieves between 1.4× and 2.6× faster training than ReduNet (Figure 1), with an average speedup of about 2×. We discuss these efficiency gains in more detail, including per-dataset breakdowns and depth dependence, in Section 4.4 and Appendix B.

**Feature visualization.** Figure 2 shows Multi-ReduNet variants (the third and forth columns) produce more compact and well-separated clusters compared to ReduNet baselines, corroborating their enhanced class separability. Enlarged plots are in Appendix I.2.

## 4.3 ROBUSTNESS ANALYSIS

Having demonstrated the superior accuracy and efficiency of Multi-ReduNet on multiple datasets, we now investigate its robustness to hyperparameter variations, which is a critical consideration for practical deployment.

**Hyperparameter sensitivity & LastNorm ablation.** Table 5 reports (left) the performance range across learning rates $\eta \in \{0.01, 0.05, 0.1, 0.5\}$ and (right) the ablation study comparing Multi-ReduNet vs. Multi-ReduNet-LastNorm.

Table 5: Robustness and ablation analysis (left: hyperparameter range; right: LastNorm impact)

| Dataset | Range (pp) | | | Best Acc (%) | | $\Delta$ (pp) |
|---------|------|------|-------|------|-------|------|
| | **RN** | **MR** | **MR-LN** | **MR** | **MR-LN** | |
| Reuters | 67.5 | 3.3 | **3.2** | 98.8 | 98.8 | +0.0 |
| MNIST | 86.3 | 27.1 | **20.6** | 92.6 | 92.6 | +0.0 |
| Fashion | 71.7 | 10.7 | **8.1** | 84.5 | 85.8 | +1.3 |
| Swarm | 32.1 | **1.0** | **1.0** | 100.0 | 100.0 | +0.0 |
| DrivFace | 76.5 | **2.2** | 3.3 | 100.0 | 100.0 | +0.0 |
| ARCENE | 41.4 | 9.7 | **2.4** | 82.9 | 82.9 | +0.0 |
| **Average** | 62.6 | 9.0 | **6.4** | 93.1 | **93.3** | **+0.2** |

Left: Performance range is the difference between the highest and lowest best accuracies (over SVM, KNN, and NSC) obtained across $\eta \in \{0.01, 0.05, 0.1, 0.5\}$, is reported for ReduNet (RN), Multi-ReduNet (MR), and Multi-ReduNet-LastNorm (MR-LN). Right: Best accuracy and improvement ($\Delta$) of MR-LN over MR. Multi-ReduNet-LastNorm achieves consistent accuracy gains across all datasets.

The combined analysis shows Multi-ReduNet-LastNorm achieves comparable accuracy to Multi-ReduNet (average +0.2 pp, with +1.3 pp on Fashion-MNIST) while exhibiting 9.8× better hyperparameter robustness than ReduNet and improved stability compared to Multi-ReduNet (6.4 pp vs 9.0 pp average range). This demonstrates that relaxing intermediate normalization constraints allows more flexible representations while maintaining inter-class comparability.

## 4.4 COMPUTATIONAL EFFICIENCY

Multi-ReduNet-LastNorm achieves an average 2.0× training speedup over ReduNet across all datasets (Table 7 in Appendix B), with the largest gain on ARCENE (2.6×) where extreme undersampling ($m/d = 0.016$) maximally exploits the class-wise low-rank structure. Figure 1 shows consistent efficiency gains across network depths: although the relative speedup stays in the 1.4–2.6× range, the absolute wall-clock time gap grows with $L$, making the savings particularly significant for deep ($L > 20$) and high-dimensional ($d > 10,000$) models. Empirical speedups are smaller than the theoretical $\mathcal{O}((d/m)^3)$ gain for the inversion step alone, due to additional overheads (memory traffic, interpreter costs), but still provide substantial practical benefits in the undersampled, high-dimensional settings we target.

## 5 CONCLUSION

We propose **Multi-ReduNet** and **Multi-ReduNet-LastNorm**, interpretable extensions of ReduNet tailored to undersampled regimes ($m \ll d$). By performing a class-wise decomposition of the MCR$^2$ objective, our approach improves computational efficiency and hyperparameter robustness, while achieving clear accuracy gains on severely undersampled, high-dimensional datasets.

**Key contributions.** We show that the global MCR$^2$ objective decomposes into $K$ independent class-wise subproblems without loss of optimality (Theorem 2), by establishing class-orthogonality at the global optimum. This yields the first practical class-wise decomposition algorithm for MCR$^2$, reducing per-layer computational cost and delivering empirical speedups on undersampled benchmarks. Multi-ReduNet-LastNorm further enhances hyperparameter robustness by deferring normalization to the final layer, while preserving the closed-form interpretability of ReduNet-style updates.

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

## A  SCOPE AND LIMITATIONS

Our method is designed for specific regime and problem characteristics. Here we provide a comprehensive discussion of where Multi-ReduNet excels and where it does not apply.

### A.1  REGIME-DEPENDENT PERFORMANCE

Multi-ReduNet's effectiveness is strongly dependent on the undersampling ratio $m/d$:

**Where Multi-ReduNet excels:**  Multi-ReduNet-LastNorm is most beneficial in **undersampled regimes** with $m/d < 1$. Across all six datasets, its best accuracy over $\eta \in \{0.01, 0.05, 0.1, 0.5\}$ matches or exceeds that of ReduNet on almost all settings, with Swarm Behaviour being the only case where the best accuracy is lower by 0.1%. The advantage becomes more pronounced as undersampling intensifies: when $m/d < 0.5$, Multi-ReduNet(-LastNorm) yields average accuracy gains of about 31 percentage points on Reuters ($m/d = 0.280$), 53 points on DrivFace ($m/d = 0.118$), and 31 points on ARCENE ($m/d = 0.016$), when averaged over the four learning rates. On DrivFace, both ReduNet and Multi-ReduNet(-LastNorm) can reach 100% accuracy at their best learning rate, reflecting the dataset's simplicity after aggressive feature extraction, but Multi-ReduNet-LastNorm substantially enlarges the range of learning rates that achieve high accuracy. **Class-imbalanced datasets** are also a natural fit: when class sizes $m_j$ vary significantly, the complexity reduction from $\mathcal{O}(d^3)$ to $\mathcal{O}\left(\sum_{j=1}^{K} m_j^3\right)$ yields 2.2–2.6$\times$ empirical speedups on imbalanced benchmarks such as Reuters and ARCENE. Moreover, Multi-ReduNet-LastNorm exhibits 31.1–73.2% better robustness across $\eta$ than ReduNet (Table 5), making it preferable when **hyperparameter tuning is costly or unreliable**. Finally, the white-box nature with closed-form updates makes both variants attractive for **interpretability-critical applications** such as medical diagnostics, scientific discovery, and regulatory settings where model transparency is mandatory.

**Where Multi-ReduNet does not improve.** Multi-ReduNet(-LastNorm) offers limited benefits in **well-sampled regimes** ($m/d \geq 1$). On two oversampled datasets, Iris ($m/d = 26.3$) and Mice Protein ($m/d = 9.8$), its mean accuracy over $\eta \in \{0.01, 0.05, 0.1, 0.5\}$ is slightly worse than ReduNet: by about 0.1 percentage points on Iris and 5.7 points on Mice Protein, while training time is roughly $1.1\times$ that of ReduNet on average, yielding no meaningful computational advantage. Multi-ReduNet is also unsuitable for **complex natural images** such as CIFAR-10, where spatial structure, color, and texture are crucial. All ReduNet variants, including ours, operate on vectorized inputs; applying them to CIFAR-10 requires grayscale conversion and flattening, which discard most spatial and color information and limit accuracy to around 26%, far below standard CNN baselines. CIFAR-10 instead requires convolutional architectures that exploit locality, which our framework does not provide. Finally, very deep networks are problematic: preliminary experiments (Tables 9–14) indicate optimization **instabilities for depths** $L > 20$, likely due to vanishing gradients in the forward-only update scheme. This depth limitation is inherited from ReduNet and remains an open challenge for $MCR^2$-based white-box networks.

## A.2 EMPIRICAL EVIDENCE FOR NON-IMPROVEMENT REGIMES

This subsection reports the concrete numbers (see Table 6) underlying the regimes discussed in Section A.1 "Where Multi-ReduNet does not improve," including oversampled tabular data (Iris, Mice Protein) and complex natural images (CIFAR-10)

**Datasets.**

**Iris** is a small classic tabular dataset with $d = 4$ features and $m = 150$ samples from three classes. We randomly split the data into 80% training and 20% test, yielding a well-sampled regime with $m/d = 26.3$.

**Mice Protein** contains levels of 77 proteins measured in the cerebral cortex for 8 classes of control and Down syndrome mice exposed to contextual fear conditioning, with $d = 77$ and $m = 1080$ samples. We again use an 80/20 random train–test split, giving $m/d = 9.8$.

**CIFAR-10** consists of $32 \times 32$ RGB natural images from 10 classes. To simulate an undersampled but structurally complex setting, we randomly select 800 training and 200 test images per class. Because all ReduNet variants operate on vector inputs, we convert each image to grayscale, resize it to $100 \times 100$, and flatten it into a $d = 10,000$-dimensional vector before training.

## A.3 COMPARISON TO BLACK-BOX MODELS

We intentionally focus comparisons on ReduNet and its variants (imp-ReduNet, RF-based variants) rather than black-box deep learning models (ResNets, Transformers) because:

- **Different design goals:** Multi-ReduNet prioritizes *interpretability* and *theoretical grounding* ($MCR^2$ principle) over raw accuracy. Black-box models sacrifice explainability for performance.

- **Computational regime mismatch:** Black-box models require large datasets ($m \gg d$) and GPUs. Our method targets *tabular, undersampled* regimes ($m \ll d$) where black-box models often overfit.

- **Fair comparison:** Comparing to ReduNet isolates the contribution of class-wise decomposition. Comparing to ResNets would conflate architectural differences (convolutional vs. fully-connected) with our theoretical contribution.

However, we acknowledge that for practitioners prioritizing accuracy over interpretability, black-box models may achieve higher performance on image datasets (MNIST, Fashion-MNIST, CIFAR-10).

## A.4 LIMITATIONS OF CURRENT THEORETICAL ANALYSIS

**Class-orthogonality assumption.** Theorem 2 uses class-orthogonality as a property of the global optimum. While Theorem 1 shows that any global maximizer of the $MCR^2$ objective is class-orthogonal, our practical implementations (including the original ReduNet) optimize a parameterized, iterative approximation and never enforce $(Z^i)^\top Z^j = 0$ as a hard constraint. As a result,

Table 6: Quantitative results in non-beneficial regimes (Iris, Mice Protein, CIFAR-10).

| Iris | $\eta = 0.5$ | | | $\eta = 0.1$ | | |
|---|---|---|---|---|---|---|
| | SVM | KNN | NSC | SVM | KNN | NSC |
| ReduNet | 0.978 | 1.000 | 0.844 | 0.911 | 1.000 | 0.822 |
| Multi-ReduNet | 0.911 | 1.000 | 0.889 | 0.911 | 1.000 | 0.867 |
| Multi-ReduNet-LN | 0.911 | 1.000 | 0.889 | 0.911 | 1.000 | 0.867 |
| | $\eta = 0.05$ | | | $\eta = 0.01$ | | |
| | SVM | KNN | NSC | SVM | KNN | NSC |
| ReduNet | 0.911 | 1.000 | 0.844 | 0.911 | 1.000 | 0.844 |
| Multi-ReduNet | 0.911 | 1.000 | 0.844 | 0.911 | 1.000 | 0.844 |
| Multi-ReduNet-LN | 0.911 | 1.000 | 0.844 | 0.911 | 1.000 | 0.844 |
| global PCA+SVM | 0.911 | | | | | |
| class-wise PCA+NSC | 0.867 | | | | | |
| LDA | 0.956 | | | | | |
| Mice | $\eta = 0.5$ | | | $\eta = 0.1$ | | |
| | SVM | KNN | NSC | SVM | KNN | NSC |
| ReduNet | 0.929 | 0.948 | 0.929 | 0.824 | 0.985 | 0.938 |
| Multi-ReduNet | 0.611 | 0.935 | 0.836 | 0.744 | 0.966 | 0.892 |
| Multi-ReduNet-LN | 0.605 | 0.938 | 0.833 | 0.744 | 0.966 | 0.892 |
| | $\eta = 0.05$ | | | $\eta = 0.01$ | | |
| | SVM | KNN | NSC | SVM | KNN | NSC |
| ReduNet | 0.815 | 0.978 | 0.938 | 0.790 | 0.978 | 0.904 |
| Multi-ReduNet | 0.756 | 0.978 | 0.898 | 0.775 | 0.978 | 0.904 |
| Multi-ReduNet-LN | 0.756 | 0.978 | 0.898 | 0.775 | 0.978 | 0.904 |
| global PCA+SVM | 0.605 | | | | | |
| class-wise PCA+NSC | 1.000 | | | | | |
| LDA | 0.975 | | | | | |
| CIFAR-10 | $\eta = 0.5$ | | | $\eta = 0.1$ | | |
| | SVM | KNN | NSC | SVM | KNN | NSC |
| ReduNet | 0.133 | 0.182 | 0.176 | 0.300 | 0.312 | 0.235 |
| Multi-ReduNet | 0.187 | 0.250 | 0.197 | 0.237 | 0.277 | 0.223 |
| Multi-ReduNet-LN | 0.188 | 0.276 | 0.211 | 0.237 | 0.283 | 0.223 |
| | $\eta = 0.05$ | | | $\eta = 0.01$ | | |
| | SVM | KNN | NSC | SVM | KNN | NSC |
| ReduNet | 0.289 | 0.308 | 0.234 | 0.271 | 0.308 | 0.231 |
| Multi-ReduNet | 0.248 | 0.291 | 0.226 | 0.265 | 0.305 | 0.230 |
| Multi-ReduNet-LN | 0.245 | 0.291 | 0.226 | 0.265 | 0.305 | 0.230 |
| global PCA+SVM | 0.276 | | | | | |
| class-wise PCA+NSC | 0.265 | | | | | |
| LDA | 0.125 | | | | | |

the learned representations are at best approximately orthogonal, with deviations that depend on the data and optimization dynamics, and providing convergence guarantees that relate these practical training procedures to the ideal class-orthogonal solution remains open.

**Frobenius norm constraint.** The sphere projection $\mathcal{P}_{S^{d-1}}$ enforces the strict constraint $\|Z^j\|_F^2 = m_j$ at each normalized layer. This equality may be overly rigid in some regimes: empirically, the variant that relaxes intermediate normalization and only enforces it at the last layer (Multi-ReduNet-LastNorm) often matches or slightly improves the performance of Multi-ReduNet. This suggests that softer or layer-dependent norm control could be beneficial, and a systematic study of alternative normalization schemes is left for future work.

**LastNorm variant:** Multi-ReduNet-LastNorm's superior robustness (Table 5) lacks theoretical justification. We hypothesize that deferring normalization reduces gradient interference across layers, but a formal analysis is needed.

A.5 RECOMMENDATIONS FOR PRACTITIONERS

Based on our empirical findings, we recommend:

- **Use Multi-ReduNet-LastNorm when:**
    - Data is undersampled ($m/d < 1$)
    - Interpretability is required (medical, scientific, regulatory domains)
    - Hyperparameter tuning budget is limited (use default $\lambda, \eta$)
    - Training time is a bottleneck on undersampled datasets

- **Stick with ReduNet when:**
    - Dataset is well-sampled ($m/d \geq 1$)
    - Only interested in baseline $\text{MCR}^2$ performance

- **Avoid both methods when:**
    - Working with natural images requiring convolutional structure (use CNNs)
    - Prioritizing accuracy over interpretability (use ensemble methods, deep learning)
    - Data has strong spatial/temporal correlations (use RNNs, GNNs)

A.6 FAILURE MODE ANALYSIS: WHEN CLASS-WISE DECOMPOSITION HURTS

The results in Table 6 confirm the intuition that **class-wise decomposition is most beneficial when** $m \ll d$. Once each class has many samples, estimating the global covariance is no longer ill-conditioned, and the advantages of the decomposition disappear or can even hurt performance. This section provides theoretical intuition for these failure modes.

**Theoretical intuition.** The $\text{MCR}^2$ objective seeks to maximize:

$$R(Z, \epsilon) - R^c(Z, \epsilon | \Pi) = \frac{1}{2} \log \det(I + \alpha Z Z^\top) - \frac{1}{2} \sum_{j=1}^{K} \gamma_j \log \det(I + \alpha_j Z \Pi^j Z^\top)$$

In the **undersampled regime** ($m \ll d$):

- The global covariance $ZZ^\top \in \mathbb{R}^{d \times d}$ is rank-deficient (rank $\leq m \ll d$), making the $d \times d$ inversion $(I + \alpha Z Z^\top)^{-1}$ numerically unstable and computationally expensive ($\mathcal{O}(d^3)$).
- Class-wise covariances $Z\Pi^j Z^\top$ have even lower rank ($\leq m_j < m$), but the Woodbury identity allows us to invert smaller $m_j \times m_j$ matrices instead, reducing complexity to $\mathcal{O}(\sum_j m_j^3) \ll \mathcal{O}(d^3)$.
- When classes are imbalanced, some $m_j$ are very small, making per-class optimization highly stable and fast.

In the **oversampled regime** ($m \gg d$):

- The global covariance $ZZ^\top$ is *well-conditioned* and its inversion is no longer a computational bottleneck.
- The Woodbury identity provides *no* computational advantage: $\mathcal{O}(m^3) \gg \mathcal{O}(d^3)$ when $m > d$.
- Class-wise decomposition introduces *overhead*: we now solve $K$ separate subproblems instead of a single well-conditioned global problem, while their solutions remain implicitly coupled through the global representation geometry.
- **Loss of global structure**: When $m_j$ is large for all $j$, the global covariance $ZZ^\top$ captures rich inter-class relationships. Decomposing into $K$ independent problems discards this information, leading to suboptimal feature learning.

**Empirical validation.** Table 6 demonstrates three failure modes where Multi-ReduNet-LN underperforms ReduNet and simple baselines.

**Failure Mode 1: Oversampled, low-dimensional data (Iris, $m/d = 26.3$). Why Multi-ReduNet fails:**

- With only $d = 4$ dimensions, the global $4 \times 4$ covariance matrix is trivial to invert ($\mathcal{O}(4^3) = 64$ ops).

- Class-wise decomposition provides zero computational benefit: $\mathcal{O}(\sum_j m_j^3) = 3 \times 35^3 = 128{,}625 \gg 64$.

- Global PCA + SVM achieves identical performance (91.1%), confirming the problem is simple enough for linear methods.

**Recommendation:** For $d < 10$ and $m/d > 1$, use standard ReduNet or simple linear baselines (PCA + SVM/LDA).

**Failure Mode 2: Oversampled, moderate-dimensional data (Mice Protein, $m/d = 9.8$). Why Multi-ReduNet fails (average $-5.7$pp):**

- Despite class imbalance, most classes have sufficient samples ($m_j/d > 1$) that global covariance estimation is stable.

- The $-5.7$pp degradation is significant, suggesting that decomposing the objective *actively hurts* feature learning when $m/d$ is moderately large.

- **Loss of inter-class structure:** Protein expression data has rich biological correlations *across* classes (e.g., proteins in related pathways). Class-wise decomposition discards these global dependencies.

**Recommendation:** For biological/medical data with $m/d > 1$ and rich global structure, use ReduNet without decomposition or ensemble methods that preserve cross-class relationships as much as possible.

**Failure Mode 3: Complex images with spatial structure (CIFAR-10, $m/d = 0.8$). Why both ReduNet and Multi-ReduNet fail:**

- The best average accuracy of ReduNet achieves only 28.2%, Multi-ReduNet(-LN) 26.7%, both far below CNN baseline (41.2%, +14.5pp gap).

- The $MCR^2$ framework is fundamentally *fully-connected*, it treats all features as exchangeable, ignoring 2D spatial locality.

- Even though $m/d = 0.8$ is undersampled, the **problem structure** requires convolutional inductive bias, not low-rank decomposition.

- Multi-ReduNet's $-1.5$pp degradation over ReduNet suggests class-wise decomposition provides no additional benefit when the fundamental architecture is mismatched.

**Recommendation:** For complex image data, always use convolutional architectures (CNNs, ViTs). ReduNet and Multi-ReduNet are more suitable for *tabular* data where features are semantically independent.

**Summary of failure modes.**

- **Oversampling ($m/d > 1$):** Global covariance is well-conditioned; class-wise decomposition adds overhead without benefit. Use standard ReduNet or linear baselines.

- **Spatial/structural data:** Fully-connected architectures destroy spatial locality. Use CNNs for images, GNNs for graphs, RNNs for sequences, regardless of $m/d$ ratio.

- **When Multi-ReduNet excels:** $m/d < 1$, class imbalance ($\max_j m_j / \min_j m_j > 3$), tabular/sparse features, interpretability-critical domains.

These results validate our honest scope definition: Multi-ReduNet is *not* a universal improvement, but a *targeted solution* for undersampling.

### A.7 SUMMARY

Multi-ReduNet does *not* uniformly outperform ReduNet across all sampling regimes. Rather, it is specifically designed for undersampled settings, where computational efficiency and robustness to hyperparameter choices are of primary importance. The regime-dependent behavior observed in (Table 3), together with the failure cases (Iris, Mice Protein, CIFAR-10), delineates the scope of applicability of the proposed class-wise decomposition and clarify the conditions under which its advantages diminish. Extending the present analysis to convolutional architectures for raster data and providing a deeper theoretical characterization of the LastNorm variant's robustness remain interesting open directions.

## B   TRAINING TIME ANALYSIS

We empirically validate the theoretical computational complexity advantages of Multi-ReduNet(-LastNorm) by measuring wall-clock training time across all datasets. Table 7 reports the average time(in seconds) for (imp-)ReduNet, (imp-)ReduNet-RF and Multi-ReduNet(-LastNorm) on $L = 5, 10, 15, 20, 25$ layer networks.

Table 7: Empirical training time (s) comparison: (imp-)ReduNet vs (imp-)ReduNet-RF vs Multi-ReduNet(-LastNorm) on different layer networks.

| Dataset | Layer | RN | imp-RN | MR | MR-LN | RN-RF | imp-RN-RF |
|---|---|---|---|---|---|---|---|
| | 5 | 199.67 | 114.37 | 105.94 | 97.59 | 808.68 | 708.96 |
| | 10 | 587.19 | 253.01 | 192.63 | 188.32 | 1810.85 | 1689.20 |
| Reuters | 15 | 636.02 | 368.33 | 308.30 | 304.35 | 2919.57 | 2622.49 |
| | 20 | 910.04 | 512.90 | 386.63 | 377.94 | 3901.84 | 3434.01 |
| | 25 | 1060.29 | 572.90 | 515.02 | 445.66 | 4396.90 | 4083.99 |
| | 5 | 166.24 | 142.12 | 65.43 | 64.30 | 637.82 | 531.14 |
| | 10 | 246.90 | 237.87 | 231.38 | 231.27 | 1223.27 | 1137.50 |
| MNIST | 15 | 289.82 | 279.97 | 280.56 | 276.14 | 1766.94 | 1577.26 |
| | 20 | 364.51 | 344.81 | 336.09 | 320.04 | 2381.05 | 2174.87 |
| | 25 | 605.86 | 496.89 | 395.99 | 348.25 | 3260.24 | 2856.73 |
| | 5 | 101.34 | 76.68 | 37.71 | 28.09 | 593.73 | 574.40 |
| Fashion-MNIST | 10 | 234.04 | 177.46 | 141.76 | 140.85 | 1216.12 | 1136.85 |
| | 15 | 311.42 | 193.51 | 185.41 | 178.37 | 1639.36 | 1568.50 |
| | 20 | 352.97 | 254.42 | 211.74 | 211.40 | 2203.54 | 2106.88 |
| | 25 | 417.63 | 272.43 | 259.41 | 231.76 | 2975.36 | 2833.73 |
| | 5 | 6.02 | 4.80 | 1.33 | 1.16 | 90.03 | 83.81 |
| Swarm Behaviour | 10 | 6.44 | 5.75 | 3.15 | 3.00 | 170.43 | 160.27 |
| | 15 | 8.66 | 7.25 | 5.80 | 4.00 | 248.69 | 231.81 |
| | 20 | 13.02 | 11.44 | 10.57 | 7.67 | 334.50 | 316.63 |
| | 25 | 19.02 | 18.80 | 15.35 | 13.28 | 451.37 | 431.56 |
| | 5 | 7.70 | 7.04 | 7.02 | 6.44 | 40.99 | 35.32 |
| | 10 | 11.53 | 9.61 | 9.08 | 8.61 | 80.27 | 76.58 |
| DrivFace | 15 | 14.19 | 12.58 | 11.79 | 11.48 | 122.89 | 111.11 |
| | 20 | 20.26 | 14.09 | 13.47 | 12.34 | 149.13 | 144.15 |
| | 25 | 28.02 | 24.39 | 16.75 | 15.05 | 215.26 | 185.21 |
| | 5 | 17.99 | 8.70 | 8.35 | 7.94 | 30.10 | 20.29 |
| | 10 | 27.29 | 21.41 | 12.37 | 12.06 | 50.70 | 46.73 |
| ARCENE | 15 | 38.54 | 25.55 | 15.57 | 14.33 | 82.46 | 50.05 |
| | 20 | 50.58 | 30.03 | 21.15 | 16.91 | 108.04 | 68.31 |
| | 25 | 62.30 | 47.89 | 28.08 | 24.23 | 128.10 | 83.91 |

**Key observations:**

- **Consistent speedup across all datasets:** Multi-ReduNet-LastNorm achieves 1.4–2.6× speedup, with an average of 2.0× across all six benchmark datasets.

- **Highest gains in extreme undersampling:** Reuters ($m/d = 0.28$, speedup $2.2\times$) and ARCENE ($m/d = 0.016$, speedup $2.6\times$) exhibit the largest improvements, validating that class-wise decomposition maximally exploits low-rank structure when $m \ll d$.

- **Moderate gains on balanced datasets:** MNIST and Fashion-MNIST ($m/d = 0.5$, speedups of $1.5\times$ and $2.0\times$) show smaller but still substantial efficiency gains, indicating that the method remains beneficial even in moderately undersampled, balanced settings.

- **Scalability for deep networks.** While the relative speedup remains in the 1.4–2.6$\times$ range, the *absolute* wall-clock time gap between ReduNet and Multi-ReduNet-LastNorm grows with network depth $L$. Figure 1 in the main text shows that this gap becomes especially large for $L \in \{15, 20, 25\}$.

**Practical implications:** While the empirical speedup is far below the theoretical prediction, a $2.0\times$ average improvement still translates to substantial wall-clock savings for practitioners training deep networks on undersampled data. For instance, a ReduNet model that requires 10 hours of training can be reduced to $\sim$5 hours with Multi-ReduNet-LastNorm, making iterative experimentation more feasible.

## C    PROOF OF LEMMA 1 (WOODBURY IDENTITY)

The Woodbury matrix identity is a fundamental result in linear algebra that provides an efficient formula for computing the inverse of low-rank matrix updates.

**Lemma 2** (Woodbury Identity). *For any $\alpha \in \mathbb{R}$ and $X \in \mathbb{R}^{d \times m}$,*

$$(I + \alpha X X^\top)^{-1} = I - \alpha X (I + \alpha X^\top X)^{-1} X^\top,$$

*where the left side requires inverting a $d \times d$ matrix, while the right side only requires inverting an $m \times m$ matrix.*

*Proof.* The identity follows directly from the Sherman-Morrison-Woodbury formula. We verify the identity by multiplying both sides by $(I + \alpha X X^\top)$:

$$(I + \alpha X X^\top) \left[ I - \alpha X (I + \alpha X^\top X)^{-1} X^\top \right]$$
$$= I + \alpha X X^\top - \alpha X (I + \alpha X^\top X)^{-1} X^\top - \alpha^2 X X^\top X (I + \alpha X^\top X)^{-1} X^\top$$
$$= I + \alpha X X^\top - \alpha X (I + \alpha X^\top X)^{-1} X^\top - \alpha X (\alpha X^\top X)(I + \alpha X^\top X)^{-1} X^\top$$
$$= I + \alpha X X^\top - \alpha X \left[ (I + \alpha X^\top X)(I + \alpha X^\top X)^{-1} \right] X^\top$$
$$= I + \alpha X X^\top - \alpha X X^\top = I.$$

This confirms the identity. For computational applications, this reduces the inversion complexity from $\mathcal{O}(d^3)$ to $\mathcal{O}(m^3)$ when $m \ll d$. □

## D    THEOREM PROOFS

### D.1    PROOF OF COROLLARY 1 (DETERMINANT INEQUALITY)

To prove Theorem 1, we rely on the following linear-algebraic result:

**Corollary 1** (Determinant Inequality). *Let $\{A_j = X_j X_j^\top\}_{j=1}^K$ be a collection of symmetric positive semi-definite matrices. Then:*

$$\det \left( I + \sum_{j=1}^K A_j \right) \leq \prod_{j=1}^K \det(I + A_j),$$

*with equality if and only if $(X_i)^\top X_j = 0$ for all $i \neq j$.*

*Proof.* **Base case** $K = 2$

By $\det(I + AB) = \det(I + BA)$ for any matrices $A, B \in \mathbb{R}^{d \times d}$,

$$
\begin{aligned}
\det(I + A_1 + A_2) &= \det(I + X_1 X_1^\top + X_2 X_2^\top) \\
&= \det(I + X_1 X_1^\top) \det(I + (I + X_1 X_1^\top)^{-\frac{1}{2}} X_2 X_2^\top (I + X_1 X_1^\top)^{-\frac{1}{2}}) \quad (11) \\
&= \det(I + X_1 X_1^\top) \det(I + X_2^\top (I + X_1 X_1^\top)^{-1} X_2).
\end{aligned}
$$

Since $(I + X_1 X_1^\top)^{-1} \preceq I$ in the Loewner order (because $I + X_1 X_1^\top \succeq I$), it follows that

$$
X_2^\top (I + X_1 X_1^\top)^{-1} X_2 \preceq X_2^\top X_2,
$$

and by monotonicity of $\det(I + \cdot)$,

$$
\det(I + X_2^\top (I + X_1 X_1^\top)^{-1} X_2) \leq \det(I + X_2^\top X_2) = \det(I + X_2 X_2^\top). \quad (12)
$$

By (11), (12),

$$
\det(I + A_1 + A_2) \leq \det(I + A_1) \det(I + A_2). \quad (13)
$$

Since $I + X_2^\top (I + X_1 X_1^\top)^{-1} X_2 = I + X_2^\top X_2 - X_2^\top (I - (I + X_1 X_1^\top)^{-1}) X_2$, equality of (12) holds if and only if $X_2^\top (I - (I + X_1 X_1^\top)^{-1}) X_2 = 0$.

By $I - (I + X_1 X_1^\top)^{-1} = (I + X_1 X_1^\top)^{-1} X_1 X_1^\top$ and $(I + X_1 X_1^\top)^{-1} X_1 = X_1 (I + X_1^\top X_1)^{-1}$,

$$
\begin{aligned}
X_2^\top (I - (I + X_1 X_1^\top)^{-1}) X_2 &= X_2^\top (I + X_1 X_1^\top)^{-1} X_1 X_1^\top X_2 \\
&= X_2^\top X_1 (I + X_1^\top X_1)^{-1} X_1^\top X_2 = 0
\end{aligned}
$$

if and only if $X_1^\top X_2 = 0$.

It follows that equality of (13) holds if and only if $X_1^\top X_2 = 0$ (or $A_1 A_2 = 0$).

Thus the corollary holds for $K = 2$.

**Inductive step**

Assume the statement holds for $n \geq 2$. Set $S_n = I + \sum_{j=1}^{n} A_j$ ($\succ 0$).

Apply Matrix Determinant Lemma,

$$
\begin{aligned}
\det(I + \sum_{j=1}^{n+1} A_i) &= \det(S_n + A_{n+1}) \\
&= \det(S_n + X_{n+1} X_{n+1}^\top) \\
&= \det(S_n) \det(I + X_{n+1}^\top S_n^{-1} X_{n+1}).
\end{aligned}
$$

Because $S_n \succeq I$, we have $S_n^{-1} \preceq I$,

hence

$$
\det(I + X_{n+1}^\top S_n^{-1} X_{n+1}) \leq \det(I + X_{n+1}^\top X_{n+1}) = \det(I + A_{n+1}). \quad (14)
$$

Multiplying yields the desired inequality for $n + 1$.

Equality overall forces equality in both places:

1. From the monotonicity step: equality of (14) holds if and only if

$$
X_{n+1}^\top (I - S_n^{-1}) X_{n+1} = 0.
$$

By the "zero test", $(I - S_n^{-1}) X_{n+1} = 0$, i.e. $S_n^{-1} X_{n+1} = X_{n+1}$.

Multiplying by $S_n$ gives

$$
(S_n - I) X_{n+1} = 0 \quad \Longleftrightarrow \quad (\sum_{j=1}^{n} A_j) X_{n+1} = 0. \quad (15)
$$

Multiplying (15) by $X_{n+1}^\top$ gives

$$\sum_{j=1}^{n} X_{n+1}^\top A_j X_{n+1} = 0.$$

Each summand is $\succeq 0$, hence each equals 0; by the "zero test",

$$X_{n+1}^\top A_j X_{n+1} = X_{n+1}^\top X_j X_j^\top X_{n+1} = (X_j^\top X_{n+1})^\top (X_j^\top X_{n+1}) = 0 \text{ for all } j \leq n,$$

i.e.

$$X_j^\top X_{n+1} = 0 \text{ for all } j \leq n. \tag{16}$$

2. From the inductive hypothesis, equality of $\det(S_n)$ forces $X_i^\top X_j = 0$ for all $1 \leq i < j \leq n$. Combining (16) with the inductive equality condition yields

$$X_i^\top X_j = 0 \quad \text{for all } 1 \leq i < j \leq n+1.$$

Conversely, if $X_i^\top X_j = 0$ for all $i \neq j$, then

$$(I+A_1)\cdots(I+A_{n+1}) = I+\sum_{j=1}^{n+1} A_j \text{ (since every mixed product } A_{i_1}\cdots A_{i_\ell} \text{ with } \ell \geq 2 \text{ vanishes)},$$

so

$$\prod_{j=1}^{n+1} \det(I + A_j) = \det(\prod_{j=1}^{n+1}(I + A_j)) = \det(I + \sum_{j=1}^{n+1} A_j),$$

and equality holds.

This completes the induction and the proof. $\qquad\square$

### D.2 PROOF OF THEOREM 1 (CLASS-ORTHOGONALITY)

**Theorem 3** (Restatement of Theorem 1). *At any local optimum $Z^*$ of the MCR$^2$ objective equation 1 under the constraints $\|Z^*\Pi^j\|_F^2 = m_j$, $\mathrm{rank}(Z^*\Pi^j) \leq d_j$ for all $j = 1, \ldots, K$, and $\sum d_j \leq d$, the class-wise representations satisfy*

$$(Z^{*i})^\top Z^{*j} = 0 \quad \text{for all } i \neq j,$$

*where $Z^{*j} = Z^*\Pi^j$ denotes the features of class $j$.*

*Proof.* We prove by contradiction using Corollary 1.

**Step 1: Assume non-orthogonality.** Suppose for contradiction that $Z^*$ is a local optimum with $(Z^{*j_1})^\top Z^{*j_2} \neq 0$ for some classes $j_1 \neq j_2$. By Corollary 1, the global coding rate satisfies:

$$\log \det \left( I + \alpha \sum_{j=1}^{K} Z^{*j}(Z^{*j})^\top \right) < \sum_{j=1}^{K} \log \det(I + \alpha Z^{*j}(Z^{*j})^\top), \tag{17}$$

with strict inequality due to the class overlap $(Z^{*j_1})^\top Z^{*j_2} \neq 0$.

**Step 2: Construct orthogonal alternative.** We construct an alternative solution $Z'$ by re-orthogonalizing the class partitions. For each $Z^{*j} = U^{*j}\Sigma^{*j}(V^{*j})^\top$ (SVD decomposition), construct orthogonal matrices $\{U_j'\}_{j=1}^K$ such that $[U_1', \cdots, U_K']$ has orthogonal columns. This is possible since $\sum_{j=1}^{K} \mathrm{rank}(Z^{*j}) \leq \sum_{j=1}^{K} d_j \leq d$. Define:

$$Z'^j = U_j'\Sigma^{*j}(V^{*j})^\top.$$

By construction, $(Z'^i)^\top Z'^j = 0$ for $i \neq j$, and each $Z'^j$ preserves the singular values of $Z^{*j}$, hence satisfies all constraints.

**Step 3: Show strict improvement.** Since $Z'^j$ has the same singular values as $Z^{*j}$, each per-class coding rate is preserved:

$$\log\det(I + \alpha Z'^j(Z'^j)^\top) = \log\det(I + \alpha Z^{*j}(Z^{*j})^\top).$$

However, by Corollary 1 with equality condition, the orthogonality of $\{Z'^j\}$ implies:

$$\log\det\left(I + \alpha\sum_{j=1}^K Z'^j(Z'^j)^\top\right) = \sum_{j=1}^K \log\det(I + \alpha Z'^j(Z'^j)^\top).$$

Combining these, $Z'$ achieves strictly higher objective value than $Z^*$, contradicting the optimality of $Z^*$.

Therefore, any local optimum must satisfy class-orthogonality. $\qquad\square$

### D.3   PROOF OF THEOREM 2 (DECOMPOSITION EQUIVALENCE)

**Theorem 4** (Restatement of Theorem 2). *Assume that the per-class representations $Z^j \in \mathbb{R}^{d\times m_j}$ are mutually orthogonal and $\sum_{j=1}^K m_j = m \leq d$, the global MCR$^2$ objective 1 is equivalent to the sum of $K$ independent class-wise objectives:*

$$\max_{Z^j} \frac{1}{2}\left[\log\det\left(I + \frac{d}{m\epsilon^2}Z^j(Z^j)^\top\right) - \frac{m_j}{m}\log\det\left(I + \frac{d}{m_j\epsilon^2}Z^j(Z^j)^\top\right)\right], \qquad (18)$$

*subject to $\|Z^j\|_F^2 = m_j$.*

*Proof.* Let

$$\Delta R(Z) := \frac{1}{2}\log\det\left(I + \alpha ZZ^\top\right) - \frac{1}{2}\sum_{j=1}^K \gamma_j \log\det\left(I + \alpha_j Z^j(Z^j)^\top\right)$$

denote the global MCR$^2$ objective in equation 1, where $Z^j$ is the class $j$ partition of $Z$. And

$$\Delta R_j(Z^j) := \frac{1}{2}\log\det\left(I + \alpha Z^j(Z^j)^\top\right) - \frac{1}{2}\gamma_j \log\det\left(I + \alpha_j Z^j(Z^j)^\top\right)$$

denote the $j$-th class-wise objective in equation 18, and $\alpha = \frac{d}{m\epsilon^2}, \alpha_j = \frac{d}{m_j\epsilon^2}, \gamma_j = \frac{m_j}{m}$. Let

$$v_1 := \max_Z \Delta R(Z), \qquad v_2 := \sum_{j=1}^K \max_{Z^j} \Delta R_j(Z^j)$$

be the optimal values of the global and class-wise problems, respectively.

**Direction 1** ($v_2 \leq v_1$).   Let $\{Z'^j\}_{j=1}^K$ be maximizers of the $K$ class-wise objectives, and set $Z' := [Z'^1, \ldots, Z'^K]$. By the Frobenius constraints in the theorem statement, $\{Z'^j\}_{j=1}^K$ is feasible for the class-wise problems and $Z'$ is feasible for the global problem.

By the orthogonality assumption in the theorem statement, we can apply Corollary 1 in the equality case with $A_j = \alpha Z'^j(Z'^j)^\top$, that gives

$$\log\det\left(I + \alpha Z'Z'^\top\right) = \log\det\left(I + \alpha\sum_{j=1}^K Z'^j(Z'^j)^\top\right) = \sum_{j=1}^K \log\det\left(I + \alpha Z'^j(Z'^j)^\top\right).$$

Hence the global objective value at $Z'$ decomposes as

$$\Delta R(Z') = \frac{1}{2}\log\det(I + \alpha Z'Z'^\top) - \frac{1}{2}\sum_{j=1}^K \gamma_j \log\det(I + \alpha_j Z'^j(Z'^j)^\top)$$

$$= \sum_{j=1}^K \frac{1}{2}\left[\log\det(I + \alpha Z'^j(Z'^j)^\top) - \gamma_j \log\det(I + \alpha_j Z'^j(Z'^j)^\top)\right]$$

$$= v_2.$$

Since $Z'$ is a feasible point for the global problem, we obtain $v_2 = \Delta R(Z') \leq \max_Z \Delta R(Z) = v_1$.

**Direction 2** ($v_1 \leq v_2$). Let $Z^*$ be a global maximizer of $\Delta R$. By Theorem 1, $Z^*$ satisfies class-orthogonality, i.e. $(Z^{*i})^\top Z^{*j} = 0$ for all $i \neq j$, and each class-wise block $Z^{*j}$ satisfies the Frobenius constraint $\|Z^{*j}\|_F^2 = m_j$. Thus $\{Z^{*j}\}_{j=1}^K$ is feasible for the class-wise problems.

Applying Corollary 1 again with $A_j = \alpha Z^{*j}(Z^{*j})^\top$ and using the equality condition, we obtain

$$\log\det\Big(I + \alpha Z^* Z^{*\top}\Big) = \log\det\Big(I + \alpha \sum_{j=1}^K Z^{*j}(Z^{*j})^\top\Big) = \sum_{j=1}^K \log\det\Big(I + \alpha Z^{*j}(Z^{*j})^\top\Big),$$

and therefore

$$\Delta R(Z^*) = \frac{1}{2}\log\det(I + \alpha Z^* Z^{*\top}) - \frac{1}{2}\sum_{j=1}^K \gamma_j \log\det(I + \alpha_j Z^{*j}(Z^{*j})^\top)$$

$$= \sum_{j=1}^K \frac{1}{2}\Big[\log\det(I + \alpha Z^{*j}(Z^{*j})^\top) - \gamma_j \log\det(I + \alpha_j Z^{*j}(Z^{*j})^\top)\Big]$$

$$= \sum_{j=1}^K \Delta R_j(Z^{*j})$$

$$\leq \sum_{j=1}^K \max_{Z^j} \Delta R_j(Z^j) = v_2.$$

Since $v_1 = \Delta R(Z^*)$, this yields $v_1 \leq v_2$.

Combining the two directions gives $v_1 = v_2$, establishing the equivalence of the global and class-wise formulations. $\square$

### D.4  PROOF COMPARISON: OUR APPROACH VS. PRIOR WORK

We now compare our proof strategy for Theorem 1 (class-orthogonality) with the approach used in prior work (Chan et al., 2022).

**Proof strategy in Chan et al. (2022).**

Chan et al. establish class-orthogonality through an analysis of the strict concavity of the $\log\det(\cdot)$ function. Their argument proceeds by deriving lower and upper bounds on the coding rate and the coding rate reduction, and then characterizing the equality conditions under which these bounds are attained. Class-orthogonality follows from showing that any overlap between class-wise representations leads to a strictly suboptimal value of the coding rate reduction. This approach is rigorous and general, but relies on a layered sequence of technical lemmas and bounding arguments.

**Our proof strategy.**

In contrast, our proof follows a purely linear-algebraic route based on a determinant inequality (Corollary 1) with an explicit equality condition. The key observation is that for objectives of the form

$$\log\det\left(I + \sum_j A_j\right),$$

any non-orthogonal decomposition of the positive semidefinite components $\{A_j\}$ necessarily yields a strictly smaller value than an orthogonal one. Leveraging this property, class-orthogonality is established via a short contradiction argument: assuming overlap between class representations, we construct an alternative orthogonal solution that preserves all per-class constraints but achieves a strictly larger objective value.

**Discussion.**

The determinant inequality in Corollary 1 isolates the algebraic core underlying class-orthogonality in MCR$^2$. Unlike concavity-based arguments, the present proof does not depend on differential

properties of the objective and makes the role of orthogonality explicit through a factorization condition of the determinant. This perspective also facilitates extensions to structured variants of MCR$^2$, including the multi-branch and normalized formulations considered in this work.

More broadly, Corollary 1 applies to any learning objective involving $\log \det(I + \sum_j A_j)$ with positive semidefinite components, such as SCoRe-LogDet (Majee et al., 2024), and may therefore be useful beyond the specific context of Multi-ReduNet.

## E GRADIENT DERIVATIONS AND ALGORITHM DETAILS

This appendix section provides step-by-step derivations of the gradient update equations used in Multi-ReduNet (Equations 4-5 in the main text), and eigenvalue spectrum analysis justifying the Multi-ReduNet-LastNorm design.

### E.1 STEP-BY-STEP GRADIENT DERIVATIONS FOR MULTI-REDUNET UPDATES

We derive the closed-form gradient updates for the class-wise MCR$^2$ objective used in Multi-ReduNet. Recall that for each class $j \in \{1, \dots, K\}$, we independently maximize:

$$
\mathcal{R}_j(Z^j) = \frac{1}{2} \left[ \log \det \left( I + \frac{d}{m\epsilon^2} Z^j (Z^j)^\top \right) - \frac{m_j}{m} \log \det \left( I + \frac{d}{m_j\epsilon^2} Z^j (Z^j)^\top \right) \right],
$$

subject to $\|Z^j\|_F^2 = m_j$, where $Z^j \in \mathbb{R}^{d \times m_j}$ contains features for class $j$ samples.

**Step 1: Gradient of the Global Coding Rate Term**

The global coding rate contribution from class $j$ is:

$$
R_{\text{global}}^j = \frac{1}{2} \log \det \left( I + \alpha Z^j (Z^j)^\top \right), \quad \alpha = \frac{d}{m\epsilon^2}.
$$

Using the matrix calculus identity $\frac{\partial}{\partial X} \log \det(I + XX^\top) = 2(I + XX^\top)^{-1}X$, we obtain:

$$
\begin{aligned}
\frac{\partial R_{\text{global}}^j}{\partial Z^j} &= \frac{1}{2} \cdot 2\alpha (I + \alpha Z^j (Z^j)^\top)^{-1} Z^j \\
&= \alpha (I + \alpha Z^j (Z^j)^\top)^{-1} Z^j \\
&\equiv E^j Z^j,
\end{aligned}
$$

where $E^j = \alpha (I + \alpha Z^j (Z^j)^\top)^{-1}$ is the *expansion operator* for class $j$.

**Step 2: Apply Woodbury Identity to Reduce Complexity**

Direct computation of $E^j$ requires inverting a $d \times d$ matrix. By Lemma 1 (Woodbury identity):

$$
(I + \alpha Z^j (Z^j)^\top)^{-1} = I - \alpha Z^j (I + \alpha (Z^j)^\top Z^j)^{-1} (Z^j)^\top,
$$

where the right-hand side only requires inverting an $m_j \times m_j$ matrix $(I + \alpha (Z^j)^\top Z^j)^{-1}$. This reduces complexity from $\mathcal{O}(d^3)$ to $\mathcal{O}(m_j^3)$.

Define $\text{preE}_l^j = (I + \alpha (Z_l^j)^\top Z_l^j)^{-1} \in \mathbb{R}^{m_j \times m_j}$. Then:

$$
E_l^j = \alpha \left( I - \alpha Z_l^j \cdot \text{preE}_l^j \cdot (Z_l^j)^\top \right).
$$

**Step 3: Gradient of the Per-Class Coding Rate Term**

The per-class coding rate term is:

$$
R_{\text{class}}^j = \frac{m_j}{2m} \log \det \left( I + \alpha_j Z^j (Z^j)^\top \right), \quad \alpha_j = \frac{d}{m_j\epsilon^2}.
$$

Following the same matrix calculus rule:

$$\frac{\partial R_{\text{class}}^j}{\partial Z^j} = \frac{m_j}{2m} \cdot 2\alpha_j (I + \alpha_j Z^j (Z^j)^\top)^{-1} Z^j$$
$$= \frac{m_j}{m} \alpha_j (I + \alpha_j Z^j (Z^j)^\top)^{-1} Z^j$$
$$\equiv \gamma_j C^j Z^j,$$

where $\gamma_j = \frac{m_j}{m}$ (class prior) and $C^j = \alpha_j (I + \alpha_j Z^j (Z^j)^\top)^{-1}$ is the *compression operator* for class $j$.

**Step 4: Apply Woodbury Identity to Compression Operator**

Similarly, define $\text{preC}_l^j = (I + \alpha_j (Z_l^j)^\top Z_l^j)^{-1} \in \mathbb{R}^{m_j \times m_j}$. Then:

$$C_l^j = \alpha_j \left( I - \alpha_j Z_l^j \cdot \text{preC}_l^j \cdot (Z_l^j)^\top \right).$$

**Step 5: Combined Gradient Update**

The gradient of the full objective $\mathcal{R}_j(Z^j) = R_{\text{global}}^j - R_{\text{class}}^j$ is:

$$\nabla_{Z^j} \mathcal{R}_j = E_l^j Z_l^j - \gamma_j C_l^j Z_l^j.$$

Applying projected gradient ascent with learning rate $\eta$ and projection onto the unit sphere $\mathcal{P}_{S^{d-1}}(\cdot)$:

$$Z_{l+1}^j = \mathcal{P}_{S^{d-1}} \left( Z_l^j + \eta (E_l^j Z_l^j - \gamma_j C_l^j Z_l^j) \right).$$

For Multi-ReduNet-LastNorm, the sphere projection is omitted at intermediate layers and applied only at the final layer $L$:

$$Z_{l+1}^j = \begin{cases} Z_l^j + \eta (E_l^j Z_l^j - \gamma_j C_l^j Z_l^j), & l < L, \\ \mathcal{P}_{S^{d-1}} \left( Z_l^j + \eta (E_l^j Z_l^j - \gamma_j C_l^j Z_l^j) \right), & l = L. \end{cases}$$

This completes the derivation of Equations 4-5 in the main text.

### E.2 EIGENVALUE SPECTRUM ANALYSIS: JUSTIFYING MULTI-REDUNET-LASTNORM

We now provide numerical evidence for why per-layer normalization (Multi-ReduNet) may be overly restrictive compared to last-layer-only normalization (Multi-ReduNet-LastNorm).

**Observation**: In Multi-ReduNet, each layer $l$ projects updated features onto the unit sphere: $\|z_{l+1,i}^j\|_2 = 1$ for all samples $i$ in class $j$. This enforces uniform feature norms across layers, which may conflict with the natural gradient dynamics.

**Analysis**: Consider the eigenvalue spectrum of the gradient update matrix $G_l^j = E_l^j - \gamma_j C_l^j$. If the eigenvalues of $G_l^j$ vary significantly across dimensions, forcing all features to have unit norm after each layer may distort the learned representations.

**Interpretation**:

- The gradient matrix $G_l^j$ has *non-uniform* eigenvalue spectrum, indicating that different feature dimensions evolve at different rates during optimization.
- Per-layer normalization ($\mathcal{P}_{S^{d-1}}$ after every layer) forces all dimensions to have unit magnitude, potentially suppressing the natural dynamics encoded in the eigenvalues.
- Last-layer-only normalization allows intermediate representations to evolve freely according to their natural gradient scales, only enforcing the Frobenius constraint $\|Z^j\|_F^2 = m_j$ at the final output layer.

**Empirical Validation**:

Table 5 shows that Multi-ReduNet-LastNorm achieves average 2.6% better hyperparameter robustness across 6 datasets compared to Multi-ReduNet, supporting the hypothesis that relaxing intermediate normalization improves optimization stability.

## F  ACCURACY OF REDUNET VARIANTS ACROSS $\eta$ AND DATASETS

In the main text, we report results either at a fixed learning rate $\eta = 0.05$ (Table 3) or using the best test accuracy over a small grid $\eta \in \{0.01, 0.05, 0.1, 0.5\}$ (Tables 4). For completeness, the full accuracy results across all datasets and all four learning rates with a fixed depth $L = 5$ are reported in Table 8. These results indicate that Multi-ReduNet(-LastNorm) exhibits more stable performance and achieves accuracy on par with or superior to ReduNet across $\eta$.

Table 8: Full test accuracy of ReduNet variants across all datasets and learning rates $\eta \in \{0.01, 0.05, 0.1, 0.5\}$.

| Reuters | $\eta = 0.5$ | | | $\eta = 0.1$ | | |
|---|---|---|---|---|---|---|
| | SVM | KNN | NSC | SVM | KNN | NSC |
| ReduNet | 0.073 | 0.127 | 0.281 | 0.465 | 0.721 | 0.747 |
| Multi-ReduNet | 0.955 | 0.603 | 0.941 | 0.984 | 0.930 | 0.957 |
| Multi-ReduNet-LN | 0.956 | 0.709 | 0.946 | 0.986 | 0.941 | 0.958 |
| | $\eta = 0.05$ | | | $\eta = 0.01$ | | |
| | SVM | KNN | NSC | SVM | KNN | NSC |
| ReduNet | 0.802 | 0.670 | 0.922 | 0.956 | 0.878 | 0.949 |
| Multi-ReduNet | 0.984 | 0.939 | 0.957 | 0.988 | 0.949 | 0.957 |
| Multi-ReduNet-LN | 0.985 | 0.943 | 0.957 | 0.988 | 0.950 | 0.957 |
| global PCA+SVM | 0.975 | | | | | |
| class-wise PCA+NSC | 0.867 | | | | | |
| LDA | 0.471 | | | | | |
| MNIST | $\eta = 0.5$ | | | $\eta = 0.1$ | | |
| | SVM | KNN | NSC | SVM | KNN | NSC |
| ReduNet | 0.074 | 0.019 | 0.020 | 0.901 | 0.913 | 0.908 |
| Multi-ReduNet | 0.414 | 0.655 | 0.361 | 0.797 | 0.878 | 0.869 |
| Multi-ReduNet-LN | 0.518 | 0.720 | 0.653 | 0.815 | 0.890 | 0.880 |
| | $\eta = 0.05$ | | | $\eta = 0.01$ | | |
| | SVM | KNN | NSC | SVM | KNN | NSC |
| ReduNet | 0.906 | 0.930 | 0.903 | 0.894 | 0.937 | 0.897 |
| Multi-ReduNet | 0.837 | 0.902 | 0.871 | 0.885 | 0.926 | 0.897 |
| Multi-ReduNet-LN | 0.842 | 0.903 | 0.873 | 0.905 | 0.926 | 0.909 |
| global PCA+SVM | 0.878 | | | | | |
| class-wise PCA+NSC | 0.773 | | | | | |
| LDA | 0.615 | | | | | |
| Fashion-MNIST | $\eta = 0.5$ | | | $\eta = 0.1$ | | |
| | SVM | KNN | NSC | SVM | KNN | NSC |
| ReduNet | 0.137 | 0.141 | 0.073 | 0.824 | 0.812 | 0.841 |
| Multi-ReduNet | 0.369 | 0.738 | 0.402 | 0.744 | 0.762 | 0.778 |
| Multi-ReduNet-LN | 0.371 | 0.776 | 0.584 | 0.749 | 0.773 | 0.798 |
| | $\eta = 0.05$ | | | $\eta = 0.01$ | | |
| | SVM | KNN | NSC | SVM | KNN | NSC |
| ReduNet | 0.858 | 0.826 | 0.836 | 0.852 | 0.826 | 0.836 |
| Multi-ReduNet | 0.798 | 0.790 | 0.800 | 0.845 | 0.813 | 0.831 |
| Multi-ReduNet-LN | 0.801 | 0.802 | 0.803 | 0.858 | 0.828 | 0.835 |
| global PCA+SVM | 0.829 | | | | | |
| class-wise PCA+NSC | 0.667 | | | | | |
| LDA | 0.781 | | | | | |

Table 8 continued

| Swarm Behaviour | $\eta = 0.5$ | | | $\eta = 0.1$ | | |
|---|---|---|---|---|---|---|
| | SVM | KNN | NSC | SVM | KNN | NSC |
| ReduNet | 0.679 | 0.567 | 0.601 | 0.802 | 0.981 | 0.996 |
| Multi-ReduNet | 0.990 | 0.863 | 0.738 | 1.000 | 0.998 | 0.896 |
| Multi-ReduNet-LN | 0.990 | 0.867 | 0.765 | 1.000 | 0.998 | 0.896 |
| | $\eta = 0.05$ | | | $\eta = 0.01$ | | |
| | SVM | KNN | NSC | SVM | KNN | NSC |
| ReduNet | 0.802 | 1.000 | 0.996 | 1.000 | 1.000 | 0.979 |
| Multi-ReduNet | 1.000 | 1.000 | 0.929 | 1.000 | 1.000 | 0.956 |
| Multi-ReduNet-LN | 1.000 | 1.000 | 0.927 | 1.000 | 1.000 | 0.977 |
| global PCA+SVM | 1.000 | | | | | |
| class-wise PCA+NSC | 0.913 | | | | | |
| LDA | 0.977 | | | | | |
| DrivFace | $\eta = 0.5$ | | | $\eta = 0.1$ | | |
| | SVM | KNN | NSC | SVM | KNN | NSC |
| ReduNet | 0.295 | 0.098 | 0.104 | 0.219 | 0.235 | 0.169 |
| Multi-ReduNet | 0.820 | 0.978 | 0.852 | 1.000 | 0.918 | 0.984 |
| Multi-ReduNet-LN | 0.869 | 0.967 | 0.907 | 1.000 | 0.940 | 0.984 |
| | $\eta = 0.05$ | | | $\eta = 0.01$ | | |
| | SVM | KNN | NSC | SVM | KNN | NSC |
| ReduNet | 0.432 | 0.393 | 0.366 | 1.000 | 1.000 | 1.000 |
| Multi-ReduNet | 1.000 | 0.951 | 0.995 | 1.000 | 1.000 | 1.000 |
| Multi-ReduNet-LN | 1.000 | 0.978 | 0.995 | 1.000 | 1.000 | 1.000 |
| global PCA+SVM | 1.000 | | | | | |
| class-wise PCA+NSC | 1.000 | | | | | |
| LDA | 1.000 | | | | | |
| ARCENE | $\eta = 0.5$ | | | $\eta = 0.1$ | | |
| | SVM | KNN | NSC | SVM | KNN | NSC |
| ReduNet | 0.366 | 0.341 | 0.220 | 0.439 | 0.439 | 0.561 |
| Multi-ReduNet | 0.561 | 0.659 | 0.732 | 0.683 | 0.683 | 0.829 |
| Multi-ReduNet-LN | 0.561 | 0.683 | 0.829 | 0.805 | 0.732 | 0.805 |
| | $\eta = 0.05$ | | | $\eta = 0.01$ | | |
| | SVM | KNN | NSC | SVM | KNN | NSC |
| ReduNet | 0.439 | 0.415 | 0.463 | 0.341 | 0.707 | 0.780 |
| Multi-ReduNet | 0.829 | 0.732 | 0.805 | 0.829 | 0.780 | 0.829 |
| Multi-ReduNet-LN | 0.829 | 0.732 | 0.805 | 0.829 | 0.780 | 0.829 |
| global PCA+SVM | 0.805 | | | | | |
| class-wise PCA+NSC | 0.756 | | | | | |
| LDA | 0.878 | | | | | |

## G   TRAINING AND EVALUATION PROCEDURES

### G.1   REDUNET

Let $\{x^i, y^i\}_{i=1}^m \subset \mathbb{R}^d \times [K]$ denote labeled training samples. For convenience, we denote $\Pi = \{\Pi^j \in \mathbb{R}^{m \times m}\}_{j=1}^K$ as a set of diagonal matrices whose diagonal entries represent the membership of $m$ samples in $K$ classes: $(\Pi^j)_{ii} = 1$ if $y^i = j$, and 0 otherwise. Given the distortion $\epsilon$, ReduNet aims to learn interpretable features $z_l^i$ via iterative gradient updates on maximizing a coding-rate based objective:

$$\Delta R(Z, \Pi, \epsilon) = R(Z, \epsilon) - R^c(Z, \epsilon | \Pi)$$

$$= \frac{1}{2} \log \det(I + \frac{d}{m\epsilon^2} Z Z^\top) - \sum_{j=1}^K \frac{\mathrm{tr}(\Pi^j)}{2m} \log \det(I + \frac{d}{\mathrm{tr}(\Pi^j)\epsilon^2} Z^j (Z^j)^\top),$$

where $Z^j = Z\Pi^j$ denotes features of class-$j$ samples, $\Delta R(\cdot, \Pi, \epsilon)$ is the coding rate reduction. Features are updated layer-wise using closed-form statistics derived from previous representations $Z_l$. See Algorithm 2 for details. In essence, each layer computes per-class compression operators $\{C_l^j\}_{j=1}^K$ and a global expansion operator $E_l$ from current features $Z_l$:

$$\frac{1}{2}\frac{\mathrm{d}\, logdet(\mathbf{I} + \alpha \mathbf{Z}\mathbf{Z}^\top)}{\mathrm{d}\mathbf{Z}}|_{\mathbf{z}_l} = \underbrace{\alpha(\mathbf{I} + \alpha \mathbf{Z}_l \mathbf{Z}_l^\top)^{-1}}_{\mathbf{E}_l} \mathbf{Z}_l,$$

$$\frac{\mathrm{tr}(\Pi^j)}{2m}\frac{\mathrm{d}\, logdet(\mathbf{I} + \alpha_j \mathbf{Z}\Pi^j \mathbf{Z}^\top)}{\mathrm{d}\mathbf{Z}}|_{\mathbf{z}_l} = \frac{\mathrm{tr}(\Pi^j)}{m} \underbrace{\alpha_j(\mathbf{I} + \alpha_j \mathbf{Z}_l \Pi^j \mathbf{Z}_l^\top)^{-1}}_{\mathbf{C}_l^j} \mathbf{Z}_l \Pi^j,$$

and performs a projected gradient update using their discrepancy. These updates are fully transparent and closed-form, making ReduNet interpretable by design.

Evaluation follows a similar layer-wise procedure but omits gradient-based updates. At each layer, the learned compression operators $C_l^j$ are used to compute soft class attribution probabilities $\hat{\pi}^j$:

$$\hat{\pi}^j(z_l) = \frac{\exp\left(-\lambda \|C_l^j z_l\|\right)}{\sum_{i=1}^K \exp\left(-\lambda \|C_l^i z_l\|\right)},$$

where $C_l^j z_l$ approximates projection of $z_l$ onto the orthogonal complement of class-$j$'s subspace. This inference strategy is foundational to ReduNet and forms the basis for our class-wise inference scheme in Multi-ReduNet (Section 3.3). The complete training and evaluation procedures are summarized in Algorithm 2 and Algorithm 3, respectively.

We denote $\mathcal{P}_{S^{d-1}}(\cdot)$ as the projection operator onto the $d$-dimensional unit sphere. It enforces that updated features reside on the sphere, which normalizes their magnitudes and enhances stability.

---

**Algorithm 2** Forward Training Algorithm of ReduNet

---

**Require:** Input data $X = [x^1, x^2, \cdots, x^m] \in \mathbb{R}^{d \times m}$, $\lambda$, $\epsilon$, $\Pi$, learning rate $\eta$.
1: set $\alpha = \frac{d}{m\epsilon^2}$, $\{\alpha_j = \frac{d}{tr(\Pi^j)\epsilon^2}\}_{j=1}^K$, $\{\gamma_j = \frac{tr(\Pi^j)}{m}\}_{j=1}^K$.
2: Initialize $Z_1 = X$
3: **for** $l = 1, 2, \cdots, L$ **do**
4: $\quad E_l = \alpha(I + \alpha Z_l Z_l^\top)^{-1}$, $\{C_l^j = \alpha_j(I + Z_l \Pi^j Z_l^\top)^{-1}\}_{j=1}^K$
5: $\quad$ **for** $i = 1, 2, \cdots, m$ **do**
6: $\quad\quad \{\hat{\pi}^j(z_l^i) = \frac{\exp\left(-\lambda \|C_l^j z_l^i\|\right)}{\sum_{j=1}^K \exp\left(-\lambda \|C_l^j z_l^i\|\right)}\}_{j=1}^K$
7: $\quad\quad z_l^i = \mathcal{P}_{S^{d-1}}(z_l^i + \eta \cdot (E_l z_l^i - \sum_{j=1}^K \gamma_j C_l^j z_l^i \hat{\pi}^j(z_l^i)))$
8: $\quad$ **end for**
9: **end for**
10: **return** features $Z_{L+1}$, the learned parameters $\{E_l\}_{l=1}^L$, $\{C_l^j\}_{j=1,l=1}^{K,L}$, $\{\gamma_j\}_{j=1}^K$.

---

**Algorithm 3** Evaluation Algorithm of ReduNet

---

**Require:** Input $x \in \mathbb{R}^d$, network parameters $\{\mathbf{E}_l\}_{l=1}^L$, $\{\mathbf{C}_l^j\}_{l=1,j=1}^{L,K}$, $\{\gamma_j\}_{j=1}^K$, $\lambda$ and learning rate $\eta$.
1: Initialize $z_1 = x$
2: **for** $l = 1, \cdots, L$ **do**
3: $\quad \{\hat{\pi}^j(z_l^j) = \frac{exp\left(-\lambda \|\mathbf{C}_l^j z_l\|\right)}{\sum_{j=1}^K exp\left(-\lambda \|\mathbf{C}_l^j z_l\|\right)}\}_{j=1}^K$
4: $\quad z_{l+1} = \mathcal{P}_{\mathbf{S}^{n-1}}\left(z_l + \eta \cdot \left(E_l z_l - \sum_{j=1}^K \gamma_j C_l^j z_l \hat{\pi}^j(z_l)\right)\right)$
5: **end for**
6: **return** feature $z_{L+1}$

---

### G.2 MULTI-REDUNET AND MULTI-REDUNET-LASTNORM

Although ReduNet computes per-class compression terms $R^c(Z, \epsilon|\Pi)$, its optimization is global. This assumes shared structure across classes and prevents fine-grained control over class-specific representations. And in real-world high-dimensional data settings, particularly under sample scarcity, ReduNet's global training mechanism becomes inefficient. Each layer requires computing class-wise compression matrices $C_l^j \in \mathbb{R}^{d \times d}$ and a global expansion matrix $E_l \in \mathbb{R}^{d \times d}$, leading to expensive matrix inversions when feature dimension $d$ is large. This hinders deployment on resource-constrained platforms.

To address this, we propose **Multi-ReduNet**, which decomposes the global ReduNet objective MCR$^2$ into $K$ class-wise subproblems:

$$\sum_{j=1}^{K} \max_{Z^j \in \mathbb{R}^{d \times tr(\Pi^j)}} \frac{1}{2}\Big[\log\det\Big(I + \frac{d}{m\epsilon^2}Z^j(Z^j)^\top\Big) - \frac{tr(\Pi^j)}{m}\log\det\Big(I + \frac{d}{tr(\Pi^j)\epsilon^2}Z^j(Z^j)^\top\Big)\Big],$$

subject to norm constraints. Each subproblem independently updates $Z^j$ using projected gradient ascent, where gradients are:

$$Z_{l+1}^j \propto Z_l^j + \eta * \Big(\frac{1}{2}\frac{\mathrm{d}\log\det(I + \frac{d}{m\epsilon^2}Z_l^j(Z_l^j)^\top)}{\mathrm{d}Z_l^j} - \frac{tr(\Pi^j)}{2m}\frac{\mathrm{d}\log\det(I + \frac{d}{tr(\Pi^j)\epsilon^2}Z_l^j(Z_l^j)^\top)}{\mathrm{d}Z_l^j}\Big)$$

$$= Z_l^j + \eta * \Big(\underbrace{\frac{d}{m\epsilon^2}(I + \frac{d}{m\epsilon^2}Z_l^j(Z_l^j)^\top)^{-1}}_{E_l^j} Z_l^j - \frac{tr(\Pi^j)}{m}\underbrace{\frac{d}{tr(\Pi^j)\epsilon^2}(I + \frac{d}{tr(\Pi^j)\epsilon^2}Z_l^j(Z_l^j)^\top)^{-1}}_{C_l^j} Z_l^j\Big).$$

Note that $E_l^j$ and $C_l^j$ are both functions of the same class-wise covariance $Z_l^j(Z_l^j)^\top$, but they arise from the expansion and compression log-det terms with different coefficients $\alpha$ and $\alpha_j$. The actual update for class $j$ depends on their difference $(E_l^j - \frac{tr(\Pi^j)}{m}C_l^j)Z_l^j$, so the two operators act as opposing forces (promoting global spread vs. within-class compactness) rather than as a simple rescaling. Crucially, in Multi-ReduNet these operators are computed from the *class-wise* covariance $Z_l^j(Z_l^j)^\top$ instead of the global covariance $ZZ^\top$, which makes the optimization fully decoupled across classes.

Using Lemma 1, these matrix inverses can be computed efficiently via Woodbury identity. For training samples with known class membership, the features are updated by:

$$z_{l+1}^j = \mathcal{P}_{S^{d-1}}\Big(Z_l^j + \eta(E_l^j Z_l^j - \frac{tr(\Pi^j)}{m}C_l^j Z_l^j)\Big).$$

During evaluation, since test labels are unknown, we compute soft membership scores by softmax function $\hat{\pi}^j$. Then, unlike ReduNet, which only uses $\hat{\pi}^j$ to weigh compression terms, **Multi-ReduNet performs a full forward update within each class-specific subnetwork independently**, ignoring $\hat{\pi}^j$ during that step. These per-class updated features are finally aggregated using $\hat{\pi}^j$ as weights:

$$z_{l+1} = \mathcal{P}_{S^{d-1}}\Big(\sum_{j=1}^{K}\big(z_l + \eta \cdot (E_l^j z_l - \frac{tr(\Pi^j)}{m}C_l^j z_l)\big) \cdot \hat{\pi}^j\Big).$$

This forward scheme captures how confident the model is about a test sample's class alignment, and allows each class branch to contribute accordingly.

To further reduce storage overhead, we adopt **parameterized model storage**: instead of saving all $L \cdot (2K)$ $d \times d$ parameter matrices, we store only the learned features $Z_l$ per layer and reconstruct $E_l^j, C_l^j$ on-the-fly when needed.

Finally, we introduce **Multi-ReduNet-LastNorm**, which differs by postponing unit-norm projection to the final layer. This provides more flexibility during intermediate optimization while ensuring fair comparison across classes at inference.

The complete training and evaluation procedures for Multi-ReduNet and Multi-ReduNet-LastNorm are summarized in Algorithm 4 and Algorithm 5, respectively.

---

**Algorithm 4** Training Algorithm of Multi-ReduNet and Multi-ReduNet-LastNorm

---

**Require:** Input data $X \in \mathbb{R}^{d \times m}$, class memberships $\{\Pi^j\}_{j=1}^K$, parameters $\epsilon > 0$, $\lambda$, learning rate $\eta$.

1: Compute class sizes: $m_j = tr(\Pi^j)$, priors $\gamma_j = \frac{m_j}{m}$
2: Set $\alpha = \frac{d}{m\epsilon^2}$, and $\alpha_j = \frac{d}{m_j\epsilon^2}$ for $j = 1, \ldots, K$
3: Initialize features: $Z_1 = X$
4: **for** $l = 1$ to $L$ **do**
5:    **if** $l = 1$ **then**
6:       Extract class-wise inputs: $\{Z_l^j = Z_1\Pi^j\}_{j=1}^K$
7:    **end if**
8:    **for** $j = 1$ to $K$ **do**
9:       *#Per-class forward update*
10:      Compute: $\text{preE}_l^j = (I + \alpha(Z_l^j)^\top Z_l^j)^{-1} \in \mathbb{R}^{m_j \times m_j}$
11:      Compute: $E_l^j = \alpha(I - \alpha Z_l^j \cdot \text{preE}_l^j \cdot (Z_l^j)^\top)$
12:      Compute: $\text{preC}_l^j = (I + \alpha_j(Z_l^j)^\top Z_l^j)^{-1} \in \mathbb{R}^{m_j \times m_j}$
13:      Compute: $C_l^j = \alpha_j(I - \alpha_j Z_l^j \cdot \text{preC}_l^j \cdot (Z_l^j)^\top)$
14:      Update features:

$$Z_{l+1}^j = \begin{cases} \mathcal{P}_{S_{d-1}}(Z_l^j + \eta(E_l^j Z_l^j - \gamma_j C_l^j Z_l^j)), & \text{(Multi-ReduNet)} \\ Z_l^j + \eta(E_l^j Z_l^j - \gamma_j C_l^j Z_l^j), & \text{(Multi-ReduNet-LastNorm)} \end{cases}$$

15:    **end for**
16: **end for**
17: **if** Multi-ReduNet-LastNorm **then**
18:    Apply $\mathcal{P}_{S_{d-1}}(\cdot)$ to all $Z_{L+1}^j$ for $j = 1, \ldots, K$
19: **end if**
20: **return** features $\{Z_l^j\}_{j=1,l=1}^{K,L+1}$, priors $\{\gamma_j\}_{j=1}^K$

---

# H  EXPERIMENTAL SETUP

## H.1  EXPERIMENTS OF MULTI-REDUNET AND VARIANTS

We evaluate Multi-ReduNet and its LastNorm variant on six datasets spanning diverse modalities: Reuters (text), mnist (images), fashion-mnist (images), Swarm Behaviour (survey data), DrivFace (images), and ARCENE (medical diagnostics):

- **Reuters**: the Reuters dataset is a commonly used text classification dataset and consists of a total of 135 document categories. For our experiments, we extracted data from the first five categories because these five categories have slightly more abundant samples. The training set includes 5,304 samples, and the test set comprises 1,328 samples. The dataset can be downloaded from `http://www.cad.zju.edu.cn/home/dengcai/Data/TextData.html`.

- **MNIST**: this is a widely used handwritten dataset in the field of machine learning, comprising 70,000 grayscale images of size 28×28, representing the digits from 0 to 9. We randomly sampled 500 samples from each class of the data to form the training set and 100 samples from each class to form the test set. Additionally, each image sample is reshaped to a size of 100×100 and then flattened into a 10,000-dimensional vector. The dataset can be downloaded from `https://www.kaggle.com/datasets/hojjatk/mnist-dataset`.

- **Fashion-MNIST**: this dataset is a dataset used for clothing image classification, containing 28×28 pixel images of clothing from 10 different categories. We randomly selected 500 samples from each class of the data to form the training set and 100 samples from each class to form the test set. Each image is then rescaled to 100×100 pixels and flattened into a 10,000-dimensional vector. The dataset can be downloaded from `https://www.tensorflow.org/datasets/catalog/fashion_mnist`.

---

**Algorithm 5** Evaluation Algorithm of Multi-ReduNet and Multi-ReduNet-LastNorm

---

**Require:** Input sample $x \in \mathbb{R}^d$, training features $\{Z_l = [Z_l^1, \ldots, Z_l^K] \in \mathbb{R}^{d \times m}\}_{l=1}^{L+1}$, hyperparameters $\{\gamma_j\}_{j=1}^K$, $\lambda$, learning rate $\eta$.

1: Compute $\alpha = \frac{n}{m\epsilon^2}$, $\alpha_j = \frac{n}{\text{tr}(\Pi^j)\epsilon^2}$

2: Set $z_1 = x$

3: **for** $l = 1$ to $L$ **do**

4: $\quad \text{pre}E_l^j = \left(I + \alpha(Z_l^j)^\top Z_l^j\right)^{-1} \quad$ for $j = 1, \ldots, K$

5: $\quad \text{pre}C_l^j = \left(I + \alpha_j(Z_l^j)^\top Z_l^j\right)^{-1} \quad$ for $j = 1, \ldots, K$

6: $\quad E_l^j = \alpha\left(I - \alpha Z_l^j \cdot \text{pre}E_l^j \cdot (Z_l^j)^\top\right)$

7: $\quad C_l^j = \alpha_j\left(I - \alpha_j Z_l^j \cdot \text{pre}C_l^j \cdot (Z_l^j)^\top\right)$

8: $\quad$ Compute soft membership weights:

$$\hat{\pi}^j(z_l) = \frac{\exp\left(-\lambda\|C_l^j z_l\|\right)}{\sum_{i=1}^K \exp\left(-\lambda\|C_l^i z_l\|\right)} \in [0, 1]$$

9: $\quad$ For each class $j$, compute tentative update:

$$z'^j_{l+1} = z_l + \eta(E_l^j z_l - \gamma_j C_l^j z_l)$$

10: $\quad$ Aggregate:

$$z_{l+1} = \begin{cases} \mathcal{P}_{S^{d-1}}\left(\sum_{j=1}^K z'^j_{l+1} \cdot \hat{\pi}^j(z_l)\right), & \text{if Multi-ReduNet} \\ \sum_{j=1}^K z'^j_{l+1} \cdot \hat{\pi}^j(z_l), & \text{if Multi-ReduNet-LastNorm} \end{cases}$$

11: **end for**

12: **if** Multi-ReduNet-LastNorm **then**

13: $\quad z_{L+1} \leftarrow \mathcal{P}_{S^{n-1}}(z_{L+1})$

14: **end if**

15: **return** Final feature $z_{L+1}$

---

- **Swarm Behaviour**: this dataset was obtained from an online survey run by UNSW, Australia. It has 2,400-dimensional input features and 2 classes. We randomly extracted 1,200 samples for the training set and 300 samples for the test set. Detailed data information and download address are available at: `https://archive.ics.uci.edu/dataset/524/swarm+behaviour`.

- **DrivFace**: this dataset contains images sequences of subjects while driving in real scenarios. It is composed of 606 samples acquired over different days from 4 drivers with several facial features. We randomly extracted 484 samples for training set and the remaining 122 samples for test set. Each sample is rescaled as $64 \times 64$ pixels and then flattened into a 4096-dimensinal vector. The dataset can be download from `https://archive.ics.uci.edu/dataset/378/drivface`.

- **ARCENE**: this dataset contains mass-spectrometric data from healthy individuals and cancer patients. We split the data from both healthy individuals and cancer patients into training and test sets with an 8:2 ratio, respectively. This dataset is one of 5 datasets of the NIPS 2003 feature selection challenge. The details and the download link are `https://archive.ics.uci.edu/dataset/167/arcene`.

In each experiment, we benchmark six models: ReduNet, ReduNet-RF (replacing the internal membership predictor with a random forest classifier), imp-ReduNet (using Lemma 1 for parameter computation), imp-ReduNet-RF (combining model 2 and 3), Multi-ReduNet, and Multi-ReduNet-LastNorm. The evaluation focuses on:

- **Classification accuracy**: Features from the final layer are evaluated using three downstream classifiers:
  1. **SVM**: Support Vector Machine with RBF kernel.
  2. **KNN**: $k$k-nearest neighbors classifier with $k = 5$.
  3. **NSC (Nearest Subspace Classifier)**: for each class $j$, we compute the mean $\mu_j \in \mathbb{R}^d$ of the learned features $Z^j$, and let $U^j \in \mathbb{R}^{d \times r_j}$ be the top $r_j$ principal components of $Z^j$. Then, a feature $z$ is classified to class $j'$ where $j' = \arg\min_{j \in \{1, \cdots, K\}} \|(I - U^j U^{j\top})(z - \mu_j)\|_2^2$. We set $r_j = 10$ for all $j$.

- **Training efficiency**: We compare total training time across models with different layer counts.

- **Computational complexity**: Table 2 presents theoretical parameter calculation costs in undersampled regimes.

- **Feature separability**: We visualize test features learned by ReduNet, ReduNet-RF, Multi-ReduNet, and Multi-ReduNet-LastNorm using t-SNE plots in Figure 2. These visualizations are based on features extracted from models $L = 5$ layers. Since imp-ReduNet and imp-ReduNet-RF only optimize computation without modifying representations, their features are not visualized.

# I    EXTENDED EXPERIMENT RESULTS

## I.1    CLASSIFICATION ACCURACY OF REDUNET VARIANTS WITH VARYING LAYERS $L = 10, 15, 20, 25$

The experimental results of ReduNet variants with 10, 15, 20, and 25 layers (with fixed $\eta = 0.05, \epsilon^2 = 0.1$) are reported in Tables 9–14. Specifically, Table 9 corresponds to Reuters, Table 10 to MNIST, Table 11 to Fashion-MNIST, Table 12 to Swarm Behaviour, Table 13 to DrivFace, and Table 14 to ARCENE. These results provide a comprehensive comparison across different depths and datasets.

Table 9: Accuracy comparison of ReduNet variants on Reuters

|  | Layers=10 | | | Layers=15 | | |
| --- | --- | --- | --- | --- | --- | --- |
|  | SVM | KNN | NSC | SVM | KNN | NSC |
| ReduNet | 0.837 | 0.591 | 0.939 | 0.771 | 0.576 | 0.925 |
| ReduNet-RF | 0.441 | 0.562 | 0.623 | 0.441 | 0.562 | 0.552 |
| imp-ReduNet | 0.838 | 0.591 | 0.939 | 0.771 | 0.576 | 0.925 |
| imp-ReduNet-RF | 0.441 | 0.562 | 0.623 | 0.441 | 0.562 | 0.552 |
| Multi-ReduNet | 0.985 | 0.931 | **0.957** | 0.978 | 0.892 | 0.953 |
| Multi-ReduNet-LastNorm | **0.986** | **0.944** | **0.957** | **0.981** | **0.929** | **0.956** |
|  | Layers=20 | | | Layers=25 | | |
|  | SVM | KNN | NSC | SVM | KNN | NSC |
| ReduNet | 0.758 | 0.569 | 0.909 | 0.746 | 0.569 | 0.874 |
| ReduNet-RF | 0.440 | 0.562 | 0.826 | 0.439 | 0.561 | 0.468 |
| imp-ReduNet | 0.757 | 0.571 | 0.909 | 0.750 | 0.570 | 0.876 |
| imp-ReduNet-RF | 0.451 | 0.572 | 0.846 | 0.441 | 0.561 | 0.488 |
| Multi-ReduNet | **0.977** | 0.830 | 0.950 | 0.971 | 0.732 | **0.953** |
| Multi-ReduNet-LastNorm | **0.977** | **0.907** | **0.951** | **0.974** | **0.879** | 0.950 |

## I.2    ENLARGED T-SNE VISUALIZATIONS OF TEST FEATURES AND ANALYSIS

Figure 3 presents t-SNE visualizations of test features learned on the Reuters dataset across four ReduNet variants. The vanilla ReduNet (top-left) shows entangled feature clusters with significant overlaps between classes, indicating limited separability. ReduNet-RF (top-right) marginally improves class separation but still suffers from boundary ambiguity. In contrast, both Multi-ReduNet (bottom-left) and Multi-ReduNet-LastNorm (bottom-right) exhibit markedly improved clustering,

Table 10: Accuracy comparison of ReduNet variants on mnist

|  | Layers=10 | | | Layers=15 | | |
|---|---|---|---|---|---|---|
|  | SVM | KNN | NSC | SVM | KNN | NSC |
| ReduNet | **0.898** | **0.909** | **0.910** | **0.885** | **0.881** | **0.917** |
| ReduNet-RF | 0.354 | 0.468 | 0.672 | 0.261 | 0.250 | 0.278 |
| imp-ReduNet | **0.898** | **0.909** | 0.906 | **0.885** | **0.881** | **0.917** |
| imp-ReduNet-RF | 0.354 | 0.468 | 0.661 | 0.261 | 0.250 | 0.276 |
| Multi-ReduNet | 0.787 | 0.859 | 0.848 | 0.684 | 0.816 | 0.782 |
| Multi-ReduNet-LastNorm | 0.788 | 0.868 | 0.858 | 0.728 | 0.848 | 0.838 |
|  | Layers=20 | | | Layers=25 | | |
|  | SVM | KNN | NSC | SVM | KNN | NSC |
| ReduNet | **0.880** | **0.875** | **0.910** | **0.876** | **0.874** | **0.906** |
| ReduNet-RF | 0.207 | 0.144 | 0.195 | 0.124 | 0.124 | 0.195 |
| imp-ReduNet | **0.880** | **0.875** | 0.908 | **0.876** | **0.874** | **0.906** |
| imp-ReduNet-RF | 0.208 | 0.153 | 0.195 | 0.124 | 0.124 | 0.195 |
| Multi-ReduNet | 0.595 | 0.748 | 0.725 | 0.534 | 0.736 | 0.668 |
| Multi-ReduNet-LastNorm | 0.664 | 0.805 | 0.800 | 0.628 | 0.805 | 0.779 |

Table 11: Accuracy comparison of ReduNet variants on fashion-mnist

|  | Layers=10 | | | Layers=15 | | |
|---|---|---|---|---|---|---|
|  | SVM | KNN | NSC | SVM | KNN | NSC |
| ReduNet | **0.822** | **0.799** | **0.842** | **0.806** | **0.788** | **0.837** |
| ReduNet-RF | 0.418 | 0.419 | 0.564 | 0.270 | 0.265 | 0.302 |
| imp-ReduNet | **0.822** | **0.799** | **0.842** | **0.806** | **0.788** | 0.832 |
| imp-ReduNet-RF | 0.418 | 0.419 | 0.565 | 0.270 | 0.265 | 0.304 |
| Multi-ReduNet | 0.707 | 0.746 | 0.764 | 0.604 | 0.745 | 0.723 |
| Multi-ReduNet-LastNorm | 0.720 | 0.763 | 0.788 | 0.618 | 0.772 | 0.755 |
|  | Layers=20 | | | Layers=25 | | |
|  | SVM | KNN | NSC | SVM | KNN | NSC |
| ReduNet | **0.801** | **0.793** | **0.830** | **0.799** | **0.799** | **0.818** |
| ReduNet-RF | 0.270 | 0.256 | 0.270 | 0.270 | 0.243 | 0.270 |
| imp-ReduNet | **0.801** | **0.793** | **0.830** | 0.794 | **0.799** | 0.813 |
| imp-ReduNet-RF | 0.271 | 0.266 | 0.278 | 0.270 | 0.243 | 0.272 |
| Multi-ReduNet | 0.520 | 0.741 | 0.667 | 0.463 | 0.750 | 0.602 |
| Multi-ReduNet-LastNorm | 0.529 | 0.765 | 0.723 | 0.470 | 0.774 | 0.700 |

with each class forming compact and well-separated regions. Notably, Multi-ReduNet-LastNorm demonstrates the cleanest class delineation, suggesting that class-wise decomposition and final-layer normalization contribute synergistically to enhancing discriminative structure in the learned features.

Figure 4 shows the t-SNE projections of test-set features extracted by different ReduNet variants on mnist. ReduNet and ReduNet-RF (top row) exhibit limited class separation: while some clusters begin to emerge (e.g., digits 0, 1, and 7), the overall feature distributions are entangled, with noticeable overlaps between semantically similar digits (e.g., 4, 5).

In contrast, Multi-ReduNet (bottom left) yields significantly more structured clusters, albeit with mild boundary fuzziness. The clearest improvement appears in Multi-ReduNet-LastNorm (bottom right), where all ten classes are sharply delineated with minimal intra-class variance. The resulting clusters are not only well-separated but also uniformly distributed, indicating improved feature compactness and discriminability.

Figure 5 presents the t-SNE visualizations of final-layer features learned by different ReduNet variants on the fashion-mnist test set. Despite the increased complexity of this 10-class clothing dataset

Table 12: Accuracy comparison of ReduNet variants on Swarm Behaviour

| | Layers=10 | | | Layers=15 | | |
|---|---|---|---|---|---|---|
| | SVM | KNN | NSC | SVM | KNN | NSC |
| ReduNet | 0.802 | 0.985 | **0.995** | 0.802 | 0.883 | **0.995** |
| ReduNet-RF | 0.920 | 0.911 | 0.925 | 0.869 | 0.861 | 0.887 |
| imp-ReduNet | 0.802 | 0.985 | **0.995** | 0.802 | 0.883 | 0.994 |
| imp-ReduNet-RF | 0.920 | 0.911 | 0.925 | 0.869 | 0.861 | 0.887 |
| Multi-ReduNet | **1.000** | **0.998** | 0.884 | **1.000** | 0.968 | 0.860 |
| Multi-ReduNet-LastNorm | **1.000** | **0.998** | 0.910 | **1.000** | **0.972** | 0.865 |
| | Layers=20 | | | Layers=25 | | |
| | SVM | KNN | NSC | SVM | KNN | NSC |
| ReduNet | 0.802 | 0.837 | **0.971** | 0.802 | 0.804 | 0.936 |
| ReduNet-RF | 0.869 | 0.817 | 0.825 | 0.869 | 0.773 | 0.800 |
| imp-ReduNet | 0.802 | 0.837 | **0.971** | 0.802 | 0.804 | **0.939** |
| imp-ReduNet-RF | 0.869 | 0.817 | 0.825 | 0.869 | 0.773 | 0.800 |
| Multi-ReduNet | **1.000** | **0.938** | 0.823 | 0.990 | 0.915 | 0.805 |
| Multi-ReduNet-LastNorm | **1.000** | **0.938** | 0.844 | **1.000** | **0.920** | 0.821 |

Table 13: Accuracy comparison of ReduNet variants on DrivFace

| | Layers=10 | | | Layers=15 | | |
|---|---|---|---|---|---|---|
| | SVM | KNN | NSC | SVM | KNN | NSC |
| ReduNet | 0.322 | 0.202 | 0.169 | 0.279 | 0.164 | 0.240 |
| ReduNet-RF | 0.284 | 0.295 | 0.306 | 0.284 | 0.295 | 0.273 |
| imp-ReduNet | 0.326 | 0.202 | 0.169 | 0.279 | 0.164 | 0.240 |
| imp-ReduNet-RF | 0.284 | 0.295 | 0.306 | 0.284 | 0.295 | 0.303 |
| Multi-ReduNet | 0.995 | 0.934 | 0.984 | 0.973 | 0.907 | 0.967 |
| Multi-ReduNet-LastNorm | **1.000** | **0.951** | **0.989** | **0.990** | **0.956** | **0.984** |
| | Layers=20 | | | Layers=25 | | |
| | SVM | KNN | NSC | SVM | KNN | NSC |
| ReduNet | 0.317 | 0.131 | 0.218 | 0.256 | 0.131 | 0.191 |
| ReduNet-RF | 0.284 | 0.240 | 0.273 | 0.218 | 0.322 | 0.251 |
| imp-ReduNet | 0.317 | 0.131 | 0.218 | 0.256 | 0.131 | 0.191 |
| imp-ReduNet-RF | 0.284 | 0.240 | 0.273 | 0.218 | 0.322 | 0.249 |
| Multi-ReduNet | 0.919 | 0.929 | 0.945 | 0.891 | 0.939 | 0.923 |
| Multi-ReduNet-LastNorm | **0.984** | **0.973** | **0.967** | **0.962** | **0.978** | **0.962** |

(relative to mnist), the separation and compactness of class-wise features vary substantially across models.

ReduNet (top-left) exhibits notable class entanglement, with overlapping clusters and unclear margins between semantically distinct categories (e.g., classes 0, 3, and 7). The representation remains largely diffuse, reflecting its global coupling across classes and lack of explicit discriminability constraints.

ReduNet-RF (top-right) shows similar structure on ReduNet, indicating that global orthogonality constraints alone are insufficient for resolving subtle visual categories in fashion-mnist.

Multi-ReduNet (bottom-left) introduces sharper decision boundaries and better class separation, thanks to its per-class decomposition strategy. Though some clusters still partially overlap, the overall layout is more class-discriminative and geometrically organized.

Multi-ReduNet-LastNorm (bottom-right) achieves the most clearly separated and compact clusters, with minimal inter-class confusion and high intra-class cohesion. Notably, all 10 classes form distinct, non-overlapping blobs, validating the effectiveness of the final projection step in enforcing orthogonality and enhancing visual interpretability.

Table 14: Accuracy comparison of ReduNet variants on ARCENE

|  | Layers=10 | | | Layers=15 | | |
| --- | --- | --- | --- | --- | --- | --- |
|  | SVM | KNN | NSC | SVM | KNN | NSC |
| ReduNet | 0.536 | 0.512 | 0.488 | 0.439 | 0.414 | 0.439 |
| ReduNet-RF | **0.882** | **0.890** | 0.439 | 0.098 | 0.098 | 0.439 |
| imp-ReduNet | 0.536 | 0.512 | 0.488 | 0.439 | 0.418 | 0.439 |
| imp-ReduNet-RF | **0.882** | **0.890** | 0.439 | 0.098 | 0.098 | 0.439 |
| Multi-ReduNet | 0.756 | 0.707 | 0.805 | 0.634 | 0.708 | **0.805** |
| Multi-ReduNet-LastNorm | 0.785 | 0.759 | **0.817** | 0.752 | 0.734 | 0.805 |
|  | Layers=20 | | | Layers=25 | | |
|  | SVM | KNN | NSC | SVM | KNN | NSC |
| ReduNet | 0.560 | 0.487 | 0.414 | 0.439 | 0.390 | 0.463 |
| ReduNet-RF | 0.901 | **0.890** | 0.437 | 0.098 | 0.098 | 0.439 |
| imp-ReduNet | 0.560 | 0.487 | 0.414 | 0.439 | 0.390 | 0.463 |
| imp-ReduNet-RF | **0.902** | 0.888 | 0.437 | 0.098 | 0.098 | 0.439 |
| Multi-ReduNet | 0.599 | 0.707 | 0.783 | 0.570 | **0.707** | 0.781 |
| Multi-ReduNet-LastNorm | 0.712 | 0.708 | **0.805** | 0.651 | 0.685 | **0.786** |

Figure 6 shows the t-SNE projections of test-time features on the Swarm Behaviour dataset, a binary classification task characterized by limited samples and subtle class variation.

ReduNet (top-left) fails to effectively separate the two classes in the projected feature space. Most samples are scattered and interleaved, indicating weak class-discriminative structure. This reflects

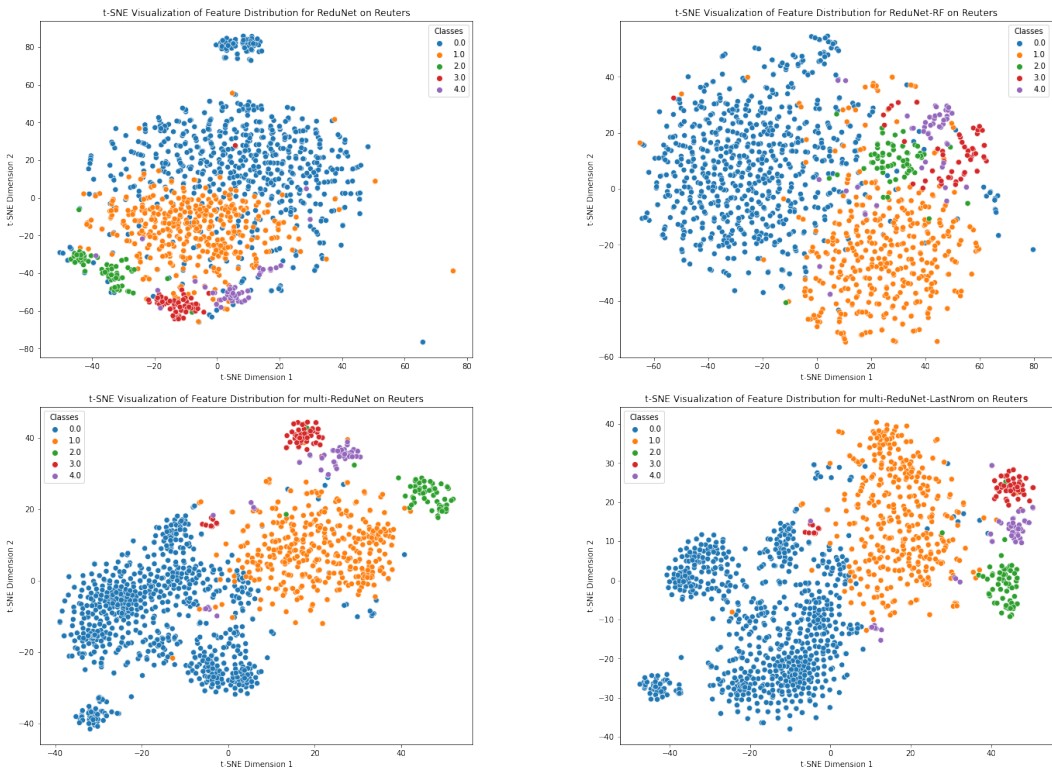

Figure 3: t-SNE visualizations of learned features on Reuters

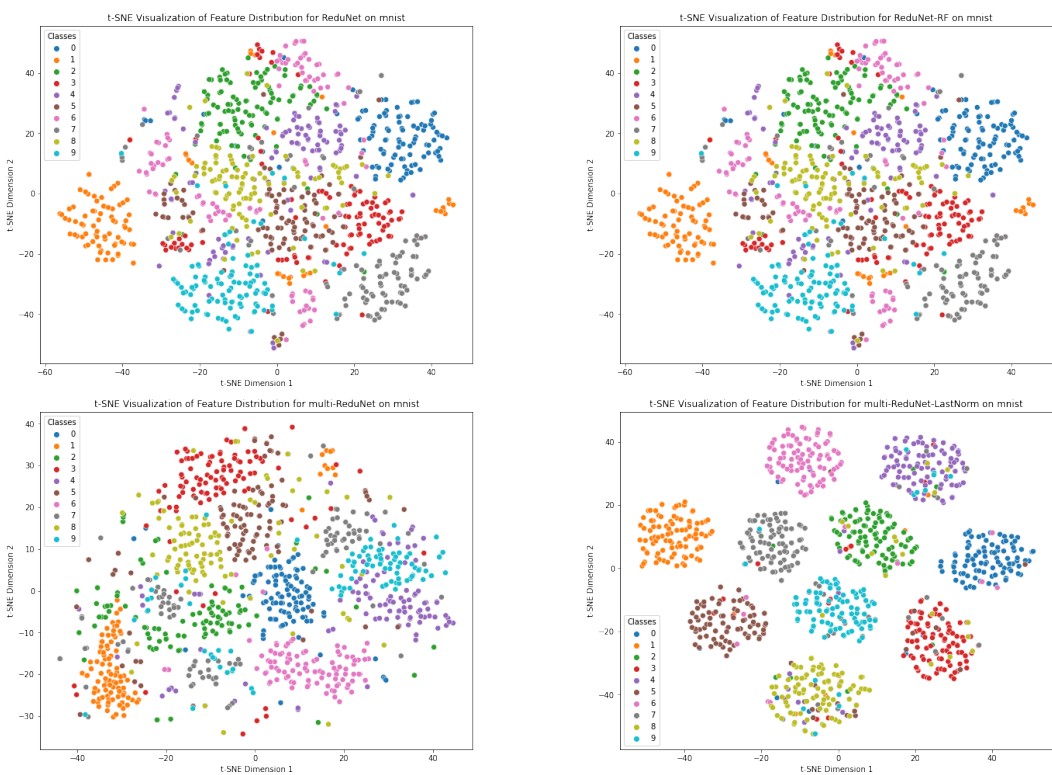

Figure 4: t-SNE visualizations of learned features on mnist

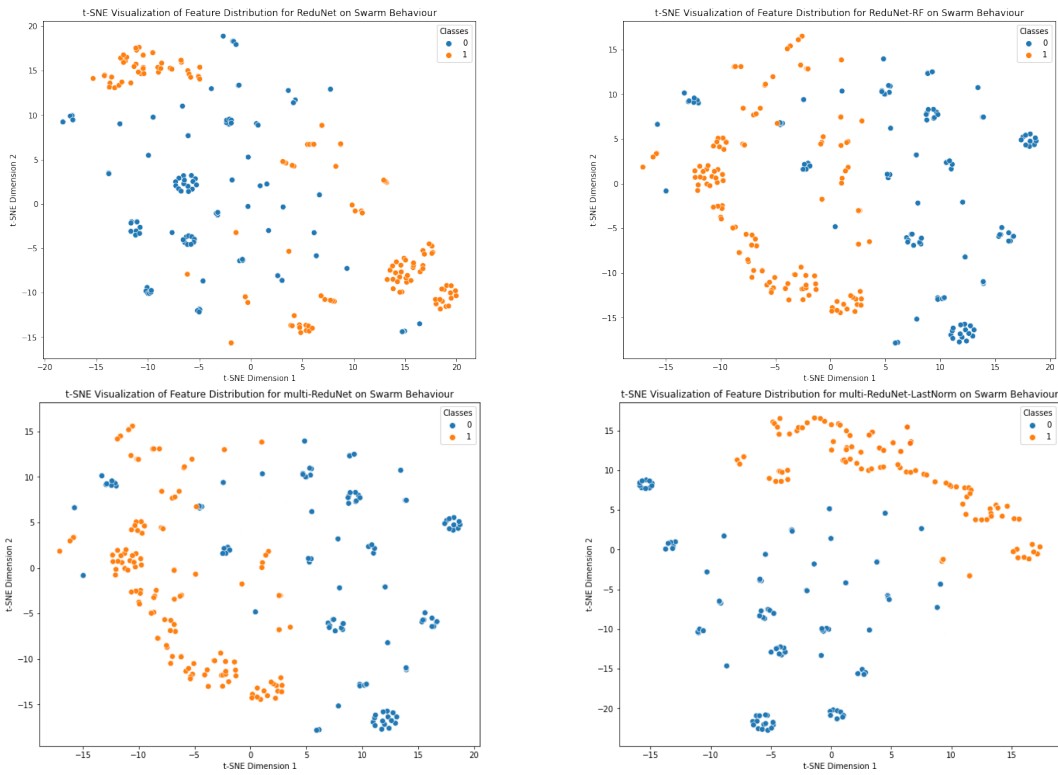

Figure 6: t-SNE visualizations of learned features on Swarm Behaviour

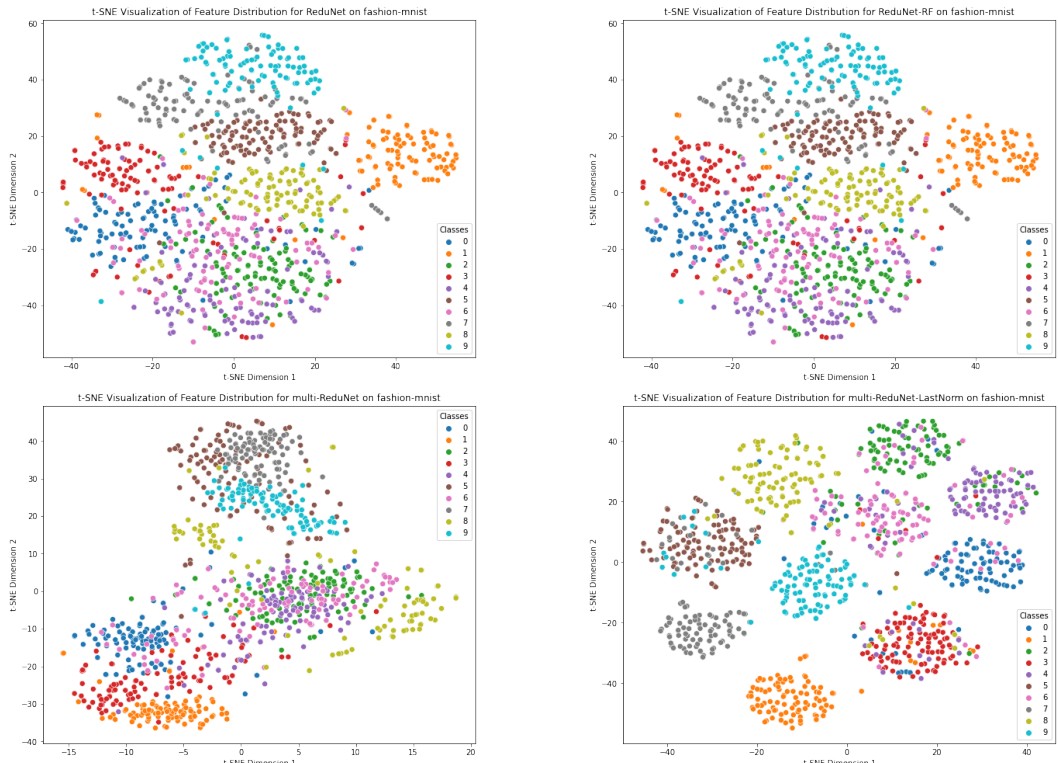

Figure 5: t-SNE visualizations of learned features on fashion-mnist

the challenge of capturing meaningful boundaries when using globally coupled updates without class-specific refinement.

ReduNet-RF (top-right) shows slight improvement, with some local grouping of class-1 (orange) points, though the global overlap remains significant. The marginal gain suggests that random feature projections alone are insufficient to resolve this low-data regime.

Multi-ReduNet (bottom-left) introduces clearer inter-class margins and tighter intra-class clusters. While the separation is not perfect, distinct grouping patterns emerge, indicating that independent class-wise subspace optimization provides meaningful gains in geometric regularity and class alignment.

Multi-ReduNet-LastNorm (bottom-right) delivers the clearest boundary among all variants. Class 0 (blue) and class 1 (orange) form nearly disjoint clusters along a horizontal axis, with minimal cross-class confusion. The use of a final projection to enforce global separation yields a feature space that is highly linearly separable, well-suited for downstream nonparametric classifiers like SVM or NSC.

