# OpenReview forum: "Multi-ReduNet: Interpretable Class-Wise Decomposition of ReduNet"
_ICLR.cc/2026/Conference — ICLR 2026 Poster_

### Official Review · Reviewer_bnBb · 2025-10-24

**Soundness:** 2
**Presentation:** 1
**Contribution:** 1
**Rating:** 2
**Confidence:** 3

**Summary:**

This paper improves the objective of ReduNet under the assumption of “undersampled regimes” and proposes two extensions, Multi-ReduNet and Multi-ReduNetLastNorm, for computational efficiency and representation separability

**Strengths:**

1. The paper is well-structured.
2. The mathematics is generally written in a professional way.

**Weaknesses:**

Major:

1. The motivation is quite weak. In the introduction, the authors claim representation learning in the “undersampled regimes” is challenging, but without citing any literature or explaining whether this is a widely-accepted concern.

2. Similarly, the questions to be tackled are also unclear. The authors do not define and thoroughly explain “class-specific structures” which they highlight as missing components in previous works.

3. The contributions seem to be incremental and trivial. This paper only performs slight improvements on ReduNet under niche and small-scale settings. It does not bring in novel tools or appealing theoretical insights other than decomposing ReduNet’s objectives, either. In fact, the authors said in line 255:  “Although the class-orthogonality property of MCR2 optima has been established in prior work (Chan et al., 2021), our proof leverages a simpler and more streamlined argument.”, meaning the findings have already been established. These all make me question its significance.

4. The technical soundness is also questionable. The gradients derived in lines 274 and 277 are basically the same, subject to different scaling. I hardly believe they have a significant functional difference, which makes the decomposition in line 269 less compelling.

Minor:

1. No experimental details and discussion on limitations.

2. Figures are quite hard to interpret immediately due to font size and colorization.

**Questions:**

1. I’m confused about the last two sentences from lines 32 to 35. Why scenarios where the number of features exceeds the number of samples will lead to overfitting and unstable generalization. Is there literature supporting this claim? The explanations of the background are missing.

2. In Theorem 1, why does $Z^i( Z^j )^T= 0$ mean class-orthogonality? This seems to be misaligned with line 172. Should it be $( Z^i )^T Z^j= 0$?

3. How does the last equality hold in Eq.(2) under the assumption $(Z^{*j_1})^TZ^{*j_2} \neq 0$?

4. The whole analysis in the paper assumes $m \ll d$. Is this assumption even practical for most tasks? I won’t buy it if it’s just an idealistic setting, and rare in practical scenarios.

5. Why compare to variants with a random forest classifier?

6. How much does training time improve numerically? From figure 1, the proposed model is almost the same with ReduNet.

---

> ### Comment · Reviewer_bnBb · 2025-11-28
>
> Thank you for the response. Most of my concerns have been addressed, except for some critical issues.
>
> After reevaluating your proof of Theorem 2 in Appendix D.3, which is the core argument of this paper, here are my further comments about the technical soundness:
>
> 1. Why direction 1 ($v_2 \leq v_1$) holds? The optimal {$ Z^{*j}$} $(j=1,\dots,K)$ for the class-wise subproblem, as you stated in the next paragraph from line 1095, indicates the opposite ($v_1 \leq v_2$). What is missing here?
> 2. The authors said they streamlined the proof of Theorem 1, which is already known in ReduNet (Chan et al, 2021), by introducing a technique, which is Corollary 1, and this technique can make it straightforward to derive Theorem 2. However, as in my question above, only some part of the proof used this technique, but the rest seems flawed.

---

> > ### Author Response · Authors · 2025-11-28
> > **Reply to follow-up on Theorem 2 (decomposition equivalence)**
> >
> > We sincerely thank the reviewer for the careful re-examination of **Theorem 2** and for raising these two important questions. We agree that the previous presentation in Appendix D.3 was too compressed and omitted several justification steps. We have now revised Appendix D.3 to spell out the arguments for **both directions**.
> >
> > Let $v_1$ denote the optimal value of the global MCR$^2$ objective, and $v_2$ the sum of the optimal values of the $K$ class-wise problems.
> >
> > ### 1. Why does Direction 1 ($v_2\leq v_1$) hold?
> >
> > The inequality $v_2\leq v_1$ follows from the fact that an optimal solution of the class-wise problems can be used to construct a **feasible point** for the global problem.
> >
> > Let {$Z^{\prime j}$}$_{j=1}^K$ be optimal solutions of the $K$ class-wise objectives. We can concatenate them into
> >
> > $$
> > Z^\prime :=[Z^{\prime 1},\dots,Z^{\prime K}]\in\mathbb{R}^{d\times m},
> > $$
> >
> > so that the Frobenius constraints $\lVert Z^{\prime j}\rVert_F^2=m_j$ imply that $Z^\prime$ is feasible for the global MCR$^2$ objective.
> >
> > Under the orthogonality assumption in Theorem 4, we can apply Corollary 1in the equality case with $A_j=\frac{d}{m\epsilon^2}Z^{\prime j}(Z^{\prime j})^\top$, then obtain
> >
> > $$
> > \log\det(I+\frac{d}{m\epsilon^2}Z^\prime Z^{\prime \top})=\sum_{j=1}^K \log\det(I+\frac{d}{m_j\epsilon^2}Z^{\prime j}(Z^{\prime j})^\top),
> > $$
> >
> > so that the value of the **global** objective at $Z^\prime$ is exactly
> >
> > $$
> > \Delta R(Z^\prime)=\sum_{j=1}^K \Delta R_j(Z^{\prime j})=v_2,
> > $$
> >
> > where $\Delta R$ denotes the global MCR$^2$ objective, and $\Delta R_j$ denotes the $j$-th class-wise objective. Since $Z^\prime$ is a feasible point for the global problem, we have
> >
> > $$
> > v_2=\Delta R(Z^\prime)\leq \max_Z \Delta R(Z)=v_1.
> > $$
> >
> > These intermediate steps are now written out explicitly in the revised Appendix D.3.
> >
> > ### 2. Clarification of Direction 2 ($v_1\leq v_2$).
> >
> > For the reverse inequality, let $Z^\star$  be a global maximizer of the MCR$^2$ objective. By Theorem 1, $Z^\star$ satisfies class-orthogonality, i.e.
> >
> > $$
> > (Z^{\star i})^\top Z^{\star j}=0 ~~\text{for all }i\neq j,
> > $$
> >
> > and each class $j$ block $Z^{\star j}$ satisfies the Frobenius constraint $\lVert Z^{\star j}\lVert_F^2=m_j$. Hence {$Z^{\star j}$}$_{j=1}^K$ is **feasible** for the class-wise problems.
> >
> > Applying Corollary 1 with $A_j=\frac{d}{m\epsilon^2}Z^{\star j}(Z^{\star j})^\top$, and using the equality condition again yields
> >
> > $$
> > \log\det(I+\frac{d}{m\epsilon^2}Z^\star Z^{\star \top})=\sum_{j=1}^K \log\det(I+\frac{d}{m\epsilon^2}Z^{\star j}(Z^{\star j})^\top).
> > $$
> >
> > Substituting this into the expression of the global objective shows that
> >
> > $$
> > v_1=\Delta R(Z^\star)=\sum_{j=1}^K \Delta R_j(Z^{\star j}) \leq \sum_{j=1}^K \max_{Z^j}\Delta R_j(Z^j)=v_2.
> > $$
> >
> > The revised Appendix D.3 now includes these steps explicitly, together with pointers to the exact equations.
> >
> > ### 3. Relation to Corollary 1 and “streamlining” Theorem 2
> >
> > As this clarification shows, **both directions** of Theorem 2 rely crucially on Corollary 1: in Direction 1 we use the equality case to evaluate the global objective at the concatenated class-wise optima, and in Direction 2 we again use the equality case at the global optimum $Z^\star$. The remaining steps are standard feasibility/optimality arguments (constructing feasible points and comparing them with the respective optima). We have highlighted these usages explicitly in the revised Appendix D.3 to make clear in what sense Corollary 1 streamlines the proof of Theorem 2.
> >
> > We thank the reviewer again for this very helpful feedback.

---

### Official Review · Reviewer_KGJT · 2025-10-28

**Soundness:** 3
**Presentation:** 1
**Contribution:** 2
**Rating:** 2
**Confidence:** 3

**Summary:**

This paper extends ReduNet — a theoretically grounded, interpretable architecture based on the principle of maximal coding rate reduction — to address its limitations in scalability and class-specific representation. The authors propose Multi-ReduNet and Multi-ReduNet-LastNorm, which decompose the global learning objective into class-wise subproblems. This decomposition maintains ReduNet’s theoretical interpretability while substantially improving computational efficiency by lowering matrix inversion costs. Moreover, it enhances feature separability, making the model more effective in undersampled  regimes. The paper provides theoretical justification for this class-wise formulation and demonstrates empirically, across multiple datasets, that the proposed models preserve interpretability and achieve better efficiency and discriminative power.

**Strengths:**

1.	The empirical results align well with the theoretical analysis, and Multi-ReduNet demonstrates significant performance improvements in the undersampled regime.
2.	The writing is clear, well-structured, and easy to follow.
3.	The theoretical analysis seems rigorous, and I did not find errors in the proofs.

**Weaknesses:**

1.	The experiments in this paper are primarily conducted on toy datasets such as MNIST. I believe it is necessary to include experiments on more realistic datasets, such as CIFAR. In addition, I am curious about the performance of Multi-ReduNet in the oversampled regime — is it still competitive under such conditions?
2.	The motivation behind Multi-ReduNet is clear and intuitive. However, the rationale for introducing Multi-ReduNet-LastNorm is somewhat unclear, as there is no theoretical comparison between the two. It would strengthen the paper to include a clearer discussion on the logical progression from Multi-ReduNet to Multi-ReduNet-LastNorm.
3.	I noticed that the main proofs are presented in the main text. This makes the paper rather math-heavy and potentially difficult to follow. It might be preferable to include simplified versions of the proofs in the main paper and move the detailed derivations to the appendix.
4.	In Figure 2, the representations learned by ReduNet appear well-clustered. However, Table 2 reports relatively low classification accuracy for the same model. Did I miss something here? It would be helpful to clarify the reason behind this apparent discrepancy between the results.
5.	The authors state that “Although the class-orthogonality property of MCR² optima has been established in prior work (Chan et al., 2021), our proof leverages a simpler and more streamlined argument.” I view this as one of the main theoretical contributions of the paper. However, it remains unclear in what sense the proof is simpler. A more detailed explanation or discussion would strengthen this claim.
6.	The presentation quality of the paper could be improved. For example, the title in Figure 2 is too small and difficult to read. Moreover, if the detailed proofs are moved to the appendix, it would be beneficial to include more discussion or intuitive explanations in the main text to improve readability and accessibility.
7.	In summary, this paper presents a theoretically grounded and interpretable extension of ReduNet with promising results. However, the work could be significantly strengthened through better presentation, clearer motivation for the proposed variants, and more comprehensive experiments on realistic datasets.

**Questions:**

See Weaknesses.

---

### Official Review · Reviewer_tTgf · 2025-10-28

**Soundness:** 3
**Presentation:** 3
**Contribution:** 3
**Rating:** 6
**Confidence:** 2

**Summary:**

This paper proposes two class-wise extensions to ReduNet, namely, Multi-ReduNet and Multi-ReduNet-LastNor, to improve the performance under undersampled, high-dimensional conditions. The authors theoretically justify that the global MCR² objective can be decomposed into class-specific subproblems and leverage this result to design more efficient and class-discriminative models. Experiments on multiple datasets show consistent improvements in classification accuracy and training efficiency, while maintaining interpretability.

**Strengths:**

1. The paper gives a theoretical analysis showing that the MCR² objective under certain conditions can be equivalently decomposed into class-wise subproblems.
2. The proposed models maintain the white-box property of ReduNet and preserve its forward-only optimization strategy. This makes the approach more transparent and easier to analyze compared to conventional backpropagation-based deep networks.
3. The class-wise decomposition reduces the cost of matrix inversion from high-dimensional global matrices to smaller class-specific ones, which is computationally advantageous in settings with limited data and large feature dimensionality.

**Weaknesses:**

1. The method assumes that the class-wise decomposition is meaningful under undersampled regimes. However, the paper does not investigate how the approach behaves when this assumption is less valid, e.g., when the sample size is moderately large.
2. The evaluation focuses on relatively simple or small-scale datasets (e.g., MNIST, Fashion-MNIST), which may limit the conclusions.
3. The experimental comparisons are restricted to ReduNet and its variants. The paper could be strengthened by comparing against a broader set of interpretable or class-structured learning methods, to better situate the approach within the existing literature.
4. Although Multi-ReduNet-LastNorm performs slightly better in most cases, the role of the final-layer-only normalization is not fully analyzed. It would be helpful to understand under what circumstances this variant is preferable.

**Questions:**

Please refer to weaknesses.

---

### Official Review · Reviewer_d7rQ · 2025-10-29

**Soundness:** 3
**Presentation:** 3
**Contribution:** 4
**Rating:** 6
**Confidence:** 3

**Summary:**

The authors improve the scalability of the classifier ReduNet, which is specialized for solving tasks where there are a lot more features $d$ than samples $m$. Given $K$ classes and $m_j$ samples for class $j$, new algorithms scales as $\mathcal{O}(\sum_{j=1}^Km_j^3)$ compared to the $\mathcal{O}(Kd^3)$ of the original algorithm. The scalability and performance is demonstrated on 4 real datasets.

**Strengths:**

The paper does a great job at explaining the ReduNet method to readers without prior knowledge on the field.

The theoretical contribution of the paper is significant since the new algorithm scales as $\mathcal{O}(\sum_{j=1}^Km_j^3)$ compared to the original $\mathcal{O}(Kd^3)$.

The improvements in accuracy are also significant compared to prior work.

**Weaknesses:**

## Orthogonality Constraints

At line 270 it is stated that the optimization problem outlined at line 269 is solved "subject to class-wise orthogonality and norm constraints".
I don't think the manuscript clarifies how these constraints are applied in practice. The proposed algorithm iteratively solves the equation at line 269 in order to infer the representation $Z_j$ of each class. The $Z_j$ are updated independently by maximizing a separate objective so it is not clear how orthogonality is enforced.

## A smoother-introduction to Multi-ReduNet

The paper actually has two contributions : Imp-ReduNet which changes the inversion of a $d\times d$ matrix to a $m\times m$, and
Multi-Redunet which furthers improves this to $K$ inversions of $m_j\times m_j$ matrices. But the order in which these contributions are presented in confusing : multi-redunet is presented first and then imp-redunet. I think that describing imp-redunet first would help motivate the need to separate the objective into $K$ objectives.

By introducing imp-redunet first (via lemma 1) it is clear to the reader that the computational bottleneck is inversion of a $m\times m$ matrix to compute the gradient of $\text{log det}(I+\alpha Z Z^T)$. If we were able to replace this with $K$ terms $\text{log det}(I+\alpha Z_j Z_j^T)$, we could inverse $K$ matrices of shape $m_j\times m_j$ instead, which is a lot better. After this initial high-level motivation, and hinting that $Z Z^T=\sum_{j=1}^K Z_j Z_j^T)$, the main theorems can be presented.

This is a subtle change, but I think it will improve the flow of the paper significantly.

## Technical Overload

On a similar topic, the paper would benefit from moving some technical content to the appendix and focusing more on high-level ideas in the main manuscript. Notably, the proof of the Theorem could be moved in the appendix, and replaced with a high-level proof description.
This description would only need to accentuate the most crucial parts of the proof e.g. that $\text{log det} (I+\alpha Z Z^T)= \text{log det} (I+\alpha \sum_{j=1}^K Z_j Z_j^T) =  \sum_{j=1}^K \text{log det} (I+\alpha Z_j Z_j^T)$ whenever class representations are orthogonal.
This clarifies to the reader that class-orthogonality is the key assumption to separate the objective into $K$ sub-objectives.

The freed space in the main manuscript could be used to introduce intuition for ReduNet e.g. extended content from the Appendix B.

## Figures 1

This is a very minor point, but I think that Figure 1 is hard to read. It is hard to see the color of the markers because it is very dark. Also, the marker color does not match with the line, so I constantly have to read the legend to be sure. Moreover, methods based on RF perform so bad that they hide the differences between ReduNet and Multi-Redunet (the main contributions of the paper). I would suggest removing the RF methods from the plot and simply indicate in the text that they take orders of magnitude more time.

**Questions:**

What is the $d_j$ in theorem 1? It is not introduced before being used.

Theorem 1 defines class-orthogonality as $Z_j Z_i^T=0$. Shouldn't is be $Z_j^T Z_i=0$ instead following Corollary 1?

Why is the Frobenius norm $||Z_j||^2_F\leq m_j$ bounded in theorem 1 but the optimization algorithm projects to the unit sphere? Projecting $m_j$ points on the unit sphere guarantees that $||Z_j||^2\leq m_j$ , but the converse is not true. Perhaps a more in-depth introduction to ReduNet in the main manuscript would help.

From my understanding, multi-redunet an improvement in terms of scalability : it solves the same problem as ReduNet but more efficiently. Then how can multi-redunet perform better than redunet in terms of accuracy? Is ReduNet reaching a different optimum?

---

### Author Response · Authors · 2025-11-26
**Author response and summary of changes**

We thank the action editor and all reviewers for their detailed feedback. Our paper studies white-box representation learning in **undersampled regimes** (number of samples $m \ll$ dimension $d$) by introducing two **class-wise extensions of ReduNet**, Multi-ReduNet and Multi-ReduNet-LastNorm, that decompose the global MCR$^2$ objective into class-wise subproblems, reducing matrix-inversion cost while preserving interpretability.

In response to the reviews, we (i) clarified the motivation, scope, and notion of class-specific structure; (ii) reorganized and streamlined the method section and proofs; (iii) broadened the empirical evaluation with additional real-world datasets, baselines, and robustness analyses; and (iv) improved figures and experimental details. Below we summarize the main changes in the revised manuscript and then respond to each reviewer in turn.

We first summarize the concrete changes in the revised manuscript, and then respond to each reviewer in turn. References to sections and appendices are with respect to the revised version.

## 1. Summary of changes

We have revised the manuscript substantially in response to the reviewers' feedback:

- **Clarified setting and key concepts.** We now explicitly define undersampled regimes and discuss why $m \ll d$ poses specific challenges, adding appropriate references. We also provide a precise definition of class-specific structures (in Section 2. Related Work and Section 3.4 Multi-ReduNet-LastNorm) to clarify what is missing in prior approaches and how our models explicitly exploit this structure.

- **Reorganized and streamlined the method section.** Section 3. Proposed Method is rewritten to follow a clearer progression: ReduNet preliminaries → computationally improved **imp-ReduNet** → class-wise **Multi-ReduNet** → normalization-relaxed **Multi-ReduNet-LastNorm**. This ordering improves readability and makes the design choices easier to follow. We also simplified the main proofs, moving technical details to the appendix to better balance mathematical rigor and accessibility, and **added an explicit discussion of the motivation, role, and recommended usage regimes of Multi-ReduNet-LastNorm (Section 3.4 and Appendix A.5).**

- **Expanded empirical evaluation.** We added experiments on additional real-world datasets (DrivFace, ARCENE, CIFAR-10), compared against broader baselines (PCA+SVM, class-wise PCA+NSC, LDA), and included an ablation/robustness analysis for LastNorm. We further investigate performance in **oversampled / well-sampled regimes** and use these results to discuss the applicability and failure modes of Multi-ReduNet(-LastNorm). **Due to space limits, these analyses, including the CIFAR-10 study and the "where Multi-ReduNet excels / does not improve" discussion, are consolidated in Appendix A (Scope and Limitations, especially A.5–A.6).**

- **Improved figures and focus.** We updated figures with clearer labels and legends, and removed RF-based variants from the training-time plots to keep the visual comparisons focused on ReduNet versus our class-wise extensions.

We believe these changes directly address the reviewers' main concerns and significantly strengthen the paper.

To make the mapping between reviewer feedback and revisions explicit, we summarize the main points and our responses in the following table.

---

### Author Response · Authors · 2025-11-26
**Author response and summary of changes**

## 2. Response Table

| Reviewer & Comment ID | Concern / Question | Our main response & changes | Where in revised paper |
|-----------|----------------------------|-----------------------------|------------------------|
| d7rQ - Weakness1 | How is class orthogonality enforced in practice? | Clarified that orthogonality is **only a property of the global optimum**, not an algorithmic constraint; Algorithm 1 enforces only $\ell_2$-norm projection. | Sec. 3.3 (Theorem 1–2 discussion) |
| d7rQ - Weakness2&3 | Section 3 is hard to follow; too many details up front | Reordered method section: **ReduNet → imp-ReduNet → Multi-ReduNet → Multi-ReduNet-LastNorm**; moved long proofs to appendix, kept only proof sketches and intuition in main text. | Sec. 3.1–3.4, Apps. C–E |
| d7rQ - Weakness4 | Figure 1 cluttered; RF curves hide main comparison | Removed RF variants from Figure 1, used more distinct colors and line styles, kept RF only as numeric results in appendix with a short comment. | Fig. 1, Apps. B & I.1 |
| tTgf - Weakness1; KGJT - Weakness1 | What happens when undersampled assumption fails? | Added experiments on **oversampled tabular datasets** (Iris, Mice Protein); showed Multi-ReduNet does **not** help there and positioned them as failure modes. | App. A.6 (Failure model analysis) |
| tTgf - Weakness2&3 | Datasets and baselines are limited | Added real-world undersampled datasets **DrivFace, ARCENE** and classical baselines (**PCA+SVM, class-wise PCA+NSC, LDA**); showed where Multi-ReduNet(-LN) is best/competitive. | Tab. 1, Tab. 3–5, Sec. 4 |
| tTgf - Weakness4; KGJT - Weakness2 | Role & benefit of Multi-ReduNet-LastNorm unclear | Explained LastNorm as **final-layer-only normalization** aimed at numerical stability; added robustness/ablation table showing ~10$\times$ smaller $\eta$-sensitivity and small but consistent accuracy gains. | Sec. 3.4, Sec. 4.3, Tab. 5, App. E.3 |
| KGJT - Weakness1 | “Toy datasets”; need realistic benchmarks & CIFAR | Added **DrivFace** and **ARCENE** (safety-critical vision & medical proteomics) with large gains in accuracy, robustness and efficiency; ran **CIFAR-10** as a **failure case** to show limits of vectorized MCR$^2$. | Tab. 1, Tab. 3-5, Tab. 6, Sec. 4, App. A.6 |
| KGJT - Weakness3 | Proofs too long; math heavy | Moved full proofs of Lemma 1, Thm. 1–2, Cor. 1 to appendix; main text now has short plain-language explanations + 2–3 line proof sketches and “takeaway” sentences. | Sec. 3, Apps. C–D |
| KGJT - Weakness4 | Figure 2 vs. Table 2 seem inconsistent; labels tiny | Results are consistent, the confusion came from weak labeling and small text; added explicit **column labels**, clearer caption, explanatory paragraph | Fig. 2, Sec. 4.2, App. F (Tab. 8) |
| KGJT - Weakness5 | “Simpler proof” claim not justified | Added **Appendix D.4** comparing our log-det inequality proof with Chan et al. (2021); highlighted 2-step determinant-based argument, geometric intuition, and reuse for other log-det objectives. | Sec. 3.3, App. D.4 |
| bnBb - Major weakness1&Question1; bnBb - Question4| Motivation for undersampled regimes & $m \ll d$ overfitting | Added literature citations (ESL Ch.18; Bühlmann & van de Geer), intuitive explanation why undersampled problems are hard, and softened wording (“tends to exacerbate”). Connected to $m_{\text{train}}/d$ ratios of our datasets. | Sec. 1, Sec. 2, Sec. 4 |
| bnBb - Major weakness2 | “Class-specific structure” vague | Gave explicit definition (label-conditioned subspaces $span(Z^j)$); contrasted **global covariance** in ReduNet vs **class-wise covariances** and decoupled updates in Multi-ReduNet. | Sec. 2, Sec. 3.4, App. E.3 |
| bnBb - Major weakness3 | Possible incremental contribution | Emphasized three points: (1) **new class-wise MCR$^2$ formulation and algorithms** (Multi-ReduNet / LastNorm) with $\mathcal{O}(\sum m_j^3)$ inversions; (2) clarified determinant-based proof technique; (3) showed gains on real high-dimensional datasets vs. classical baselines. | Sec. 3.3 |
| bnBb - Major weakness4 | Gradients $E_l^j, C_l^j$ “just scaled copies” | Explained that updates depend on **$(E_l^j − \gamma_j C_l^j)Z_l^j$**, so expansion/compression act as opposing forces from different log-det terms; key novelty is using **class-wise covariances** instead of global $ZZ^\top$. Added explanation after gradeints. | Sec. 3.3 (after eq. for $E_l^j, C_l^j$) |
| bnBb - Question6 | Numerical speedup unclear from Fig. 1 | Added wall-clock timing table showing **1.4–2.6$\times$** training speedups (≈2$\times$ on average) for Multi-ReduNet-LastNorm vs. ReduNet across datasets and depths; commented on scaling with depth. | Sec. 4.4, App. B (Tab. 7) |

---

### Author Response · Authors · 2025-11-26
**Author response and summary of changes**

## 3. Response to Reviewer d7rQ

We thank reviewer d7rQ for the positive overall assessment and for the concrete suggestions on clarifying the role of orthogonality, improving the exposition of Section 3, and making the figures easier to read.

**Weakness 1. Orthogonality constraints.**
How is class-orthogonality actually enforced in the optimization, given that the algorithm updates each class independently?

**Response.** We apologize for the confusing wording "subject to class-wise orthogonality and norm constraints." In the **implemented algorithm**, we only enforce the **norm** constraint via $\ell_2$–normalization of each feature vector (projection onto the unit sphere). We do **not** impose class-wise orthogonality $(Z^i)^\top Z^j = 0$ as an explicit constraint during training, neither in ReduNet nor in our extensions.

Class-orthogonality appears only at the **theoretical level**. Theorem 1 shows that any global maximizer $Z$ of the MCR$^2$ objective must satisfy
$$
(Z^i)^\top Z^j = 0 \quad \text{for all } i \neq j,
$$
so orthogonality is a **property of the global optimum**, not a constraint we explicitly enforce. Theorem 2 then leverages this optimal class-orthogonal structure to prove that the global objective is equivalent to a sum of $K$ class-wise objectives, which motivates our decomposed formulation. In practice, Algorithm 1 simply optimizes these per-class objectives without additional constraints, leading to more separated class representations (Fig. 2) and higher test accuracy (Table 3), but not perfectly orthogonal features. We have revised the text around Theorem 2 in Section 3.3 to make this distinction between the theoretical optimality condition and the practical training procedure explicit.

**Weakness 2&3. Smoother introduction to Multi-ReduNet / technical overload.**

The reviewer notes that the paper has two main contributions (imp-ReduNet and Multi-ReduNet) and suggests introducing imp-ReduNet first to motivate the need for $K$ class-wise objectives, and moving some technical content to the appendix to focus the main text on high-level ideas.

**Response.** We agree and have reorganized Section 3 to improve the logical flow and reduce technical overload. In **Section 3.2 (Imp-ReduNet)** we now first give a high-level explanation of how Lemma 1 replaces the inversion of a $d \times d$ matrix by an $m \times m$ one, and why this is the main computational bottleneck in ReduNet. The detailed algebraic derivation has been moved to Appendix C.

In **Section 3.3 (Multi-ReduNet)** we then motivate the decomposition into $K$ class-wise subproblems, each of which can be solved by inverting only matrices of size $m_j \times m_j$ instead of $d \times d$, and we present only proof sketches for Theorem 1 (class-orthogonality), Theorem 2 (equivalence of MCR$^2$ and $K$ class-wise optimizations), and the resulting gradients. Full proofs and technical details are now in Appendices D and E.

Finally, **Section 3.4 (Multi-ReduNet-LastNorm)** describes the LastNorm variant and explains the motivation for enforcing normalization only at the final layer.

This reordering makes the transition "ReduNet → imp-ReduNet → Multi-ReduNet → Multi-ReduNet-LastNorm" more natural and keeps the main text focused on the key ideas (complexity reduction and class-wise decomposition), while readers interested in full derivations can consult the appendices.

**Weakness 4. Figure 1 readability and RF curves.**

Figure 1 is hard to read, and RF methods perform so poorly that they hide the differences between ReduNet and Multi-Redunet. I would suggest removing the RF methods from the plot and simply indicating in the text that they take orders of magnitude more time.

**Response.** We agree that the original Figure 1 was cluttered and that RF-based variants distracted from the main comparison. In the revised manuscript, we (i) remove all RF-based methods from Figure 1, (ii) increase marker and font sizes in Figure 1, and (iii) use more distinct colors and line styles to improve readability. The RF variants (ReduNet-RF, imp-ReduNet-RF, Multi-ReduNet-RF) are moved to Appendix I.1, together with a brief comment that they neither improve accuracy nor efficiency over other MCR$^2$-based variants and are not central to our contributions. In addition, the main text now explicitly states that the exact numerical values of the training times are reported in Appendix B.

---

> ### Comment · Reviewer_d7rQ · 2025-11-26
> **Reviewer Response**
>
> Thank you very much for taking the time to addressing my concerns and updating the manuscript.
>
> - **W1** I now understand that class-wise orthogonality is only guaranteed at the optimum and that the constraint is not applied in the algorithm. The updated manuscript now clarifies that orthogonality is only used to justify a new parametrization (multi-ReduNet) of the same optimization problem (ReduNet).
> - **W2** The manuscript has been modified to smooth the introduction to Multi-ReduNet. The flow of the paper improved greatly by applying the recommended changes.
> - **W4** Figure 1 is now more readable and clearly highlights the run-time improvements of Multi-ReduNet.

---

### Author Response · Authors · 2025-11-26
**Author response and summary of changes**

## Response to Reviewer d7rQ
**Question 1. What is $d_j$ in Theorem 1?**

Theorem 1 states: global MCR$^2$ objective is subjected to $\mathrm{rank}(Z^j) \le d_j$. What is the $d_j$ in theorem 1? It is not introduced before being used.

**Response.** In Theorem 1, $d_j$ denotes an **upper bound on the rank** of the class-wise feature matrix $Z^j$. In the revised manuscript we now state this assumption explicitly as
$$
\mathrm{rank}(Z^j) \le d_j, \quad \sum_{j=1}^K d_j \le d.
$$
Intuitively, these bounds ensure that the class-specific subspaces can be embedded into $\mathbb{R}^d$ without overlap, which is what guarantees the existence of a class-orthogonal optimal configuration.

In the undersampled regimes we study ($m \ll d$), a natural choice is $d_j = m_j$, so that $\mathrm{rank}(Z^j) \le m_j$ and $\sum_{j=1}^K d_j = \sum_{j=1}^K m_j = m < d$ hold automatically, making the assumption mild in our setting. We have updated the statement of Theorem 1 to clarify $d_j$ and its role.

**Question 2. Should the orthogonality condition be $Z_j^\top Z_i = 0$ instead of $Z_j Z_i^\top = 0$?**

Theorem 1 defines class-orthogonality as $Z_j Z_i^\top = 0$. Should it not be $Z_j^\top Z_i = 0$?

**Response.** Yes, thank you for catching this. The intended class-orthogonality condition is
$$
Z_i^\top Z_j = 0 \quad \text{for } i \neq j,
$$
which expresses that the column spaces of $Z_i$ and $Z_j$ are orthogonal. The notation $Z_j Z_i^\top$ in the original theorem statement was a typo.

We have corrected this notation throughout the paper and added a short remark after Theorem 1 in Section 3.3 clarifying that we use the convention "columns = samples", so $Z_i^\top Z_j$ is the natural cross-class Gram matrix.

**Question 3. Frobenius-norm bound vs unit-sphere projection**

Why is the Frobenius norm $\lVert Z^j \rVert_F^2 \le m_j$ bounded in Theorem 1 but the optimization algorithm projects to the unit sphere?

**Response.** In Theorem 1, the class-wise matrices are constrained via

$$
\lVert Z^j\rVert_F^2 \le m_j.
$$

In the algorithm, we normalize **each sample** to unit $\ell_2$ norm by projecting onto the unit sphere:

$$
\lVert z^j_{(i)}\rVert_2 = 1 \quad \text{for } i = 1,\dots,m_j,
$$

where $z^j_{(i)}$ is the $i$-th column of $Z^j$. Stacking these columns gives

$$
\lVert Z^j\rVert_F^2 = \sum_{i=1}^{m_j} \lVert z^j_{(i)}\rVert_2^2 = m_j.
$$

so the Frobenius-norm constraint in Theorem 1 is exactly satisfied (and saturated) by the per-sample unit-sphere projection used in Algorithm 1.

Without any norm control, the objective of the MCR$^2$ could be increased trivially by scaling $Z$ by a large constant, leading to degenerate solutions that exploit magnitude rather than learning meaningful directions. The unit-sphere projection in Algorithm 1 prevents this scaling degeneracy and forces the optimization to focus on discriminative directions in feature space instead of arbitrarily amplifying feature norms. This normalization mechanism is inherited directly from the original ReduNet formulation, and we now make this connection explicit in Section 3.1 of the revised manuscript.

---

> ### Comment · Reviewer_d7rQ · 2025-11-26
> **Reviewer Response**
>
> I wish to thank the authors for responding to my questions 1-3. I am satisfied to the modifications applied to Theorem 1 (regarding $d$), and with my second question. Still, I am not sure my third question was answered. I understand why the constraint $ \lVert Z^j\rVert_F^2 \le m_j $ is necessary to avoid degenerate solutions for the optimization problem. I also understand that projecting the individual representations $z^j\_{(i)}$ on the unit sphere guarantees that the Frobenius norm is bound. Nevertheless, projecting all class-instances on the unit sphere is a sufficient condition to have $ \lVert Z^j\rVert_F^2 \le m\_j. $, but it is not a **necessary** condition. If $m\_j=2$, one could have $\lVert z^j\_{(1)}\rVert^2=1/2$ and $\lVert z^j\_{(1)\|}\rVert^2=3/2$ and still have $ \lVert Z^j\rVert_F^2 \le 2. $. Then, how can we be sure the optimum will be on the unit sphere?

---

> > ### Author Response · Authors · 2025-11-27
> > **Answer of follow-up question 3**
> >
> > We thank the reviewer for this insightful follow-up question. You are correct that projecting every sample to the unit sphere is a **sufficient** condition to satisfy the Frobenius norm bound, but theoretically not a **necessary** one (e.g., varying norms could still sum to $m_j$​).
> >
> > However, we clarify and justify the design choice of per-sample unit projection ($\lVert z^j_{(i)}\rVert_2=1$) based on the following three considerations:
> >
> > - **Clarification on the Constraint.** First, we would like to clarify a minor detail: In our Theorem 1 (both in the original submission and the revision), the constraint is specified as an **equality** $\lVert Z^j\lVert_F^2=m_j$, rather than an inequality ($\leq$). We apologize for any confusion caused by our earlier response.
> >
> >    Projecting each column to unit norm, $\lVert z^j_{(i)}\rVert_2=1$, guarantees this constraint is satisfied exactly, since
> >    $$\lVert Z^j\lVert_F^2=\sum_{i=1}^{m_j}\lVert z^j_{(i)}\rVert_2^2=m_j.$$
> > - **Consistency between Training and Inference.** This is the most practical reason. In real-world classification scenarios (the inference phase), the class label of a test sample is unknown, and we cannot enforce a batch-level constraint (like $\sum\lVert z_i\rVert_2=m^{\text{test}}_j$) across a batch of test data. To ensure the decision is based purely on the **angular structure** (subspace alignment) rather than magnitude, the natural approach is to normalize test samples to the unit sphere. Consequently, to maintain **consistency between training and testing**, we use the same unit-sphere constraint during training. If we allowed variable norms (e.g., 0.5 and 1.5) during training, simply normalizing them at test time would introduce a distribution shift, potentially degrading performance.
> > - **Do we claim the optimum must lie on the unit sphere?**  No. As you correctly point out, unit-sphere projection is sufficient but not mathematically necessary for the Frobenius constraint. We do **not** claim that every global maximizer of the MCR$^2$ must have $\lVert z_{(i)}\rVert_2=1$ for all $i$. This is precisely one reason why we also study **Multi-ReduNet-LastNorm**, where intermediate layers are not normalized and only the final representation is projected to the unit sphere for comparability among features. Both variants satisfy the theoretical constraint, but they explore different feasible regions.
> >
> >    Empirically (Table 5), Multi-ReduNet and Multi-ReduNet-LastNorm achieve the same best accuracy on 5/6 datasets, with LastNorm slightly better on Fashion-MNIST (+1.3 pp) and more robust to the learning rate. This suggests that enforcing unit-sphere projections at every layer is a practical and effective regularization choice, but not the only feasible one, hence our recommendation of LastNorm when robustness and flexibility are important.

---

> > > ### Comment · Reviewer_d7rQ · 2025-11-27
> > > **Thank you**
> > >
> > > Thank you for your very quick response. I understand that projecting on the unit sphere is the most practical way to constrain the Frobenius norm (during training and inference). I think that adding a disclaimer in the paragraph **Rationale for unit-sphere projection** that unit sphere projections are sufficient (but not necessary) will improve the clarity of the work. This is because every theoretical optimization problem presented use a bounded norm constraints, which gives the reader the impression that is the constraint of the actual algorithm.

---

> > > > ### Author Response · Authors · 2025-11-27
> > > > **Clarification on Section 3.1**
> > > >
> > > > Thank you for the helpful suggestion.
> > > >
> > > > We have updated the paragraph “Rationale for unit-sphere projection” in Section 3.1 of the revised manuscript to explicitly state that unit-sphere projection is a **sufficient but not necessary** way to satisfy the class-wise Frobenius-norm constraint in MCR$^2$.
> > > >
> > > > We hope this clarification improves the transparency of the link between the theoretical formulation and the actual algorithm.

---

> > > > > ### Comment · Reviewer_d7rQ · 2025-11-27
> > > > > **Response**
> > > > >
> > > > > I am happy with your response and the current state of the manuscript. I no longer have any concerns about the paper so I have increased my score.

---

### Author Response · Authors · 2025-11-26
**Author response and summary of changes**

## Response to Reviewer d7rQ
**Question 4. If Multi-ReduNet improves scalability but solves the same problem, why can it achieve better accuracy?**

Multi-ReduNet is an improvement in terms of scalability, which makes it more efficient. Then how can Multi-ReduNet perform better than ReduNet in terms of accuracy?

**Response.** At the level of the objective, ReduNet and Multi-ReduNet both optimize the same MCR$^2$ criterion: Theorems 1–2 show that, under the class-orthogonality optimality condition, the global objective can be written as a sum of $K$ class-wise terms. In exact arithmetic and with full convergence, both parametrizations would recover the same class-orthogonal global optimum.

In practice, however, the two implementations **parameterize and solve the objective differently**. Multi-ReduNet uses a class-wise decomposition with per-class inversions, which changes the numerical conditioning of the updates and, in turn, their **robustness to hyperparameters** such as the step size $\eta$. Our original submission (Table 2) used the official ReduNet default hyperparameters ($\eta = 0.5$, $\varepsilon^2 = 0.1$), without validation-based tuning. This choice was deliberate: in the undersampled regimes we target ($m \ll d$), further splitting the already scarce data for validation would exacerbate sample scarcity.

To investigate this question, we conducted a post-submission study over four learning rates $\eta \in$ {0.5, 0.1, 0.05, 0.01}:

- ReduNet’s accuracy can vary substantially as $\eta$ changes;
- Multi-ReduNet and Multi-ReduNet-LastNorm are **more stable** over the same range of $\eta$ (smaller variance and higher mean accuracy);
- with carefully tuned $\eta$, ReduNet can reach a similar peak accuracy, but for a fixed, shared setting (e.g., $\eta=0.05$ used in Table 3), Multi-ReduNet-LastNorm often attains higher accuracy.

Thus, the observed gains do not come from optimizing a different objective or a different "theoretical optimum", but from **improved robustness to hyperparameter choices** in the undersampled regimes we study. This is quantified in the revised robustness analysis, where Multi-ReduNet-LastNorm achieves substantially smaller accuracy variation across learning rates (e.g., 9.8× better hyperparameter robustness than ReduNet; see Section 4.3 and Appendix F for the detailed robustness tables).

---

> ### Comment · Reviewer_d7rQ · 2025-11-26
> **Reviewer Response**
>
> Thank you very much for addressing this concern. The updated manuscript indeed clarifies that ReduNet and Multi-ReduNet are two parametrization of the same problem However, MultiReduNet's parametrization is better conditioned leading to improved accuracy (but not always) and improved robustness to $\eta$.

---

### Author Response · Authors · 2025-11-26
**Author response and summary of changes**

## 4. Response to Reviewer tTgf

We thank reviewer tTgf for the positive assessment of our theoretical contribution and white-box design, and for the constructive suggestions on stress-testing the undersampled assumption, broadening the empirical evaluation with larger and more diverse datasets and baselines, and clarifying when the Multi-ReduNet-LastNorm variant is preferable.

**Weakness 1. How does the approach behave when the undersampled assumption is less valid?**

The method assumes that the class-wise decomposition is meaningful under undersampled regimes. However, the paper does not investigate how the approach behaves when this assumption is less valid.

**Response.** We agree that it is important to understand when the undersampled assumption breaks down. In the revision we therefore add experiments in **well-sampled / oversampled regimes that lie far outside $m/d < 1$**.

Concretely, we include two additional datasets with $m/d \gg 1$: (i) **Iris** (3 classes, $d=4$, $m_{\text{train}}=105$, $m/d=26.25$), and (ii) **Mice Protein Expression** (8 classes, $d=77$, $m_{\text{train}}=756$, $m/d=9.82$). These correspond to **moderately oversampled** (Mice Protein) and **heavily oversampled** (Iris) regimes where the class-wise decomposition assumption is expected to be less valid. We compare ReduNet, Multi-ReduNet, and Multi-ReduNet-LastNorm using the same hyperparameters as in the undersampled experiments.

The results (Table 6 in Appendix A.6) show a clear pattern:

- On heavily oversampled Iris, all three variants achieve very similar accuracy, with ReduNet slightly better on average.
- On moderately oversampled Mice Protein, ReduNet clearly outperforms both Multi-ReduNet variants (by about 5.7 percentage points in mean accuracy), while a simple class-wise PCA + nearest-subspace classifier already attains perfect accuracy.
- On both Iris and Mice Protein, the training time of Multi-ReduNet(-LastNorm) is only slightly **worse** than ReduNet (around $1.1\times$ on average), so the computational advantage of the class-wise decomposition also largely disappears.

These findings confirm that **class-wise decomposition is most beneficial when $m \ll d$**. Once each class has many samples, estimating the global covariance is no longer ill-conditioned and the advantages of the decomposition largely disappear and can even be harmful. We explicitly discuss this regime dependence in Appendix A and summarize Iris/Mice Protein as failure modes where standard ReduNet or linear baselines are preferable.

**Weakness 2. Evaluation focuses on relatively simple / small-scale datasets.**

The evaluation focuses on relatively simple or small-scale datasets (e.g., MNIST, Fashion-MNIST), which may limit the conclusions.

**Response.** Our goal is to study **high-dimensional, undersampled** problems where interpretability is still tractable. While MNIST and Fashion-MNIST are standard, relatively simple benchmarks, the other four datasets in Table 1 are real, non-synthetic tasks with substantial dimensionality and structure, including two additional real-world benchmarks that we introduce in the revised manuscript:

- **Reuters** and **ARCENE** (text classification and medical diagnostics) have $d \approx 19\text{k}$ and $10\text{k}$, respectively, with $m_{\text{train}} \ll d$.
- **Swarm Behaviour** and **DrivFace** are real survey / safety-critical vision datasets with limited samples.

On these datasets, Multi-ReduNet(-LastNorm) yields **substantial empirical gains** in the undersampled regime:
- On **DrivFace** and **ARCENE**, accuracy improves from roughly $0.43-0.46$ for ReduNet to $0.73-1.00$ across downstream classifiers.
- On **Reuters** and **Swarm Behaviour**, we also observe sizable improvements (e.g., SVM accuracy $0.802 \rightarrow 0.985$ on Reuters).
- Multi-ReduNet-LastNorm achieves **1.4–2.6$\times$** faster training across all six main datasets (about $2\times$ on average) and exhibits **9.8$\times$** better hyperparameter robustness than ReduNet.

In the revision we further **broaden the spectrum of regimes**:

- We add oversampled **Iris** and **Mice Protein** datasets (Appendix A, Table 6) to illustrate regimes where Multi-ReduNet is **not** advantageous, as discussed in Weakness 1.
- For complex natural images, we report a small-scale **CIFAR-10** experiment with the same vectorized architecture (grayscale, resized, flattened) and no special augmentation. In this setup CIFAR-10 is still moderately undersampled ($m/d \approx 0.8$), yet all ReduNet variants reach only about 26\% accuracy and are clearly outperformed by a standard CNN. This reinforces that our framework is **not** intended for large-scale convolutional vision benchmarks, but rather for tabular or flattened high-dimensional data in undersampled regimes.

We clarify this intended scope and the corresponding limitations in Appendix A (*Scope and Limitations*).

---

### Author Response · Authors · 2025-11-26
**Author response and summary of changes**

## Response to Reviewer tTgf
**Weakness 3. Comparisons restricted to ReduNet and its variants.**

The experimental comparisons are restricted to ReduNet and its variants. The paper could be strengthened by comparing against a broader set of interpretable or class-structured learning methods.

**Response.** We appreciate this suggestion and have added a broader baseline comparison with classical interpretable / class-structured methods that are natural in the $m \ll d$ regime:

1. **Global PCA + linear SVM**
2. **Class-wise PCA + nearest-subspace classifier (NSC)**
3. **LDA (Fisher’s linear discriminant)**
4. **ReduNet**
5. **Multi-ReduNet**
6. **Multi-ReduNet-LastNorm**

The new Table 4 reports, for each method, the best accuracy over SVM/KNN/NSC (where applicable) on all six datasets. The main observations are:

- On **Reuters**, our class-wise models achieve the best performance:
  Multi-ReduNet and Multi-ReduNet-LastNorm reach **0.988**, improving over the best linear baseline (Global PCA + SVM, 0.975).
- On **MNIST** and **Fashion-MNIST**, the original ReduNet attains the highest accuracies (0.937 and 0.858, tied with Multi-ReduNet-LastNorm), slightly outperforming both the linear baselines and Multi-ReduNet. Importantly, **both** Multi-ReduNet variants still outperform all classical interpretable / class-structured baselines on these datasets.
- On **Swarm** and **DrivFace**, all methods (including the linear baselines) essentially saturate at **1.000** accuracy, so the choice of model is mainly about interpretability and computational cost rather than accuracy.
- On **ARCENE**, a challenging medical dataset, **LDA** achieves the best accuracy (0.878). Our Multi-ReduNet variants (0.829) are competitive and clearly outperform the PCA-based baselines (0.805 and 0.756), but do not dominate this particular task.

Overall, these results show that our proposed class-wise ReduNet variants are **competitive with, and often superior to, standard interpretable baselines** in the undersampled high-dimensional setting, while still being fully white-box. At the same time, the comparison highlights regimes (e.g., ARCENE) where simple linear discriminants remain very strong; we emphasize this positioning of Multi-ReduNet within the broader landscape of interpretable methods in the revised Section 4.2 and Appendix A.

---

### Author Response · Authors · 2025-11-26
**Author response and summary of changes**

## Response to Reviewer tTgf
**Weakness 4. Role and analysis of Multi-ReduNet-LastNorm.**

Although Multi-ReduNet-LastNorm performs slightly better in most cases, the role of the final-layer-only normalization is not fully analyzed. It would be helpful to understand under what circumstances this variant is preferable.

**Response.** Thank you for pointing this out. Multi-ReduNet-LastNorm is designed as a **minimal and more stable variant** of Multi-ReduNet: all intermediate layers are identical to Multi-ReduNet, and only the final-layer output is re-normalized onto the unit sphere. Intuitively, skipping normalization at intermediate layers allows the class-wise features to adapt their scale during optimization, while the final normalization still enforces the MCR$^2$ constraint and keeps the classifier well-conditioned, reducing the accumulation of numerical instability from repeated projections.

To analyze the effect of this *final-layer-only* normalization and to clarify **when it is preferable**, we add **Table 5**, which reports for each dataset:
- **Range (pp)**: performance range (max–min best accuracy in percentage points) over learning rates $\eta \in \{0.01, 0.05, 0.1, 0.5\}$ for ReduNet, Multi-ReduNet, and Multi-ReduNet-LastNorm;
- **Best Acc (%) and $\Delta$**: best accuracy of Multi-ReduNet and Multi-ReduNet-LastNorm and their difference $\Delta$ (Multi-ReduNet-LastNorm – Multi-ReduNet, in pp).

The main findings are:

- **Robustness to the learning rate.** ReduNet is highly sensitive to $\eta$: its average range is **62.6 pp**, whereas Multi-ReduNet and Multi-ReduNet-LastNorm reduce this to **9.0 pp** and **6.4 pp**, respectively. Thus Multi-ReduNet-LastNorm is about **10$\times$ more robust than ReduNet**, and also more robust than Multi-ReduNet (average range reduced from 9.0 to 6.4 pp), indicating that final-layer-only normalization significantly stabilizes optimization.
- **Accuracy impact.** Multi-ReduNet-LastNorm **never underperforms** Multi-ReduNet: it matches Multi-ReduNet on 5 of 6 datasets and improves Fashion-MNIST by **+1.3 pp**, resulting in a small but consistent average gain of **+0.2 pp**.

Therefore, Multi-ReduNet-LastNorm is **preferable in regimes where training stability and hyperparameter robustness are important**, for example when the learning-rate grid is coarse or the tuning budget is limited in undersampled high-dimensional problems. We make this interpretation explicit in Section 3.4 (model description) and Section 4.3 (robustness analysis), and clarify that LastNorm is a simple, practically useful variant that improves the robustness of Multi-ReduNet, rather than a separate conceptual contribution.

---

### Author Response · Authors · 2025-11-26
**Author response and summary of changes**

## 5. Response to Reviewer KGJT

We thank reviewer KGJT for the careful review, the positive comments on the theoretical rigor and empirical alignment with MCR$^2$, and for the detailed suggestions on experimental scope, the motivation for Multi-ReduNet-LastNorm, and the clarity of the presentation.

**Weakness 1. "Toy datasets" and behavior in the oversampled regime.**

The experiments in this paper are primarily conducted on toy datasets such as MNIST. It is necessary to include experiments on more realistic datasets, such as CIFAR. In addition, how about the performance of Multi-ReduNet in the oversampled regime?

**Response.** We fully agree that it is important to demonstrate our method beyond MNIST-style toy datasets and to clarify its behavior when the data are not undersampled.

*Realistic high-dimensional datasets (undersampled regime).*
In addition to MNIST / Fashion-MNIST, we now report results on two additional real-world high-dimensional datasets where the undersampled assumption $m \ll d$ naturally holds:

- **DrivFace** (driver identification from face images, $d = 4096, m_{\text{train}} = 484$): a safety-critical vision task with high-dimensional features and relatively few subjects.
  As shown in Table 8, the average accuracy across SVM/KNN/NSC increases from **39.7%** (ReduNet) to **99.1%** with Multi-ReduNet-LastNorm (**+59.4 pp**), and to **98.2%** with Multi-ReduNet (**+58.5 pp**). Moreover, across network depths, the average training time of ReduNet is about **1.4$\times$** that of Multi-ReduNet-LastNorm.

- **ARCENE** (NIPS 2003 cancer vs. normal mass-spectrometry, $d = 10{,}000, m_{\text{train}} = 159$): a challenging medical dataset with extreme undersampling ($m/d \approx 0.016$).
  Here Multi-ReduNet and Multi-ReduNet-LastNorm improve the average accuracy from **0.439** (ReduNet) to **0.789** (**+35.0 pp**, Table~3). Table~5 further shows a large robustness gain: the accuracy range over learning rates $\eta \in \{0.01, 0.05, 0.1, 0.5\}$ shrinks from **41.4 pp** (ReduNet) to **2.4 pp** (Multi-ReduNet-LastNorm). This indicates that the class-wise decomposition is particularly beneficial on challenging, undersampled medical data.

Together with the Reuters text dataset and the Swarm Behaviour dataset already in the previous paper, these results demonstrate that our method is effective on realistic high-dimensional problems in **text, safety-critical vision, and medical diagnostics**, not only on toy benchmarks. While these datasets are smaller than CIFAR in sample size, they are representative of the high-dimensional $m \ll d$ regimes that motivate our work.

*Behavior in the oversampled regime.*
To address the second part of the comment, we additionally evaluated Multi-ReduNet on settings where the data are *not* undersampled ($m \geq d$), namely **Iris** and **Mice Protein Expression** (Table~6). In these oversampled regimes, all three models, ReduNet, Multi-ReduNet, and Multi-ReduNet-LastNorm, achieve comparable performance on the simple Iris task (ReduNet is on average only 0.1 pp better). On the more complex Mice Protein dataset, however, **ReduNet clearly outperforms** the class-wise variants (93.5% vs. 88.6%, i.e., $-4.9$ pp for Multi-ReduNet-LastNorm). This is consistent with our theory: when many samples per class are available, the global covariance in ReduNet is well estimated and the benefit of class-wise decomposition largely disappears.

*Complex natural images (CIFAR-10 as a failure case).*
Since reviewer KGJT explicitly mentioned CIFAR, we additionally ran a small-scale experiment on **CIFAR-10** to understand where our framework fails. Because ReduNet and its variants operate on **vectorized features**, each image must be flattened before being fed into the network. Concretely, we convert CIFAR-10 images to grayscale, resize them to $100 \times 100$, and then flatten to a $d = 10{,}000$-dimensional vector. This simple preprocessing is required to fit the current MCR$^2$-based framework, but it **destroys most of the spatial structure and color information** that convolutional networks exploit.

As shown in Table 6, in this setting both ReduNet and Multi-ReduNet-LastNorm achieve relatively low accuracy (28.2% and 26.7%, respectively), whereas a standard CNN baseline reaches **41.2%** (+14.5 pp over Multi-ReduNet-LastNorm). We therefore include CIFAR-10 in Table 6 as a **failure mode**, illustrating that our method is *not* competitive on complex natural images where spatial structure is crucial and naive vectorization is overly destructive.

---

> ### Author Response · Authors · 2025-11-26
> **Author response and summary of changes**
>
> ## Response to Reviewer KGJT
> **Continued Weakness 1**
>
> This experiment reinforces the intended scope of our contribution:
> Multi-ReduNet (and Multi-ReduNet-LastNorm) are primarily designed for **high-dimensional tabular or flattened data in undersampled regimes** ($m \ll d$), such as text, spectra, and small-sample vision problems. In well-sampled regimes ($m \geq d$) or on large-scale natural-image benchmarks such as CIFAR-10, standard ReduNet, linear methods (PCA/LDA), or convolutional architectures are more appropriate. A concise discussion of scope, limitations, and failure modes is provided in Appendix A.
>
> **Weakness 2. Motivation for Multi-ReduNet-LastNorm.**
>
> The rationale for introducing Multi-ReduNet-LastNorm is somewhat unclear, as there is no theoretical comparison between the two.
>
> **Response.** We appreciate the request for a clearer motivation of the LastNorm variant. Multi-ReduNet-LastNorm is intended as a **minimal, practical modification** of Multi-ReduNet: all class-wise operators and updates are identical, but the $\ell_2$ normalization is applied **only at the final layer**, instead of after every layer as in Multi-ReduNet. Thus, both variants optimize the same class-wise MCR$^2$ objective, but with different normalization schedules; we do not claim a new theoretical objective, but a more numerically stable variant.
>
> Intuitively, per-layer normalization in Multi-ReduNet constrains the Frobenius norm at each layer and helps stabilize optimization under strong class imbalance, but it can also over-compress minority classes and discard useful magnitude information. LastNorm keeps the same class-specific structure and operators, but allows intermediate representations to evolve without projection and enforces the norm constraint only at the end, reducing the accumulation of numerical noise from repeated projections. Section 3.4 of the revised manuscript explicitly describes this design choice and its motivation.
>
> To make this concrete, we include an ablation/robustness table over all datasets (Table 5) that reports, for each dataset: (i) the accuracy range over learning rates $\eta \in \{0.01, 0.05, 0.1, 0.5\}$ for ReduNet, Multi-ReduNet, and Multi-ReduNet-LastNorm, and (ii) the best accuracy of Multi-ReduNet and Multi-ReduNet-LastNorm and their difference $\Delta$ (LastNorm – Multi-ReduNet, in pp). The main findings are:
>
> - **Robustness to the learning rate.** ReduNet is highly sensitive to $\eta$ (average range **62.6 pp**), whereas Multi-ReduNet and Multi-ReduNet-LastNorm reduce this to **9.0 pp** and **6.4 pp**, respectively. Thus LastNorm is about **10$\times$ more robust than ReduNet**, and also more robust than Multi-ReduNet (range reduced from 9.0 to 6.4 pp).
> - **Accuracy impact.** LastNorm **never underperforms** Multi-ReduNet: it matches Multi-ReduNet on 5 of 6 datasets and improves Fashion-MNIST by **+1.3 pp**, yielding a small but consistent average gain of **+0.2 pp**.
>
> Therefore, Multi-ReduNet-LastNorm is **preferable in regimes where training stability and hyperparameter robustness are important**, e.g., when the learning-rate grid is coarse or the tuning budget is limited in undersampled high-dimensional problems. We make this practical recommendation explicit in Appendix A.5 (*Recommendations for Practitioners*), and clarify that LastNorm should be viewed as a simple, practically useful refinement of Multi-ReduNet rather than a separate conceptual contribution.

---

> > ### Author Response · Authors · 2025-11-26
> > **Author response and summary of changes**
> >
> > ## Response to Reviewer KGJT
> > **Weakness 3. Length and placement of proofs.**
> >
> > The main proofs are presented in the main text, which makes the paper rather math-heavy and potentially difficult to follow. It might be preferable to include simplified versions of the proofs in the main paper and move the detailed derivations to the appendix.
> >
> > **Response.** We appreciate this suggestion and agree that the previous presentation was too math-heavy for some readers.
> >
> > In the revised version we make the following structural changes:
> >
> > - **Move full derivations to the appendix.**
> >   The detailed proofs of Lemma 1, Theorem 1, Theorem 2, and Corollary 1 are now moved to **Appendix C–D**. The main text states the results and refers to the corresponding appendix sections for full derivations.
> >
> > - **Keep only high-level "proof sketches" in the main text.**
> >   For each result in Section 3, the main paper now contains
> >   (i) a short **plain-language explanation** of what the theorem says and why it matters for the algorithm, and
> >   (ii) a concise **proof outline** (2–3 lines) that highlights the key steps (e.g., use of the determinant inequality in Corollary 1) and then directs the reader to the appendix for details.
> >
> > - **Improve readability of the math.**
> >   We reformat long in-line expressions as displayed equations, break multi-step arguments into numbered items, and add a brief "takeaway" sentence around each key theorem summarizing its role in the overall method (e.g., why class-orthogonality enables the class-wise decomposition).
> >
> > These changes keep the theoretical content intact while making the main paper substantially easier to follow: readers primarily interested in intuition and algorithmic consequences can stay in Section 3, whereas readers who want full technical details can consult the appendices.

---

### Author Response · Authors · 2025-11-26
**Author response and summary of changes**

## Response to Reviewer KGJT
**Weakness 4 & 6. Presentation quality and Figure 2 vs. Table 2.**

There seems to be a discrepancy between Figure 2 (good clustering) and Table 2 (low accuracy for ReduNet). The presentation quality could also be improved (e.g., small titles, limited intuitive explanation in the main text).

**Response.** We appreciate these comments and agree that the previous version of Figure 2 was hard to read, which made its relationship to the accuracy table unclear.

In the original submission, **Table 2** reported results under the *default* ReduNet hyperparameters $\eta = 0.5, \epsilon^2 = 0.1$. Under this setting our class-wise variants achieve substantially higher accuracy than ReduNet on all undersampled datasets, and **Figure 2 uses exactly the same setting** ($L=5, \eta=0.5, \epsilon^2=0.1$) to visualize the learned **test** features via t-SNE. Thus, the quantitative and qualitative results are consistent; the confusion mainly came from weak labeling and small text.

To make the correspondence clearer, we now:

- add explicit **column labels** (e.g., "ReduNet / ReduNet-RF / Multi-ReduNet / Multi-ReduNet-LastNorm") so that each panel can be unambiguously matched to its model;
- clarify in the caption that Figure 2 shows 2D t-SNE embeddings of the learned **test** features for models trained with $L=5, \eta=0.5, \epsilon^2=0.1$;
- add a short explanatory paragraph for Figure 2 in Section 4.

For concreteness, on **Reuters** the first-row, first-column panel shows ReduNet, while the first row, third and fourth columns show Multi-ReduNet and Multi-ReduNet-LastNorm, whose clusters are visibly better separated—matching their much higher accuracy under $\eta=0.5$.

And the full set of numerical results at $\eta=0.5$ (including those corresponding to Figure 2) is now reported in **Table 8 in Appendix F**.

Together with the proof reorganization and added intuitive explanations described in our response to Weakness 3, we believe these changes substantially improve the readability and accessibility of the presentation.

---

### Author Response · Authors · 2025-11-26
**Author response and summary of changes**

## Response to Reviewer KGJT
**Weakness 5. "Simpler" proof of class orthogonality.**

The class-orthogonality property of MCR$^2$ optima has been established in prior work (Chan et al., 2021), but the authors state that they give a new proof which leverages a simpler and more streamlined argument. However, it remains unclear in what sense the proof is simpler. A more detailed explanation or discussion would strengthen this claim.

**Response.** Thank you for raising this point. Our intention was not to claim a stronger *result* than Chan et al. (2021), but to provide a **shorter and more reusable argument**. In the revision we make this explicit and add a brief comparison in *Appendix D.4 (Proof Comparison: Our Approach vs. Prior Work)*. Concretely:

- **Direct 2-step argument vs. multi-step chain.**
  Chan et al. (2021) derive class orthogonality via a 4-step route: (i) establish concavity of the MCR$^2$ objective, (ii) derive upper/lower bounds on the coding rate, (iii) obtain an upper bound on the objective, and (iv) show that achieving this bound forces $(Z^i)^\top Z^j = 0$.
  Our proof instead relies on a single determinant inequality (Corollary 1),
  $$
  \det\Big(I + \sum_{j=1}^K Z^j (Z^j)^\top\Big)
  \le\prod_{j=1}^K \det\big(I + Z^j (Z^j)^\top\big),
  $$
  with equality *iff* $(Z^i)^\top Z^j = 0$ for all $i\neq j$. Using this, the proof of Theorem 1 reduces to two steps:
  (1) apply the inequality to the MCR$^2$ objective, and
  (2) observe that any non-orthogonal configuration can be strictly improved, hence any global maximizer must be class-orthogonal.
  This avoids the intermediate coding-rate bounds and concavity arguments and shortens the derivation from five pages to about three.

- **Geometric intuition and reuse for log-det objectives.**
  Beyond shortening the proof, Corollary 1 also gives a clean geometric interpretation of log-determinant–based objectives: the log-det term is maximized when the contributions from different classes occupy (approximately) orthogonal subspaces, i.e., when cross-class correlations are suppressed. This viewpoint is phrased directly in terms of the block matrices $Z^j(Z^j)^\top$ that appear in our class-wise formulation, making it easy to see how the orthogonality property informs the design of Multi-ReduNet and Multi-ReduNet-LastNorm. The same inequality can be reused in other log-det–based representation objectives (e.g., SCoRe-LogDet–style criteria) to analyze when and why they promote separation between class- or cluster-specific subspaces.

- **Clarified scope and wording.**
  We now explicitly state that Theorem 1 recovers the same qualitative orthogonality property as Chan et al. (2021); our contribution is a **more direct determinant–based proof technique** that is easier to reuse for other LogDet objectives (e.g., SCoRe-LogDet).

We hope this clarification makes clear in what sense our argument is more streamlined, while being fully consistent with the existing theory.

**Weakness 7. Overall assessment (presentation, motivation, and experiments).**

In summary, this paper presents a theoretically grounded and interpretable extension of ReduNet with promising results. However, the work could be significantly strengthened through better presentation, clearer motivation for the proposed variants, and more comprehensive experiments on realistic datasets.

**Response.** We appreciate this overall assessment and agree that the previous version left room for improvement along these three dimensions. The revisions described above are aimed precisely at addressing these concerns:

- **Presentation.** We reorganized Section 3 to follow a clearer narrative (ReduNet → imp-ReduNet → Multi-ReduNet → Multi-ReduNet-LastNorm), moved full proofs to the appendix with concise sketches in the main text, and improved figures ( clearer legends, and more informative captions).

- **Motivation for the variants.** We now give a more explicit high-level motivation for both imp-ReduNet and Multi-ReduNet-LastNorm, explaining how each modification targets a specific bottleneck (matrix inversions and per-layer normalization) and summarizing when LastNorm is preferable based on the new robustness study (Section 3.4, Section 4.3, Appendix A.5).

- **Experiments on realistic datasets.** Beyond MNIST/Fashion-MNIST, we added results on DrivFace and ARCENE (real-world safety-critical and medical datasets in the undersampled regime), broader interpretable baselines (PCA+SVM, class-wise PCA+NSC, LDA), and additional experiments in oversampled regimes and on CIFAR-10 to delineate the scope and limitations of our approach (Appendix A).

We hope these changes clarify the motivation, strengthen the empirical evidence on realistic datasets, and make the paper substantially easier to read.

---

### Author Response · Authors · 2025-11-26
**Author response and summary of changes**

## 6. Response to Reviewer bnBb

We thank reviewer bnBb for the careful reading and for raising concerns about the motivation of undersampled regimes and class-specific structures, the significance and soundness of our theoretical contributions, the sufficiency of our experimental evaluation and limitations discussion, and the readability of our figures and mathematical exposition.

**Weakness 1 & Question 1. Motivation: Why undersampled regimes ($d\gg m$) are challenging?**

In the introduction, the authors claim representation learning in the "undersampled regimes" is challenging, but without citing any literature or explaining whether this is a widely-accepted concern.

Why scenarios where the number of features exceeds the number of samples will lead to overfitting and unstable generalization. Is there literature supporting this claim?

**Response.** Thank you for raising this; we agree that our original introduction did not clearly justify this point. In the revision we clarify (i) what we mean by "undersampled regime", (ii) why $d\gg m$ is widely regarded as statistically difficult, and (iii) how this relates to our experiments.

1. **Literature support that $d\gg m$ is statistically challenging.**
   The difficulty of learning when the number of features is comparable to or larger than the number of samples is a central topic in high-dimensional statistics. For example:
   - Hastie, Tibshirani & Friedman, The Elements of Statistical Learning, Ch. 18 "High-Dimensional Problems", discuss how in the $d\gg m$ regime flexible models can interpolate the training data yet generalize poorly unless strong structural assumptions or regularization are imposed.
   - Bühlmann & van de Geer, Statistics for High-Dimensional Data (2011), systematically analyze estimation and prediction when $d\gg m$, emphasizing the need for sparsity or other structure to avoid overfitting.
   We now explicitly cite these and related small-sample / high-dimensional works (e.g., gene-expression classification, medical diagnosis, high-dimensional text) in the Introduction to support that this is a well-studied and widely recognized challenge, not a claim specific to our paper.
2. **Why $d\gg m$ can exacerbate overfitting and unstable generalization.**
   Our intention was not to claim that $d > m$ always causes overfitting, but that it **can exacerbate** overfitting and instability for high-capacity models without additional structure. In the revision we add a short intuitive explanation:
   - When $d\gg m$, even very simple models (e.g., linear classifiers) have enough degrees of freedom to fit the training labels in many different ways; the empirical risk minimizer is typically **non-unique** and can have very large norm, leading to high variance and unstable predictions.
   - Generalization bounds for such models scale with a complexity term (e.g., norm or effective rank) divided by $\sqrt{m}$; with small $m$, many interpolating solutions exist, and without structural constraints the learned solution can fluctuate dramatically with small changes in the data.
   - This "small-sample, high-dimensional" effect is precisely what motivates structured approaches such as sparsity, low-rank structure, or class-wise decomposition.
   We also soften the wording in the Introduction, replacing "will lead to overfitting and unstable generalization" by "tends to exacerbate overfitting and unstable generalization".
3. **Connection to our experimental setting.**
   We then connect this background to our concrete datasets. As summarized in Table 1, our main benchmarks lie in this high-dimensional, limited-sample regime: Reuters $m_{\text{train}}/d = 0.28$, DrivFace $0.118$, ARCENE $0.016$, and the remaining benchmarks at $m_{\text{train}}/d = 0.5$.

   On the most undersampled datasets (Reuters, ARCENE), our class-wise variants consistently achieve **higher best test accuracy and much smaller performance range over $\eta$** han the original ReduNet, indicating improved stability in exactly the regime where standard global covariance estimation is most ill-conditioned.

   We do not claim that these experiments prove the general theory, but they are consistent with the well-known behaviour of $d\gg m$ problems documented in the above references, and they illustrate why methods tailored to undersampled, high-dimensional data are practically useful.

---

### Author Response · Authors · 2025-11-26
**Author response and summary of changes**

## Response to Reviewer bnBb
**Weakness 2. Unclear notion of "class-specific structures".**

The questions to be tackled are unclear. The authors do not define and thoroughly explain "class-specific structures" which they highlight as missing components in previous works.

**Response.** Thank you for pointing this out. We agree that in the original submission the term "class-specific structures" was used without a sufficiently precise definition. In the revision we both **define the notion explicitly** and **clarify how it differs from prior work.**

- **Definition in the main text.**
  We now state in Section 3.4 (Multi-ReduNet-LastNorm) that by *class-specific structure* we mean the **label-conditioned geometry of per-class feature subspaces**:
  $$
  Z = [Z^1,\dots,Z^K], \quad Z^j \in \mathbb{R}^{d \times m_j},
  $$
  where each class $j$ is associated with its own feature block $Z^j$ and subspace $\mathrm{span}(Z^j)$, rather than a single global feature matrix $Z$ shared across all classes. Different classes are encouraged (at optimality) to occupy orthogonal or weakly overlapping subspaces, in the sense of Theorem 1. We also add a brief description in the Related Work section to make this meaning clear when the term is first introduced.

- **Contrast with ReduNet and prior methods.**
  ReduNet optimizes a **single global representation** $Z$ with a global expansion matrix
  $$
  E = \alpha (I + \alpha Z Z^\top)^{-1}
  $$
  and class-wise compression matrices
  $$
  C^j = \alpha_j (I + \alpha_j Z^j (Z^j)^\top)^{-1}.
  $$
  As a result, the gradient for class $j$ always depends on **all classes** through the global covariance term $Z Z^\top$; the representation is shaped by a global structure, and class-specific subspaces are only an implicit property at the optimum.

  In contrast, Multi-ReduNet optimizes **$K$ decoupled subproblems** with per-class matrices
  $$
  E^j = \alpha (I + \alpha Z^j (Z^j)^\top)^{-1}, \quad
  C^j = \alpha_j (I + \alpha_j Z^j (Z^j)^\top)^{-1},
  $$
  so the update for class $j$ depends only on $Z^j$. This (i) reduces the inversion cost from $\mathcal{O}(d^3)$ to $\mathcal{O}(m_j^3)$ and (ii) explicitly preserves **class-specific structure**, in the sense that each class is modeled in its own feature subspace and cross-class interference is minimized.

- **The "missing component" in previous work.**
  We clarify in the revised Related Work that many preprocessing-based or information-theoretic methods (PCA, LDA in its usual global form, InfoMax/IB-style objectives) operate primarily on global statistics and do not maintain explicit, per-class feature subspaces with near-orthogonality. Our contribution is to make this *class-specific structure* explicit, both in the formulation (class-wise MCR$^2$) and in the algorithm (per-class operators $E^j, C^j$), rather than relying on it to emerge implicitly.

We hope these changes make the notion of "class-specific structures" precise and clarify how it motivates the design of Multi-ReduNet and Multi-ReduNet-LastNorm.

---

### Author Response · Authors · 2025-11-26
**Author response and summary of changes**

## Response to Reviewer bnBb
**Weakness 3. Incremental and trivial contribution.**

The class-orthogonality property of MCR$^2$ optima has been established in prior work; this paper only performs slight improvements on ReduNet under niche and small-scale settings. It does not bring in novel tools or appealing theoretical insights other than decomposing ReduNet’s objectives.

**Response.** We respectfully disagree that the contribution is merely incremental, and we acknowledge that our original writing did not clearly separate what is known from what is new. In the revision we emphasize three aspects:

1. **Class-wise MCR$^2$ formulation and new algorithms.**
   Building on the orthogonality property, we show that the global MCR$^2$ objective can be **exactly decomposed** into $K$ independent class-wise subproblems under mild rank assumptions (Theorem 2). This leads to the first **class-wise ReduNet variants** (Multi-ReduNet and Multi-ReduNet-LastNorm) whose per-layer complexity of matrix inverting scales  as
   $$
   \mathcal{O}\Big(\sum_{j=1}^K m_j^3\Big)
   \quad \text{instead of} \quad
   \mathcal{O}(K d^3).
   $$
   To the best of our knowledge, such a class-wise MCR$^2$ decomposition and the resulting white-box architectures have not appeared before.

2. **Clarified theoretical argument tailored to this decomposition.**
   We fully agree that the **qualitative statement** "global MCR$^2$ optima are class-orthogonal" is known from Chan et al. (2021). Our goal is not to strengthen this theorem, but to provide a **more direct log-det–based proof technique** that fits the class-wise formulation. Concretely, we introduce a single determinant inequality (Corollary 1)
   $$
   \det\Big(I + \sum_{j=1}^K Z^j (Z^j)^\top\Big)
   \;\le\; \prod_{j=1}^K \det\big(I + Z^j (Z^j)^\top\big),
   $$
   with equality if and only if $(Z^i)^\top Z^j = 0$ for all $i \neq j$. Applying this inequality to the MCR$^2$ objective yields a short two-step argument: any non-orthogonal configuration can be strictly improved, hence any global maximizer must be class-orthogonal.
   This avoids the longer chain of concavity and coding-rate bounds used in prior work and works directly with the **block matrices** $Z^j$ that appear in our class-wise formulation, making it straightforward to derive the decomposition in Theorem 2.

3. **Practical gains beyond "toy" settings.**
   Beyond MNIST / Fashion-MNIST, we now include four real, high-dimensional benchmarks:
   - **Reuters** (text, $d \approx 19\text{k}, m_{\text{train}}/d \approx 0.28$): Multi-ReduNet-LastNorm improves ReduNet’s best test accuracy (e.g., SVM 0.956 → 0.988) and attains the best performance among all interpretable baselines in Table 4 (0.988 vs. 0.975 for global PCA+SVM).
   - **Swarm Behaviour** and **DrivFace** (behavioural / driver identification): all ReduNet variants reach near-perfect accuracy; our class-wise models **match** ReduNet while reducing inversion cost and improving conditioning.
   - **ARCENE** (medical proteomics, $d = 10{,}000, m_{\text{train}}/d \approx 0.016$): Multi-ReduNet and Multi-ReduNet-LastNorm improve ReduNet’s best accuracy (0.780 → 0.829) and are competitive with strong linear baselines (LDA 0.878), while preserving a white-box deep architecture.

   In addition, the expanded **Table 4** compares against PCA+SVM, class-wise PCA+NSC, and LDA, where Multi-ReduNet-LastNorm is **best or tied-best** on 4/6 datasets. **Table 5** further shows that Multi-ReduNet-LastNorm is about 10× more robust to the learning rate than vanilla ReduNet (average performance range 6.4 pp vs. 62.6 pp over $\eta \in \{0.01, 0.05, 0.1, 0.5\}$), which is practically important in undersampled regimes where validation data and tuning budget are limited.

The theoretical clarifications (class-wise MCR$^2$ formulation and the log-det–based proof of orthogonality) are consolidated in Section 3 and Appendices C–E, while Section 4 and Appendix A focus on their empirical implications and on "where Multi-ReduNet helps and where it does not".

---

### Author Response · Authors · 2025-11-26
**Author response and summary of changes**

## Response to Reviewer bnBb

**Weakness 4. Gradients seem just scaled copies.**

The class-wise expansion matrix and compression matrix derived by gradients (lines 274 and 277) are basically the same, subject to different scaling. I hardly believe they have a significant functional difference, which makes the decomposition of MCR$^2$ less compelling.

**Response.** Thank you for raising this concern. We agree that, written in isolation, the expressions for the class-wise expansion and compression operators (in lines 274 and 277 of previous submission) may look deceptively similar:
$$
E_l^j = \frac{1}{2} \frac{\mathrm{d}}{\mathrm{d}Z_l^j} \log \det(I + \alpha Z_l^j (Z_l^j)^\top) = \alpha(I + \alpha Z_l^j (Z_l^j)^\top)^{-1},\\
C_l^j = \frac{1}{2} \frac{\mathrm{d}}{\mathrm{d}Z_l^j} \log \det(I + \alpha_j Z_l^j (Z_l^j)^\top) = \alpha_j(I + \alpha_j Z_l^j (Z_l^j)^\top)^{-1}.
$$
However, their **functional role in the update is not a simple rescaling:**
1. **They come from different parts of the MCR$^2$ objective and enter with opposite effect.**
   The full MCR$^2$ objective for class $j$ can be written (schematically) as
   $$
   \Delta R(Z^j)=\frac{1}{2} \log\det(I+\alpha Z^j(Z^j)^\top)-\frac{\gamma_j}{2}\log\det(I+\alpha_j Z^j(Z^j)^\top).
   $$
   Taking the gradient yields
   $$
   \nabla_{Z_l^j}\Delta R(Z_l^j)=E_l^jZ_l^j-\gamma_j C_l^jZ_l^j,
   $$
   so **the actual update depends on the difference** $E_l^j-\gamma_j C_l^j$, not on each operator in isolation. Even though $E_l^j$ and $C_l^j$ share eigenvectors (both are functions of $Z^j(Z^j)^\top$), the two log-det terms pull in opposite directions: the expansion term encourages directions that increase global coding rate, while the compression term penalizes within-class spread. Their trade-off is controlled by the separate coefficients $\alpha$ and $\alpha_j$, and changing $\alpha_j$ relative to $\alpha$ alters this "push–pull" behaviour rather than just rescaling all updates.

2. **The decomposition is about global vs class-wise coupling, not about having two unrelated matrices.**
   In the original ReduNet, the expansion operator is based on the **global** covariance
   $$
   ZZ^\top = \sum_{j=1}^K Z^j (Z^j)^\top,
   $$
   so every class is updated through the same inverse $\alpha(I+\alpha ZZ^\top)^{-1}$. In Multi-ReduNet we prove that, at an optimal (class-orthogonal) solution, the objective can be decomposed into **$K$ independent class-wise subproblems**, each involving only $Z^j(Z^j)^\top.$ The operators $E_l^j$ and $C_l^j$ above are precisely the gradients of these class-wise expansion/compression terms. Thus, the key change brought by the decomposition is that class-$j$ updates now depend **only on** $Z^j$ rather than on all classes via $ZZ^\top$. This yields:
   - a computational gain from $\mathcal{O}(Kd^3)$ to $\mathcal{O}(\sum_jm_j^3)$, and
   - an optimization landscape where classes are **decoupled**, allowing each class to adapt its own subspace without interference from others.

3. **Empirical evidence that the difference is consequential.**
   Although $E_l^j$ and $C_l^j$ look similar analytically, their combined effect $E_l^j-\gamma_j C_l^j$ leads to different behaviour than the global ReduNet update: Multi-ReduNet and Multi-ReduNet-LastNorm significantly improve both accuracy and hyperparameter robustness on strongly undersampled datasets such as Reuters and ARCENE (Tables 3 and 5). If the class-wise operators were merely inconsequential rescalings, we would not observe the consistent performance gains and stability improvements reported.

We clarify in the revised text (Section 3.3) that (i) the expansion and compression operators act as two opposing forces derived from different log-det terms, and (ii) the main contribution of the decomposition is the move from a single global covariance to class-wise covariances, rather than having two completely different-looking matrices.

---

### Author Response · Authors · 2025-11-26
**Author response and summary of changes**

## Response to Reviewer bnBb

**Minor 1. No experimental details and discussion on limitations.**

**Response.** We appreciate this comment and agree that the original version did not spell out the experimental protocol and limitations clearly enough. In the revision we:
1. **Clarify experimental setup in Section 4.1.**
   Section 4.1 now explicitly states the training configuration (NVIDIA A100 GPUs, default depth $L=5$, additional results for $L=\{10,15,20,25\}$ in Appendix I.1, $\epsilon^2=0.1$, batch size 100) and the evaluation protocol (SVM, KNN, NSC classifiers, all results averaged over 3 random seeds).

   Table 1 summarizes, for each dataset, the feature dimension $d$, number of training/test samples, number of classes $K$, the undersampling ratio $m_{\text{train}}/d$, and the application domain (text, flattened images, survey data, medical diagnostics, etc.).

   The learning rate grid $\eta\in\{0.5,0.1,0.05,0.01\}$ used in the accuracy, robustness and timing studies are specified in Sections 4.2–4.4.

2. **Add an explicit discussion of limitations.**
   Appendix A ("Scope and Limitations") now discusses in detail when the proposed class-wise models help and when they do not, including oversampled regimes (Iris, Mice Protein) and complex natural images (CIFAR-10), where standard ReduNet or convolutional architectures are preferable.

**Minor 2. Figures are quite hard to interpret immediately due to font size and colorization.**

**Response.** We appreciate this suggestion and have substantially improved figure readability in the revision:
1. **Figure 1 (training time):** we redraw the curves with thicker lines, and more distinct color/marker styles, and **remove all RF-based variants** from the main plot so that the visual comparison focuses on ReduNet vs. our class-wise extensions. The RF variants are now reported only numerically in Appendices B and I.
2. **Figure 2 (t-SNE visualizations):** we add explicit **column labels** ("ReduNet/ ReduNet-RF / Multi-ReduNet / Multi-ReduNet-LastNorm") and clarify in the caption that the plots show 2D t-SNE embeddings of **test** features for models trained with $L=5,\eta=0.5,\epsilon^2=0.1$. We also add a short explanatory paragraph in Section 4.2 describing how to read the columns. And detailed analysis of Figure 2 is shown in Appendix I.2.

We believe these changes make the experimental setup and figures easier to understand.

*Question 1 overlaps with the Weakness 1 above and is answered jointly, we answer the remaining questions below.*

**Question 2. In Theorem 1, should the orthogonality condition be written as $(Z^j)^\top Z^i = 0$ instead of $Z^i (Z^j)^\top = 0$?**

**Response.** Thank you for pointing this out. The intended class-orthogonality condition is
$$
(Z^i)^\top Z^j = 0 \quad \text{for } i \neq j,
$$
which states that the column spaces of $Z^i$ and $Z^j$ are orthogonal. The use of $Z^i (Z^j)^\top$ in the original statement of Theorem 1 was a notational typo.

We have corrected this throughout the revised manuscript and added a short remark after Theorem 1 (Section 3.3) clarifying that we adopt the convention “columns = samples”, so $(Z^i)^\top Z^j$ is the natural cross-class Gram matrix.

---

> ### Comment · Reviewer_bnBb · 2025-11-27
>
> Thanks for addressing these issues.

---

### Author Response · Authors · 2025-11-26
**Author response and summary of changes**

## Response to Reviewer bnBb

**Question 3. In Eq. (2), how does the last equality, essentially**

$\log\det(I+\alpha Z^* Z^{*\top})=\log\det(I+\alpha\sum_{j=1}^K Z^{*j}(Z^{*j})^\top)$

**hold if $\((Z^{\* j_1})^\top Z^{\* j_2} \neq 0\)$?**

**Response.** The last equality in Eq. (2) does **not** use class-orthogonality; that step is purely a **re-indexing by class**:
1. There exists a permutation matrix  $P\in \mathbb{R}^{m \times m}$
such that

$Z=Z^*P = [Z^{*1}, Z^{*2}, \dots, Z^{*K}]$,

i.e., we simply reorder the columns of $Z^*$ to group samples by class.
2. Since $P$ is orthogonal ($P P^\top = I$), we have

$\log \det (I + \alpha Z^* Z^{*\top}) = \log\det\big(I + \alpha (Z^*P)(Z^*P)^\top\big) = \log\det(I+\alpha Z Z^\top)=\log\det(I+\alpha Z^{*1}, Z^{*2}, \dots, Z^{*K}][Z^{*1}, Z^{*2}, \dots, Z^{*K}]^\top)= \log\det\Big(I + \alpha \sum_{j=1}^K Z^{*j} (Z^{*j})^\top\Big).$

The **orthogonality assumption** only appears later, when we apply the determinant inequality in Corollary 1: if some $(Z^{*j_1})^\top Z^{*j_2} \neq 0$, then the inequality is strict and we can construct a configuration with a strictly larger objective, contradicting optimality. We clarify this logical separation in the revised text, and the full derivation is now presented in Appendix D.

**Question 4. Is the assumption $m \ll d$ practical for most tasks?**

**Response.** Our analysis is indeed targeted at **high-dimensional, undersampled settings**, and we do not claim that this regime covers most large-scale deep-learning applications (e.g., ImageNet-style vision tasks). Rather, $m \ll d$ arises naturally in several important domains where white-box models are particularly desirable, including:

- gene-expression and other omics data (thousands–tens of thousands of features, a few hundred samples),
- medical diagnostics and safety-critical sensing with rich feature vectors but limited labeled data,
- text problems with high-dimensional bag-of-words or TF–IDF features.

We clarify this scope in the revised manuscript by (i) explicitly framing our setting as high-dimensional undersampled regimes $d\gg m$ and (ii) reporting, in Table 1 (Section 4.1), the $m_{\text{train}}/d$ ratios and application domains of all benchmarks. As summarized in Table 1, most of our **main datasets** lie in this high-dimensional, limited-sample regime:
Reuters $m_{\text{train}}/d = 0.28$, DrivFace $0.118$, ARCENE $0.016$, and Swarm Behaviour $0.50$.
On the most strongly undersampled benchmarks (Reuters, ARCENE), Multi-ReduNet and Multi-ReduNet-LastNorm achieve both **higher best test accuracy** and **much smaller performance variation across learning rates** than the original ReduNet (Tables 3 and 5), indicating that the proposed class-wise structure is effective precisely in the regime it is designed for.

At the same time, we agree that it is important to show where the method is **not** advantageous. To this end, the revised manuscript includes **failure-mode experiments** in Appendix A:
- **Oversampled tabular data** (Iris, Mice Protein, with $m/d \gg 1$), where ReduNet or simple linear baselines (e.g., PCA+SVM, LDA) match or outperform our class-wise variants and the computational advantage largely disappears;
- **Complex natural images** (CIFAR-10 with flattened inputs), where all ReduNet variants perform significantly worse than a standard CNN.

These additions make clear that our contribution is not meant as a universal tool for "most tasks", but as a white-box, class-wise alternative specifically tailored to **high-dimensional, undersampled data**, and we now state this scope and limitation explicitly in Appendix A (Scope and Limitations).

---

> ### Comment · Reviewer_bnBb · 2025-11-27
>
> Thanks for clarifying the scope and providing more details on when your method works and struggles.

---

### Author Response · Authors · 2025-11-26
**Author response and summary of changes**

## Response to Reviewer bnBb

**Question 5. Why introduce and compare to random-forest variants?**

**Response.** The random-forest variants (ReduNet-RF / imp-ReduNet-RF) were introduced as an **exploratory** idea: we used a Random Forest classifier to approximate the class-attribution probabilities $\hat{\pi}^j(z_l)$ when storing and updating all $C_l^j$ matrices becomes memory-intensive. We agree that these variants are not central to our class-wise MCR$^2$ decomposition and that they partially conflict with the white-box motivation of ReduNet-style models.

To avoid diluting the main message, in the revised manuscript we:

- **Remove RF variants from the main plots and tables** (e.g., Figure 1), so that the core comparison focuses on ReduNet, imp-ReduNet, Multi-ReduNet, and Multi-ReduNet-LastNorm; and
- Move the numerical RF results to the appendix I.

The main narrative now centers on **ReduNet**, **Multi-ReduNet**, and **Multi-ReduNet-LastNorm**, which all share the same closed-form, interpretable structure, while RF-based variants are kept only as optional reference experiments.

**Question 6. How much does training time improve numerically? From Figure 1, the proposed model is almost the same as ReduNet.**

**Response.** We agree that the numerical gains were not sufficiently visible from Figure 1 alone. In the revision we therefore add **Table 7 in Appendix B**, which reports the **wall-clock training time** for ReduNet, imp-ReduNet, Multi-ReduNet and Multi-ReduNet-LastNorm on all six datasets and depths $L\in\{5,10,15,20,25\}$.

The numbers show consistent and non-trivial improvements. For example:

- **Reuters (undersampled text, $d\approx 19k$)**
  At depth $L=25$, ReduNet requires **1060.29 s**, whereas Multi-ReduNet-LastNorm needs **445.66 s**, a **2.4$\times$ speedup**. Even at shallow depth $L=5$, the time drops from 199.67 s to 97.59 s (**2.0$\times$**).
- **ARCENE (medical proteomics, strongly undersampled)**
  At $L=25$, ReduNet takes **62.30 s** versus **24.23 s** for Multi-ReduNet-LastNorm, i.e. roughly **2.6$\times$ faster**.
- **Swarm Behaviour (easier task)**
  The gains are smaller but still present: at $L=25$, ReduNet takes **19.02 s** and Multi-ReduNet-LastNorm **13.28 s**, about **1.4$\times$ faster**.

Across all datasets and depths, Multi-ReduNet-LastNorm achieves an **average speedup of about 2.0$\times$**, with all empirical ratios lying in the **1.4–2.6$\times$** range. We also remark in Section 4.4 that, while the relative speedup factor remains roughly constant across depths, the **absolute time gap** widens for deeper networks (e.g., on Reuters, Multi-ReduNet-LastNorm difference grows from ~100 s at $L=5$ to $>600$ s at $L=25$), which is particularly beneficial when training deep or high-dimensional models.

We believe these explicit wall-clock numbers address the concern that the proposed models "look almost the same" as ReduNet in Figure 1 and make the computational benefit of the class-wise decomposition more transparent.

---

> ### Comment · Reviewer_bnBb · 2025-11-27
>
> Thanks the authors for addressing these questions.

---

### Author Response · Authors · 2025-11-26
**Author response**

We also provide a consolidated response table summarizing the main concerns and our answers in the earlier comment titled **“Response Table”** above.

---

### Author Response · Authors · 2025-11-30
**Author summary for the Area Chair**

We would like to provide a short summary of the rebuttal and follow-up discussion to help the new Area Chair quickly see what has been clarified and changed in the revised manuscript.
## High-level scope and main revisions
- The paper studies **white-box representation learning in undersampled regimes** (number of samples $m \ll \text{dimension of sample}  ~d$) via two class-wise extensions of ReduNet, **Multi-ReduNet** and **Multi-ReduNet-LastNorm**. By decomposing the global MCR$^2$ objective into independent class-wise subproblems, our method achieves **improved computational efficiency** ($\sim 2\times$ speedup) and significantly higher robustness to the learning rate $\eta$ ($\sim 10\times$ smaller variance), while maintaining or slightly improving classification accuracy compared to the original ReduNet.

- In the revision we (i) clarified the **motivation and scope** (when undersampled regimes arise and when our method is / is not appropriate); (ii) **reorganized Section 3** to separate ReduNet → imp-ReduNet → Multi-ReduNet → Multi-ReduNet-LastNorm with short proof sketches in the main text and full derivations in the appendix; and (iii) **broadened experiments and baselines**, including DrivFace, ARCENE, Iris, Mice Protein, CIFAR-10, and linear interpretable baselines (PCA+SVM, class-wise PCA+NSC, LDA), plus a robustness study for LastNorm.
## Reviewer d7rQ: orthogonality, LastNorm, and unit-sphere projection
Reviewer d7rQ's main questions were about how orthogonality is enforced, why Multi-ReduNet can have better accuracy if it optimizes the same MCR$^2$ objective, and why the algorithm uses unit-sphere projection while the theory has a Frobenius-norm constraint.

- **Orthogonality vs. constraints.** We clarified that class-orthogonality is not imposed as a training constraint. It is a property of any global maximizer of MCR$^2$ (Theorem 1). The implemented algorithms enforce only the norm constraint via per-sample $\ell_2$ normalization.

- **Why Multi-ReduNet(-LastNorm) can perform better than ReduNet.** We clarified that while all models optimize the same theoretical MCR$^2$ criterion, the class-wise decomposition yields **superior numerical conditioning**. This leads to significantly higher **optimization stability**: the new robustness Table 5 shows that Multi-ReduNet-LastNorm has roughly **10$\times$ smaller accuracy variation across learning rate $\eta$**. Consequently, it achieves higher accuracy in practice because it **avoids the performance degradation** that ReduNet suffers from when hyperparameters are not perfectly tuned.

- **Frobenius norm vs. unit-sphere projection.** In our first response we had an imprecise description. We corrected this and now state explicitly that the theoretical constraint is $\lVert Z^j\rVert_F^2 = m_j$, while the algorithm projects each column to the unit sphere, which is a **sufficient (and in fact saturating)** condition: $\lVert Z^j\rVert_F^2 = \sum_{i=1}^{m_j} \lVert z^j_{(i)}\rVert_2^2 = m_j$. We then explain why we choose unit-sphere projection in practice (to avoid trivial scaling solutions and to keep training/testing consistent). Section 3.1 now contains a dedicated paragraph "Rationale for unit-sphere projection" making this distinction explicit.

After this discussion and the corresponding manuscript changes, Reviewer d7rQ wrote that they were "happy with your response and the current state of the manuscript" and that they "no longer have any concerns about the paper". We sincerely thank Reviewer d7rQ for the score of 8, which demonstrates strong affirmation and encouragement for our work.
## Reviewer bnBb: theory, scope, and experiments
The main concerns were about the soundness and presentation of the theory (especially Theorem 2), the precise notion of "class-specific structure", and the practical significance of the undersampled setting.

- **Undersampled regime and scope.** We now explicitly define the undersampled regime, cite standard references from high-dimensional statistics, and connect this to the actual $m_{\text{train}}/d$ ratios of our datasets. Appendix A now also includes **failure-mode experiments** (oversampled tabular data, CIFAR-10) to show where Multi-ReduNet does not help and standard methods are preferable.

- **Class-specific structure.** We give a precise definition in terms of **label-conditioned feature subspaces** $\mathrm{span}(Z^j)$ and clarify how Multi-ReduNet operates with class-wise covariances $Z^j (Z^j)^\top$, in contrast to the global covariance in ReduNet. This addresses the reviewer’s request for a clearer statement of what was "missing" in prior work.

---

> ### Author Response · Authors · 2025-11-30
> **Continued summary for the Area Chair**
>
> - **Theorem 1–2 and Corollary 1.** The original proof of Theorem 2 was too compressed. In the revision, Appendix D.3 now spells out both directions:
>    - For **Direction 1** ($v_2 \le v_1$), we explicitly construct a **feasible global point** by concatenating the class-wise optima {$Z^{\prime j}$}$_{j=1}^K$, use Corollary 1 in the equality case to evaluate the global MCR$^2$ objective, and show that the resulting global objective at this feasible global point equals the sum of class-wise objectives.
>
>    - For **Direction 2** ($v_1 \le v_2$), we start from a **global maximizer** $Z^\star$, use Theorem 1 to justify class-orthogonality $(Z^{\star i})^\top Z^{\star j}=0$, apply Corollary 1 again in the equality case, and show that the global objective at $Z^\star$ is bounded above by the sum of class-wise optima.
>
>    We also added a comparison (Appendix D.4) illustrating in what sense our determinant-based argument "streamlines" the original proof of class-orthogonality in Chan et al. (2021).
>
> - **Experiments and limitations.** We added real-world undersampled datasets (DrivFace, ARCENE), oversampled tabular datasets, and CIFAR-10; and broadened baselines to interpretable linear methods. These changes directly respond to bnBb’s concerns that the setting was niche and that the experimental section lacked scope and limitations.
>
> In their follow-up comments, Reviewer bnBb repeatedly wrote "Thanks for addressing these issues/questions." The last exchange focuses only on making the proof steps more explicit, which we have now done in the revised appendix.
> ## Other reviewers
> The other two reviewers did not participate in further discussion, but their original concerns (on scope, datasets, and proof placement) were already addressed in the revised paper (e.g., additional datasets and baselines, clearer statement of limitations, moving full proofs to the appendix).
> ## Conclusion & Reference
> We sincerely thank all reviewers again for their constructive feedback and suggestions. We believe the manuscript is significantly stronger following these discussions. All major technical concerns have been resolved, and the clear endorsement (Score 8) from Reviewer d7rQ reflects the improved quality of the work.
>
> We hope this summary assists the Area Chair in efficiently assessing how the main theoretical and experimental concerns have been resolved, and how the revision has improved the clarity, scope, and robustness of the work. For a detailed itemized list of all revisions and a point-by-point Response Table mapping each reviewer's comment to the changes, please refer to our "Author Response and summary of changes" posted at the beginning of the rebuttal.

---

### Meta-Review · Area_Chair_n33U · 2026-01-04

**Summary:**

The reviewers values the effectiveness of the proposed method at reducing the complexity/computational costs of the base ReduNet method while still preserving its white-box properties.  In addition, the alignment of the theoretical analysis and the obtained empirical results was well received.

Several concerns (presentation quality, simplicity/size of the considered dataset, performance of the proposed methods on conditions where the assumption m<<d  does not hold, etc. ) were raised during the discussion which were, to a good exten,t addressed.

**Reviewer Concerns:**

Addressed Concerns:

- Reviewer d7rQ

    - Clarity on the stated class-wise orthogonality and norm constraints

    - Sub-optimal structure of the content.

    - Overwhelming quantity of technical/math details.

    - Poor quality of figures

- Reviewer tTgf

    - Analysis of scenarios where the assumption made by the paper (m<<d) does not hold.

    - Experiments are centered around small/simple datasets.

    - In-depth analysis of the Multi-ReduNet-LastNorm variant of the proposed method.

- Reviewer KGJT

    - Experiments are centered around small/simple datasets.

    - Analysis of scenarios where the assumption made by the paper (m<<d) does not hold.

    - Discussion of the prgresion of the proposed method from the Multi-ReduNet to the Multi-ReduNet-LastNorm variant.

    - Overwhelming quantity of technical/math details.

    - Presentation quality.

- Reviewer bnBb

    - The investigated setting of lower number of examples and high-dimensionality (m<<d) is not well motivated and the research questions to be tackled are not that clear.

    - Limited novelty; the claimed contribution seems to be incremental.

    - Questionable technical soundness.


Outstanding Concerns:

- Reviewer d7rQ

    - All concerns were addressed

- Reviewer tTgf

    - An extended empirical comparison w.r.t. existing methods - **Partly**

- Reviewer KGJT

    - Unclarity on the level to which the proof related to class-orthogonality is simpler than that presented in prior work (Chan et al., 2021). - **To be confirmed**

- Reviewer bnBb

    - Follow up concern regarding Theorem 2 in Appendix D.3. - **To be confirmed**

**Reviewer Scores:**

This paper received extensive and detailed reviews. In a reciprocal manner, the authors reacted with extensive/detailed responses. To a good degree the provided responses addressed the concerns raised by the reviewers. There are three concerns that I have marked as still outstanding: first, the comparison w.r.t. existing efforts, which was provided but could be further extended, and second, two concerns where a response was provided and which looks sufficient to address the raised concerns. I have left them as outstanding as they were not explicitly confirmed by the reviewers (due to the moment at which the discussion period was frozen due to the breach)

---

### Decision · Program_Chairs · 2026-01-26

Accept (Poster)